# On the Turing Completeness of Modern Neural Network Architectures

**Jorge Pérez, Javier Marinković, Pablo Barceló**
Department of Computer Science, Universidad de Chile  &  IMFD Chile
`{jperez,jmarinkovic,pbarcelo}@dcc.uchile.cl`

## Abstract

Alternatives to recurrent neural networks, in particular, architectures based on *attention* or *convolutions*, have been gaining momentum for processing input sequences. In spite of their relevance, the computational properties of these alternatives have not yet been fully explored. We study the computational power of two of the most paradigmatic architectures exemplifying these mechanisms: the *Transformer* (Vaswani et al., 2017) and the *Neural GPU* (Kaiser & Sutskever, 2016). We show both models to be Turing complete exclusively based on their capacity to compute and access internal dense representations of the data. In particular, neither the Transformer nor the Neural GPU requires access to an external memory to become Turing complete. Our study also reveals some minimal sets of elements needed to obtain these completeness results.

## 1 Introduction

There is an increasing interest in designing neural network architectures capable of learning algorithms from examples (Graves et al., 2014; Grefenstette et al., 2015; Joulin & Mikolov, 2015; Kaiser & Sutskever, 2016; Kurach et al., 2016; Dehghani et al., 2018). A key requirement for any such an architecture is thus to have the capacity of implementing arbitrary algorithms, that is, to be *Turing complete*. Turing completeness often follows for these networks as they can be seen as a control unit with access to an unbounded memory; as such, they are capable of simulating any Turing machine.

On the other hand, the work by Siegelmann & Sontag (1995) has established a different way of looking at the Turing completeness of neural networks. In particular, their work establishes that recurrent neural networks (RNNs) are Turing complete even if only a bounded number of resources (i.e., neurons and weights) is allowed. This is based on two conditions: (1) the ability of RNNs to compute *internal* dense representations of the data, and (2) the mechanisms they use for accessing such representations. Hence, the view proposed by Siegelmann & Sontag shows that it is possible to release the full computational power of RNNs without arbitrarily increasing its model complexity.

Most of the early neural architectures proposed for learning algorithms correspond to extensions of RNNs – e.g., Neural Turing Machines (Graves et al., 2014) –, and hence they are Turing complete in the sense of Siegelmann & Sontag. However, a recent trend has shown the benefits of designing networks that manipulate sequences but do not directly apply a recurrence to sequentially process their input symbols. Architectures based on *attention* or *convolutions* are two prominent examples of this approach. In this work we look at the problem of Turing completeness à la Siegelmann & Sontag for two of the most paradigmatic models exemplifying these features: the *Transformer* (Vaswani et al., 2017) and the *Neural GPU* (Kaiser & Sutskever, 2016).

The main contribution of our paper is to show that the Transformer and the Neural GPU are Turing complete based on their capacity to compute and access internal dense representations of the data. In particular, neither the Transformer nor the Neural GPU requires access to an external additional memory to become Turing complete. Thus the completeness holds for bounded architectures (bounded number of neurons and parameters). To prove this we assume that internal activations are represented as rational numbers with arbitrary precision. For the case of the Transformer we provide a direct simulation of a Turing machine, while for the case of the Neural GPU our result follows by simulating standard *sequence-to-sequence* RNNs. Our study also reveals some minimal sets of elements needed to obtain these completeness results. The computational power of Transformers and

of Neural GPUs has been compared in the current literature (Dehghani et al., 2018), but both are only informally used. Our paper provides a formal way of approaching this comparison.

For the sake of space, we only include sketch of some proofs in the body of the paper. The details for every proof can be found in the appendix.

**Background work** The study of the computational power of neural networks can be traced back to McCulloch & Pitts (1943) which established an analogy between neurons with hard-threshold activations and first order logic sentences, and Kleene (1956) that draw a connection between neural networks and finite automata. As mentioned earlier, the first work showing the Turing completeness of finite neural networks with linear connections was carried out by Siegelmann & Sontag (1992; 1995). Since being Turing complete does not ensure the ability to actually learn algorithms in practice, there has been an increasing interest in enhancing RNNs with mechanisms for supporting this task. One strategy has been the addition of inductive biases in the form of external memory, being the *Neural Turing Machine* (NTM) (Graves et al., 2014) a paradigmatic example. To ensure that NTMs are differentiable, their memory is accessed via a soft attention mechanism (Bahdanau et al., 2014). Other examples of architectures that extend RNNs with memory are the Stack-RNN (Joulin & Mikolov, 2015), and the (De)Queue-RNNs (Grefenstette et al., 2015). By Siegelmann & Sontag's results, all these architectures are Turing complete.

The Transformer architecture (Vaswani et al., 2017) is almost exclusively based on the attention mechanism, and it has achieved state of the art results on many language-processing tasks. While not initially designed to learn general algorithms, Dehghani et al. (2018) have advocated the need for enriching its architecture with several new features as a way to learn general procedures in practice. This enrichment is motivated by the empirical observation that the original Transformer architecture struggles to generalize to input of lengths not seen during training. We, in contrast, show that the original Transformer architecture is Turing complete, based on different considerations. These results do not contradict each other, but show the differences that may arise between theory and practice. For instance, Dehghani et al. (2018) assume fixed precision, while we allow arbitrary internal precision during computation. We think that both approaches can be complementary as our theoretical results can shed light on what are the intricacies of the original architecture, which aspects of it are candidates for change or improvement, and which others are strictly needed. For instance, our proof uses *hard attention* while the Transformer is often trained with *soft attention* (Vaswani et al., 2017). See Section 3.3 for a discussion on these differences.

The Neural GPU is an architecture that mixes convolutions and *gated recurrences* over tridimensional tensors. It has been shown that NeuralGPUs are powerful enough to learn decimal multiplication from examples (Freivalds & Liepins, 2018), being the first neural architecture capable of solving this problem end-to-end. The similarity of Neural GPUs and cellular automata has been used as an argument to state the Turing completeness of the architecture (Kaiser & Sutskever, 2016; Price et al., 2016). Cellular automata are Turing complete (Smith III, 1971; Ollinger, 2012) and their completeness is established assuming an unbounded number of cells. In the Neural GPU architecture, in contrast, the number of cells that can be used during a computation is proportional to the size of the input sequence (Kaiser & Sutskever, 2016). One can cope with the need for more cells by padding the Neural GPU input with additional (dummy) symbols, as much as needed for a particular computation. Nevertheless, this is only a partial solution, as for a Turing-complete model of computation, one cannot decide a priori how much memory is needed to solve a particular problem. Our results in this paper are somehow orthogonal to the previous argument; we show that one can leverage the dense representations of the Neural GPU cells to obtain Turing completeness without requiring to add cells beyond the ones used to store the input.

## 2 PRELIMINARIES

We assume all weights and activations to be rational numbers of arbitrary precision. Moreover, we only allow the use of rational functions with rational coefficients. Most of our positive results make use of the *piecewise-linear sigmoidal activation* function $\sigma : \mathbb{Q} \to \mathbb{Q}$, which is defined as

$$\sigma(x) = \begin{cases} 0 & x < 0, \\ x & 0 \leq x \leq 1, \\ 1 & x > 1. \end{cases} \tag{1}$$

We are mostly interested in *sequence-to-sequence* (seq-to-seq) neural network architectures that we next formalize. A seq-to-seq network $N$ receives as input a sequence $\boldsymbol{X} = (\boldsymbol{x}_1, \ldots, \boldsymbol{x}_n)$ of vectors $\boldsymbol{x}_i \in \mathbb{Q}^d$, for some $d > 0$, and produces as output a sequence $\boldsymbol{Y} = (\boldsymbol{y}_1, \ldots, \boldsymbol{y}_m)$ of vectors $\boldsymbol{y}_i \in \mathbb{Q}^d$. Most of these types of architectures require a *seed* vector $\boldsymbol{s}$ and some stopping criterion for determining the length of the output. The latter is usually based on the generation of a particular output vector called an *end of sequence* mark. In our formalization instead, we allow a network to produce a fixed number $r \geq 0$ of output vectors. Thus, for convenience we see a general seq-to-seq network as a function $N$ such that the value $N(\boldsymbol{X}, \boldsymbol{s}, r)$ corresponds to an output sequence of the form $\boldsymbol{Y} = (\boldsymbol{y}_1, \boldsymbol{y}_2, \ldots, \boldsymbol{y}_r)$. With this definition, we can view every seq-to-seq network as a *language recognizer* of strings as follows.

**Definition 2.1.** *A seq-to-seq language recognizer is a tuple $A = (\Sigma, f, N, \boldsymbol{s}, \mathbb{F})$, where $\Sigma$ is a finite alphabet, $f : \Sigma \to \mathbb{Q}^d$ is an embedding function, $N$ is a seq-to-seq network, $\boldsymbol{s} \in \mathbb{Q}^d$ is a seed vector, and $\mathbb{F} \subseteq \mathbb{Q}^d$ is a set of final vectors. We say that $A$ accepts the string $w \in \Sigma^*$, if there exists an integer $r \in \mathbb{N}$ such that $N(f(w), \boldsymbol{s}, r) = (\boldsymbol{y}_1, \ldots, \boldsymbol{y}_r)$ and $\boldsymbol{y}_r \in \mathbb{F}$. The language accepted by $A$, denoted by $L(A)$, is the set of all strings accepted by $A$.*

We impose two additional restrictions over recognizers. The embedding function $f : \Sigma \to \mathbb{Q}^d$ should be computed by a Turing machine in time linear w.r.t. the size of $\Sigma$. This covers the two most typical ways of computing input embeddings from symbols: the *one-hot encoding*, and embeddings computed by fixed feed-forward networks. Moreover, the set $\mathbb{F}$ should also be recognizable in linear-time; given a vector $\boldsymbol{f}$, the membership $\boldsymbol{f} \in \mathbb{F}$ should be decided by a Turing machine working in linear time with respect to the size (in bits) of $\boldsymbol{f}$. This covers the usual way of checking equality with a fixed end-of-sequence vector. We impose these restrictions to disallow the possibility of *cheating* by encoding arbitrary computations in the input embedding or the stop condition, while being permissive enough to construct meaningful embeddings and stoping criterions.

Finally, a class $\mathcal{N}$ of seq-to-seq neural network architectures defines the class $\mathcal{L}_\mathcal{N}$ composed of all the languages accepted by language recognizers that use networks in $\mathcal{N}$. From these notions, the formalization of Turing completeness of a class $\mathcal{N}$ naturally follows.

**Definition 2.2.** *A class $\mathcal{N}$ of seq-to-seq neural network architectures is Turing Complete if $\mathcal{L}_\mathcal{N}$ is exactly the class of languages recognized by Turing machines.*

Given an input sequence $\boldsymbol{X} = (\boldsymbol{x}_1, \ldots, \boldsymbol{x}_n)$, a seed vector $\boldsymbol{y}_0$, and $r \in \mathbb{N}$, an *encoder-decoder* RNN is given by the following two recursions

$$\boldsymbol{h}_0 = \boldsymbol{0}, \qquad \boldsymbol{h}_i = f_1(\boldsymbol{x}_i \boldsymbol{W} + \boldsymbol{h}_{i-1} \boldsymbol{V} + \boldsymbol{b}_1) \quad (\text{with } 1 \leq i \leq n) \tag{2}$$
$$\boldsymbol{g}_0 = \boldsymbol{h}_n, \qquad \boldsymbol{g}_t = f_2(\boldsymbol{g}_{t-1} \boldsymbol{U} + \boldsymbol{y}_{t-1} \boldsymbol{R} + \boldsymbol{b}_2), \qquad \boldsymbol{y}_t = O(\boldsymbol{g}_t) \quad (\text{with } 1 \leq t \leq r) \tag{3}$$

where $\boldsymbol{V}, \boldsymbol{W}, \boldsymbol{U}, \boldsymbol{R}$ are matrices, $\boldsymbol{b}_1$ and $\boldsymbol{b}_2$ are vectors, $O(\cdot)$ is an output function, and $f_1$ and $f_2$ are activations functions. Equation (2) is called the RNN-*encoder* and (3) the RNN-*decoder*.

The next Theorem follows by inspection of the proof by Siegelmann & Sontag (1992; 1995) after adapting it to our formalization of encoder-decoder RNNs.

**Theorem 2.3** (Siegelmann & Sontag (1992; 1995))**.** *The class of encoder-decoder RNNs is Turing complete. Turing completeness holds even if we restrict to the class in which $\boldsymbol{R}$ is the zero matrix, $\boldsymbol{b}_1$ and $\boldsymbol{b}_2$ are the zero vector, $O(\cdot)$ is the identity function, and $f_1$ and $f_2$ are the piecewise-linear sigmoidal activation $\sigma$.*

## 3 THE TRANSFORMER ARCHITECTURE

In this section we present a formalization of the Transformer architecture (Vaswani et al., 2017), abstracting away from specific choices of functions and parameters. Our formalization is not meant to produce an efficient implementation of the Transformer, but to provide a simple setting over which its mathematical properties can be established in a formal way.

The Transformer is heavily based on the attention mechanism introduced next. Consider a scoring function $\text{score} : \mathbb{Q}^d \times \mathbb{Q}^d \to \mathbb{Q}$ and a normalization function $\rho : \mathbb{Q}^n \to \mathbb{Q}^n$, for $d, n > 0$. Assume that $\boldsymbol{q} \in \mathbb{Q}^d$, and that $\boldsymbol{K} = (\boldsymbol{k}_1, \ldots, \boldsymbol{k}_n)$ and $\boldsymbol{V} = (\boldsymbol{v}_1, \ldots, \boldsymbol{v}_n)$ are tuples of elements in $\mathbb{Q}^d$.

A $\boldsymbol{q}$-*attention* over $(\boldsymbol{K}, \boldsymbol{V})$, denoted by $\mathrm{Att}(\boldsymbol{q}, \boldsymbol{K}, \boldsymbol{V})$, is a vector $\boldsymbol{a} \in \mathbb{Q}^d$ defined as follows.

$$(s_1, \ldots, s_n) \quad = \quad \rho(\mathrm{score}(\boldsymbol{q}, \boldsymbol{k}_1), \mathrm{score}(\boldsymbol{q}, \boldsymbol{k}_2), \ldots, \mathrm{score}(\boldsymbol{q}, \boldsymbol{k}_n)) \tag{4}$$

$$\boldsymbol{a} \quad = \quad s_1 \boldsymbol{v}_1 + s_2 \boldsymbol{v}_2 + \cdots + s_n \boldsymbol{v}_n. \tag{5}$$

Usually, $\boldsymbol{q}$ is called the *query*, $\boldsymbol{K}$ the *keys*, and $\boldsymbol{V}$ the *values*. We do not pose any restriction on the scoring and normalization functions, as some of our results hold in general. We only require the normalization function to satisfy that there is a function $f_\rho$ from $\mathbb{Q}$ to $\mathbb{Q}^+$ such that for each $\boldsymbol{x} = (x_1, \ldots, x_n) \in \mathbb{Q}^n$ it is the case that the $i$-th component $\rho_i(\boldsymbol{x})$ of $\rho(\boldsymbol{x})$ is equal to $f_\rho(x_i) / \sum_{j=1}^n f_\rho(x_j)$. Thus, $\boldsymbol{a}$ in Equation (5) is a convex combination of the vectors in $\boldsymbol{V}$.

When proving possibility results, we will need to pick specific scoring and normalization functions. A usual choice for the scoring function is a feed forward network with input $(\boldsymbol{q}, \boldsymbol{k}_i)$ sometimes called *additive attention* (Bahdanau et al., 2014). Another possibility is to use the dot product $\langle \boldsymbol{q}, \boldsymbol{k}_i \rangle$ called *multiplicative attention* (Vaswani et al., 2017). We use a combination of both: multiplicative attention plus a non linear function. For the normalization function, $\mathrm{softmax}$ is a standard choice. Nevertheless, in our proofs we use the $\mathrm{hardmax}$ function, which is obtained by setting $f_{\mathrm{hardmax}}(x_i) = 1$ if $x_i$ is the maximum value, and $f_{\mathrm{hardmax}}(x_i) = 0$ otherwise. Thus, for a vector $\boldsymbol{x}$ in which the maximum value occurs $r$ times, we have that $\mathrm{hardmax}_i(\boldsymbol{x}) = \frac{1}{r}$ if $x_i$ is the maximum value of $\boldsymbol{x}$, and $\mathrm{hardmax}_i(\boldsymbol{x}) = 0$ otherwise. We call it *hard attention* whenever $\mathrm{hardmax}$ is used as normalization function. As customary, for a function $F : \mathbb{Q}^d \to \mathbb{Q}^d$ and a sequence $\boldsymbol{X} = (\boldsymbol{x}_1, \boldsymbol{x}_2, \ldots, \boldsymbol{x}_n)$, with $\boldsymbol{x}_i \in \mathbb{Q}^d$, we write $F(\boldsymbol{X})$ to denote the sequence $(F(\boldsymbol{x}_1), \ldots, F(\boldsymbol{x}_n))$.

**Transformer Encoder and Decoder** A *single-layer encoder* of the Transformer is a parametric function $\mathrm{Enc}(\boldsymbol{X}; \boldsymbol{\theta})$ receiving a sequence $\boldsymbol{X} = (\boldsymbol{x}_1, \ldots, \boldsymbol{x}_n)$ of vectors in $\mathbb{Q}^d$ and returning a sequence $\boldsymbol{Z} = (\boldsymbol{z}_1, \ldots, \boldsymbol{z}_n)$ of the same length of vectors in $\mathbb{Q}^d$. In general, we consider the parameters in $\mathrm{Enc}(\boldsymbol{X}; \boldsymbol{\theta})$ as functions $Q(\cdot), K(\cdot), V(\cdot)$, and $O(\cdot)$, all of them from $\mathbb{Q}^d$ to $\mathbb{Q}^d$. The single-layer encoder is then defined as follows

$$\boldsymbol{a}_i \quad = \quad \mathrm{Att}(Q(\boldsymbol{x}_i), K(\boldsymbol{X}), V(\boldsymbol{X})) + \boldsymbol{x}_i \tag{6}$$

$$\boldsymbol{z}_i \quad = \quad O(\boldsymbol{a}_i) + \boldsymbol{a}_i \tag{7}$$

In practice $Q(\cdot), K(\cdot), V(\cdot)$ are typically matrix multiplications, and $O(\cdot)$ a feed-forward network. The $+ \boldsymbol{x}_i$ and $+ \boldsymbol{a}_i$ summands are usually called *residual connections* (He et al., 2016a;b). When the particular functions used as parameters are not important, we simply write $\boldsymbol{Z} = \mathrm{Enc}(\boldsymbol{X})$.

The Transformer *encoder* is defined simply as the repeated application of single-layer encoders (with independent parameters), plus two final transformation functions $K(\cdot)$ and $V(\cdot)$ applied to every vector in the output sequence of the final layer. Thus the $L$-layer Transformer encoder is defined by the following recursion (with $1 \le \ell \le L - 1$ and $\boldsymbol{X}^1 = \boldsymbol{X}$).

$$\boldsymbol{X}^{\ell+1} = \mathrm{Enc}(\boldsymbol{X}^\ell; \boldsymbol{\theta}_\ell), \quad \boldsymbol{K} = K(\boldsymbol{X}^L), \quad \boldsymbol{V} = V(\boldsymbol{X}^L). \tag{8}$$

We use $(\boldsymbol{K}, \boldsymbol{V}) = \mathrm{TEnc}_L(\boldsymbol{X})$ to denote an $L$-layer Transformer encoder over the sequence $\boldsymbol{X}$.

A *single-layer decoder* is similar to a single-layer encoder but with additional attention to an external pair of key-value vectors $(\boldsymbol{K}^{\mathbf{e}}, \boldsymbol{V}^{\mathbf{e}})$. The input for the single-layer decoder is a sequence $\boldsymbol{Y} = (\boldsymbol{y}_1, \ldots, \boldsymbol{y}_k)$ plus the external pair $(\boldsymbol{K}^{\mathbf{e}}, \boldsymbol{V}^{\mathbf{e}})$, and the output is a sequence $\boldsymbol{Z} = (\boldsymbol{z}_1, \ldots, \boldsymbol{z}_k)$. When defining a decoder layer we denote by $\boldsymbol{Y}_j$ the sequence $(\boldsymbol{y}_1, \ldots, \boldsymbol{y}_j)$, for $1 \le j \le k$. The layer is also parameterized by four functions $Q(\cdot), K(\cdot), V(\cdot)$ and $O(\cdot)$ and is defined as follows.

$$\boldsymbol{p}_i \quad = \quad \mathrm{Att}(Q(\boldsymbol{y}_i), K(\boldsymbol{Y}_i), V(\boldsymbol{Y}_i)) + \boldsymbol{y}_i \tag{9}$$

$$\boldsymbol{a}_i \quad = \quad \mathrm{Att}(\boldsymbol{p}_i, \boldsymbol{K}^{\mathbf{e}}, \boldsymbol{V}^{\mathbf{e}}) + \boldsymbol{p}_i \tag{10}$$

$$\boldsymbol{z}_i \quad = \quad O(\boldsymbol{a}_i) + \boldsymbol{a}_i \tag{11}$$

Notice that the first (self) attention over $(K(\boldsymbol{Y}_i), V(\boldsymbol{Y}_i))$ considers the subsequence of $\boldsymbol{Y}$ only until index $i$ and is used to generate a query $\boldsymbol{p}_i$ to attend the external pair $(\boldsymbol{K}^{\mathbf{e}}, \boldsymbol{V}^{\mathbf{e}})$. We denote the single-decoder layer by $\mathrm{Dec}((\boldsymbol{K}^{\mathbf{e}}, \boldsymbol{V}^{\mathbf{e}}), \boldsymbol{Y}; \boldsymbol{\theta})$.

The Transformer *decoder* is a repeated application of single-layer decoders, plus a transformation function $F : \mathbb{Q}^d \to \mathbb{Q}^d$ applied to the final vector of the decoded sequence. Thus, the output of the decoder is a single vector $\boldsymbol{z} \in \mathbb{Q}^d$. Formally, the $L$-layer Transformer decoder is defined as

$$\boldsymbol{Y}^{\ell+1} = \mathrm{Dec}((\boldsymbol{K}^{\mathbf{e}}, \boldsymbol{V}^{\mathbf{e}}), \boldsymbol{Y}^\ell; \boldsymbol{\theta}_\ell), \quad \boldsymbol{z} = F(\boldsymbol{y}_k^L) \qquad (1 \le \ell \le L - 1 \text{ and } \boldsymbol{Y}^1 = \boldsymbol{Y}). \tag{12}$$

We use $\boldsymbol{z} = \mathrm{TDec}_L((\boldsymbol{K}^{\mathbf{e}}, \boldsymbol{V}^{\mathbf{e}}), \boldsymbol{Y})$ to denote an $L$-layer Transformer decoder.

**The complete Tansformer**   A *Transformer network* receives an input sequence $\boldsymbol{X}$, a seed vector $\boldsymbol{y}_0$, and a value $r \in \mathbb{N}$. Its output is a sequence $\boldsymbol{Y} = (\boldsymbol{y}_1, \ldots, \boldsymbol{y}_r)$ defined as

$$\boldsymbol{y}_{t+1} \quad = \quad \text{TDec}(\text{TEnc}(\boldsymbol{X}), (\boldsymbol{y}_0, \boldsymbol{y}_1, \ldots, \boldsymbol{y}_t)), \qquad \text{for } 0 \leq t \leq r - 1. \tag{13}$$

We denote the output sequence of the transformer as $\boldsymbol{Y} = (\boldsymbol{y}_1, \boldsymbol{y}_2, \ldots, \boldsymbol{y}_r) = \text{Trans}(\boldsymbol{X}, \boldsymbol{y}_0, r)$.

## 3.1   INVARIANCE UNDER PROPORTIONS

The Transformer, as defined above, is *order-invariant*: two input sequences that are permutations of each other produce exactly the same output. This is a consequence of the following property of the attention function: if $\boldsymbol{K} = (\boldsymbol{k}_1, \ldots, \boldsymbol{k}_n)$, $\boldsymbol{V} = (\boldsymbol{v}_1, \ldots, \boldsymbol{v}_n)$, and $\pi : \{1, \ldots, n\} \to \{1, \ldots, n\}$ is a permutation, then $\text{Att}(\boldsymbol{q}, \boldsymbol{K}, \boldsymbol{V}) = \text{Att}(\boldsymbol{q}, \pi(\boldsymbol{K}), \pi(\boldsymbol{V}))$ for every query $\boldsymbol{q}$. This weakness has motivated the need for including information about the order of the input sequence by other means; in particular, this is often achieved by using the so-called *positional encodings* (Vaswani et al., 2017; Shaw et al., 2018), which we study below.

But before going into positional encodings, a natural question is what languages the Transformer can recognize without them. As a standard yardstick we use the well-studied class of regular languages, i.e., languages recognized by finite automata. Order-invariance implies that not every regular language can be recognized by a Transformer network. As an example, there is no Transformer network that can recognize the regular language $(ab)^*$, as the latter is not order-invariant. A reasonable question then is whether the Transformer can express all regular languages which are order-invariant. It is possible to show that this is not the case by proving that the Transformer actually satisfies a stronger invariance property, which we call *proportion invariance*.

For a string $w \in \Sigma^*$ and a symbol $a \in \Sigma$, we use $\text{prop}(a, w)$ to denote the ratio between the number of times that $a$ appears in $w$ and the length of $w$. Consider now the set $\text{PropInv}(w) = \{u \in \Sigma^* \mid \text{prop}(a, w) = \text{prop}(a, u) \text{ for every } a \in \Sigma\}$.

**Proposition 3.1.** *Let* $\text{Trans}$ *be a Transformer,* $\boldsymbol{s}$ *a seed,* $r \in \mathbb{N}$*, and* $f : \Sigma \to \mathbb{Q}^d$ *an embedding function. Then* $\text{Trans}(f(w), \boldsymbol{s}, r) = \text{Trans}(f(u), \boldsymbol{s}, r)$*, for each* $u, w \in \Sigma^*$ *with* $u \in \text{PropInv}(w)$*.*

As an immediate corollary we obtain the following.

**Corollary 3.2.** *Consider the order-invariant regular language* $L = \{w \in \{a, b\}^* \mid w \text{ has an even number of } a \text{ symbols}\}$*. Then* $L$ *cannot be recognized by a Transformer network.*

On the other hand, languages recognized by Transformer networks are not necessarily regular.

**Proposition 3.3.** *There is a Transformer network that recognizes the non-regular language* $S = \{w \in \{a, b\}^* \mid w \text{ has strictly more symbols } a \text{ than symbols } b\}$*.*

That is, the computational power of Transformer networks without positional encoding is both rather weak (they do not even contain order-invariant regular languages) and not so easy to capture (as they can express counting properties that go beyond regularity). As we show in the next section, the inclusion of positional encodings radically changes the picture.

## 3.2   POSITIONAL ENCODINGS AND COMPLETENESS OF THE TRANSFORMER

Positional encodings come to remedy the order-invariance issue by providing information about the absolute positions of the symbols in the input. A positional encoding is just a function $\text{pos} : \mathbb{N} \to \mathbb{Q}^d$. Function $\text{pos}$ combined with an embedding function $f : \Sigma \to \mathbb{Q}^d$ give rise to a new embedding function $f_{\text{pos}} : \Sigma \times \mathbb{N} \to \mathbb{Q}^d$ such that $f_{\text{pos}}(a, i) = f(a) + \text{pos}(i)$. Thus, given an input string $w = a_1 a_2 \cdots a_n \in \Sigma$, the result of the embedding function $f_{\text{pos}}(w)$ provides a "new" input

$$\big(f_{\text{pos}}(a_1, 1), f_{\text{pos}}(a_2, 2), \ldots, f_{\text{pos}}(a_n, n)\big)$$

to the Transformer encoder. Similarly, the Transformer decoder instead of receiving the sequence $\boldsymbol{Y} = (\boldsymbol{y}_0, \boldsymbol{y}_1, \ldots, \boldsymbol{y}_t)$ as input, it receives now the sequence

$$\boldsymbol{Y}' \quad = \quad \big(\boldsymbol{y}_0 + \text{pos}(1), \boldsymbol{y}_1 + \text{pos}(2), \ldots, \boldsymbol{y}_t + \text{pos}(t + 1)\big)$$

As for the case of the embedding functions, we require the positional encoding $\text{pos}(i)$ to be computable by a Turing machine working in linear time w.r.t. the size (in bits) of $i$.

The main result of this section is the completeness of Transformers with positional encodings.

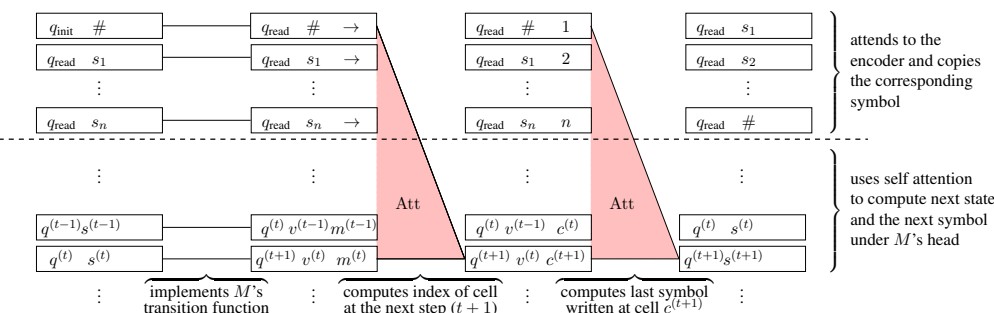

Figure 1: High-level structure of the decoder part of $\text{Trans}_M$.

**Theorem 3.4.** *The class of Transformer networks with positional encodings is Turing complete.*

*Proof Sketch.* We show that for every Turing machine $M = (Q, \Sigma, \delta, q_{\text{init}}, F)$ there exists a transformer that simulates the complete execution of $M$. We represent a string $w = s_1 s_2 \cdots s_n \in \Sigma^*$ as a sequence $\boldsymbol{X}$ of one-hot vectors with their corresponding positional encodings. Denote by $q^{(t)} \in Q$ the state of $M$ at time $t$ when processing $w$, and $s^{(t)} \in \Sigma$ the symbol under $M$'s head at time $t$. Similarly, $v^{(t)} \in \Sigma$ is the symbol written by $M$ and $m^{(t)} \in \{\leftarrow, \rightarrow\}$ the head direction. We next describe how to construct a transformer $\text{Trans}_M$ that with input $\boldsymbol{X}$ produces a sequence $\boldsymbol{y}_0, \boldsymbol{y}_1, \boldsymbol{y}_2, \ldots$ such that $\boldsymbol{y}_i$ contains information about $q^{(i)}$ and $s^{(i)}$ (encoded as one-hot vectors).

The construction and proof goes by induction. Assume the decoder receives $\boldsymbol{y}_0, \ldots, \boldsymbol{y}_t$ such that $\boldsymbol{y}_i$ contains $q^{(i)}$ and $s^{(i)}$. To construct $\boldsymbol{y}_{t+1}$, in the first layer we just implement $M$'s transition function $\delta$; note that $\delta(q^{(i)}, s^{(i)}) = (q^{(i+1)}, v^{(i)}, m^{(i)})$ thus, we use $(q^{(i)}, s^{(i)})$ to compute $(q^{(i+1)}, v^{(i)}, m^{(i)})$ for every $i$ and store them in the sequence $\boldsymbol{z}_0^1, \ldots, \boldsymbol{z}_t^1$. This computation can be done with a two-layer feed-forward network. For the next layer, lets denote by $c^{(i)}$ the index of the cell that $M$ is pointing to at time $i$. It can be proved that given $\boldsymbol{z}_0^1, \ldots, \boldsymbol{z}_t^1$ one can compute (a representation of) $c^{(i)}$ and $c^{(i+1)}$ for every $i \leq t$ with a self-attention layer, and store them in $\boldsymbol{z}_0^2, \ldots, \boldsymbol{z}_t^2$. In particular, $\boldsymbol{z}_t^2$ contains $c^{(t+1)}$ which is the index to which $M$ is going to be pointing to in the next time step. By using the residual connections we also store $q^{(i+1)}$ and $v^{(i)}$ in $\boldsymbol{z}_i^2$. The final piece of our construction is to compute the symbol that the tape holds at index $c^{(t+1)}$, that is, the symbol under $M$'s head at time $t + 1$. For this we use the following observation: the symbol at index $c^{(t+1)}$ in time $t + 1$ coincides with the last symbol written by $M$ at index $c^{(t+1)}$. Thus, we need to find the maximum value $i^\star \leq t$ such that $c^{(i^\star)} = c^{(t+1)}$ and then copy $v^{(i^\star)}$ which is the symbol that was written by $M$ at time step $i^\star$. This last computation can also be done with a self-attention layer. Thus, we attend directly to position $i^\star$ (hard attention plus positional encodings) and copy $v^{(i^\star)}$ which is exactly $s^{(t+1)}$. We finally copy $q^{(t+1)}$ and $s^{(t+1)}$ into the output to construct $\boldsymbol{y}_{t+1}$. Figure 1 shows a high-level diagram of the decoder computation.

There are several other details in the construction, in particular, at the beginning of the computation (first $n$ steps), the decoder needs to attend to the encoder and copy the input symbols so they can later be processed as described above. Another detail is when $M$ reaches a cell that has not been visited before, then the symbol under the head has to be set as $\#$ (the blank symbol). We show that all these decisions can be implemented with feed-forward networks plus attention. The complete construction uses one encoder layer, three decoder layers and vectors of dimension $d = 2|Q| + 4|\Sigma| + 11$ to store one-hot representations of states, symbols and some additional working space. All details can be found in the appendix. $\qquad\square$

## 3.3 Differences with Vaswani et al. (2017)'s framework

Although the general architecture that we presented closely follows that of Vaswani et al. (2017), some choices for functions and parameters in our positive results are different to the usual choices in practice. For instance, we use hard attention which allow us to attend directly to specific positions. In contrast, Vaswani et al. (2017) use $\text{softmax}$ to attend, plus $\text{sin-cos}$ functions as positional encodings. The $\text{softmax}$, $\text{sin}$ and $\text{cos}$ are not rational functions, and thus, are forbidden in our formalization.

An interesting line for future work is to consider arbitrary functions but with additional restrictions, such as finite precision as done by Weiss et al. (2018). Another difference is that for the function $O(\cdot)$ in Equation (11) our proof uses a feed-forward network with various layers, while in Vaswani et al. (2017) only two layers are used.

**The need of arbitrary precision** Our Turing-complete proof relies on having arbitrary precision for internal representations, in particular, for storing and manipulating positional encodings. Although having arbitrary precision is a standard assumption when studying the expressive power of neural networks (Cybenko (1989); Siegelmann & Sontag (1995)) practical implementations rely on fixed precision hardware. If fixed precision is used, then positional encodings can be seen as functions of the form $pos : \mathbb{N} \to A$ where $A$ is a finite subset of $\mathbb{Q}^d$. Thus, the embedding function $f_{pos}$ can be seen as a regular embedding function $f' : \Sigma' \to \mathbb{Q}^d$ where $\Sigma' = \Sigma \times A$. Thus, whenever fixed precision is used, the net effect of having positional encodings is to just increase the size of the input alphabet. Then from Proposition 3.1 we obtain that the Transformer with positional encodings and fixed precision is not Turing complete. Although no longer Turing complete, one can still study the computational power of fixed-precision Transformers. We left this as future work.

## 4 NEURAL GPUS

The Neural GPU (Kaiser & Sutskever, 2016) is an architecture that mixes convolutions and gated recurrences over tridimensional tensors. It is parameterized by three functions $U(\cdot)$ (update function), $R(\cdot)$ (reset function), and $F(\cdot)$. Given a tensor $\mathbf{S} \in \mathbb{Q}^{h \times w \times d}$ and a value $r \in \mathbb{N}$, it produces a sequence $\mathbf{S}^1, \mathbf{S}^2, \ldots, \mathbf{S}^r$ given by the following recursive definition (with $\mathbf{S}^0 = \mathbf{S}$).

$$\mathbf{U}^t = U(\mathbf{S}^{t-1}), \qquad \mathbf{R}^t = R(\mathbf{S}^{t-1}), \qquad \mathbf{S}^t = \mathbf{U}^t \odot \mathbf{S}^{t-1} + (\mathbf{1} - \mathbf{U}) \odot F(\mathbf{R}^t \odot \mathbf{S}^{t-1}).$$

where $\odot$ denotes the element-wise product, and $\mathbf{1}$ is a tensor with only 1's. Neural GPUs force functions $U(\cdot)$ and $R(\cdot)$ to produce a tensor of the same shape as its input with all values in $[0, 1]$. Thus, a Neural GPU resembles a gated recurrent unit (Cho et al., 2014), with $\mathbf{U}$ working as the update gate and $\mathbf{R}$ as the reset gate. Functions $U(\cdot)$, $R(\cdot)$, and $F(\cdot)$ are defined as a convolution of its input with a *4-dimensional kernel bank* with shape $(k_H, k_W, d, d)$ plus a bias tensor, followed by a point-wise transformation

$$f(\mathbf{K} * \mathbf{S} + \mathbf{B}) \tag{14}$$

with different kernels and biases for $U(\cdot)$, $R(\cdot)$, and $F(\cdot)$.

To have an intuition on how the convolution $\mathbf{K} * \mathbf{S}$ works, it is illustrative to think of $\mathbf{S}$ as an $(h \times w)$-grid of (row) vectors and $\mathbf{K}$ as a $(k_H \times k_W)$-grid of $(d \times d)$ matrices. More specifically, let $\boldsymbol{s}_{ij} = \mathbf{S}_{i,j,:}$, and $\boldsymbol{K}_{ij} = \mathbf{K}_{i,j,:,:}$, then $\mathbf{K} * \mathbf{S}$ is a regular two-dimensional convolution in which scalar multiplication has been replaced by vector-matrix multiplication as in the following expression

$$(\mathbf{K} * \mathbf{S})_{i,j,:} = \sum_u \sum_v \boldsymbol{s}_{i+\Delta_1(u),j+\Delta_2(v)} \boldsymbol{K}_{uv}, \tag{15}$$

where $\Delta_1(u) = u - \lfloor k_H/2 \rfloor - 1$ and $\Delta_2(v) = v - \lfloor k_W/2 \rfloor - 1$. This intuition makes evident the similarity between Neural GPUs and cellular automata: $\mathbf{S}$ is a grid of cells, and in every iteration each cell is updated considering the values of its neighbors according to a fixed rule given by $\mathbf{K}$ (Kaiser & Sutskever, 2016). As customary, we assume zero-padding when convolving outside $\mathbf{S}$.

### 4.1 THE COMPUTATIONAL POWER OF NEURAL GPUS

To study the computational power of Neural GPUs, we cast them as a standard seq-to-seq architecture. Given an input sequence, we put every vector in the first column of the tensor $\mathbf{S}$. We also need to pick a special cell of $\mathbf{S}$ as the *output cell* from which we read the output vector in every iteration. We pick the last cell of the first column of $\mathbf{S}$. Formally, given a sequence $\boldsymbol{X} = (\boldsymbol{x}_1, \ldots, \boldsymbol{x}_n)$ with $\boldsymbol{x}_i \in \mathbb{Q}^d$, and a fixed value $w \in \mathbb{N}$, we construct the tensor $\mathbf{S} \in \mathbb{Q}^{n \times w \times d}$ by leting $\mathbf{S}_{i,1,:} = \boldsymbol{x}_i$ and $\mathbf{S}_{i,j,:} = \mathbf{0}$ for $j > 1$. The output of the Neural GPU, denoted by $\text{NGPU}(\boldsymbol{X}, r)$, is the sequence of vectors $\boldsymbol{Y} = (\boldsymbol{y}_1, \boldsymbol{y}_2, \ldots, \boldsymbol{y}_r)$ such that $\boldsymbol{y}_t = \mathbf{S}^t_{n,1,:}$. Given this definition, we can naturally view the Neural GPUs as language recognizers (as formalized in Section 2).

Since the bias tensor **B** in Equation (14) is of the same size than **S**, the number of parameters in a Neural GPU grows with the size of the input. Thus, a Neural GPU cannot be considered as a fixed architecture. To tackle this issue we introduce the notion of *uniform Neural GPU*, as one in which for every bias **B** there exists a matrix $B \in \mathbb{Q}^{w \times d}$ such that $\mathbf{B}_{i,:,:} = B$ for each $i$. Thus, uniform Neural GPUs can be finitely specified (as they have a constant number of parameters, not depending on the length of the input). We now establish the Turing completeness of this model.

**Theorem 4.1.** *The class of uniform Neural GPUs is Turing complete.*

*Proof sketch.* The proof is based on simulating a seq-to-seq RNN; thus, completeness follows from Theorem 2.3. Consider an RNN encoder-decoder language recognizer, such that $N$ is of dimension $d$ and its encoder and decoder are defined by the equations $\boldsymbol{h}_i = \sigma(\boldsymbol{x}_i \boldsymbol{W} + \boldsymbol{h}_{i-1} \boldsymbol{V})$ and $\boldsymbol{g}_t = \sigma(\boldsymbol{g}_{t-1} \boldsymbol{U})$, respectively, where $\boldsymbol{g}_0 = \boldsymbol{h}_n$ and $n$ is the length of the input. We use a Neural GPU with input tensor $\mathbf{S} \in \mathbb{Q}^{n \times 1 \times 3d+3}$. Let $\mathbf{E}_i = \mathbf{S}_{i,1,1:d}$ and $\mathbf{D}_i = \mathbf{S}_{i,1,d+1:2d}$. The idea is to use **E** for the encoder and **D** for the decoder. We use kernel banks of shape $(2, 1, 3d + 3, 3d + 3)$ with uniform bias tensors to simulate the following computation. In every step $t$, we first compute the value of $\sigma(\mathbf{E}_t \boldsymbol{W} + \mathbf{E}_{t-1} \boldsymbol{V})$ and store it in $\mathbf{E}_t$, and then reset $\mathbf{E}_{t-1}$ to zero. Similarly, in step $t$ we update the vector in position $\mathbf{D}_{t-1}$ storing in it the value $\sigma(\mathbf{D}_{t-1} \boldsymbol{U} + \mathbf{E}_{t-1} \boldsymbol{U})$ (for the value of $\mathbf{E}_{t-1}$ before the reset). We use the gating mechanism to ensure a sequential update of the cells such that at time $t$ we update only positions $\mathbf{E}_i$ and $\mathbf{D}_j$ for $i \leq t$ and $j \leq t - 1$. Thus the updates on the **D** are always one iteration behind the update of **E**. Since the vectors in **D** are never reset to zero, they keep being updated which allows us to simulate an arbitrary long computation. In particular we prove that at iteration $t$ it holds that $\mathbf{E}_t = \boldsymbol{h}_t$, and at iteration $n + t$ it holds that $\mathbf{D}_n = \boldsymbol{g}_t$. We require $3d + 3$ components, as we need to implement several gadgets for properly using the update and reset gates. In particular, we need to store the value of $\mathbf{E}_{t-1}$ before we reset it. The detailed construction and the correctness proof can be found in the appendix. $\quad\square$

The proof above makes use of kernels of shape $(2, 1, d, d)$ to obtain Turing completeness. This is, in a sense, optimal, as one can easily prove that Neural GPUs with kernels of shape $(1, 1, d, d)$ are not Turing complete, regardless of the size of $d$. In fact, for kernels of this shape the value of a cell of **S** at time $t$ depends only on the value of the same cell in time $t - 1$.

**Zero padding vs circular convolution** The proof of Theorem 4.1 requires the application of *zero padding* in convolution. This allows us to clearly differentiate internal cells from cells corresponding to the endpoints of the input sequence. Interestingly, Turing-completeness is lost if we replace zero padding with *circular convolution*. Formally, given $\mathbf{S} \in \mathbb{Q}^{h \times w \times d}$, a circular convolution is obtained by defining $\mathbf{S}_{h+n,:,:} = \mathbf{S}_{n,:,:}$ for $n \in \mathbb{Z}$. One can prove that uniform Neural GPUs with circular convolutions cannot differentiate among periodic sequences of different length; in particular, they cannot check if a periodic input sequence is of even or odd length. This yields the following:

**Proposition 4.2.** *Uniform Neural GPUs with circular convolutions are not Turing complete.*

Related to this last result is the empirical observation by Price et al. (2016) that Neural GPUs that learn to solve *hard* problems, e.g., binary multiplication, and which generalize to most of the inputs, struggle with highly symmetric (and nearly periodic) inputs. Actually, Price et al. (2016) exhibit examples of the form $11111111 \times 11111111$ failing for all inputs with eight or more 1s. We leave as future work to explore the implications of our theoretical results on this practical observation.

**Bidimensional tensors and piecewise linear activations** Freivalds & Liepins (2018) simplified Neural GPUs and proved that, by considering piecewise linear activations and bidimensional input tensors instead of the original smooth activations and tridimensional tensors used by Kaiser & Sutskever (2016), it is possible to achieve substantially better results in terms of training time and generalization. Our Turing completeness proof also relies on a bidimensional tensor and uses piecewise linear activations, thus providing theoretical evidence that these simplifications actually retain the full expressiveness of Neural GPUs while simplifying its practical applicability.

## 5 FINAL REMARKS AND FUTURE WORK

We have presented an analysis of the Turing completeness of two popular neural architectures for sequence-processing tasks; namely, the Transformer, based on attention, and the Neural GPU, based on recurrent convolutions. We plan to further refine this analysis in the future. For example, our proof of Turing completeness for the Transformer requires the presence of *residual connections*, i.e., the $+\boldsymbol{x}_i$, $+\boldsymbol{a}_i$, $+\boldsymbol{y}_i$, and $+\boldsymbol{p}_i$ summands in Equations (6-11), while our proof for Neural GPUs heavily relies on the gating mechanism. We will study whether these features are actually essential to obtain completeness.

We presented general abstract versions of both architectures in order to prove our theoretical results. Although we closely follow their original definitions, some choices for functions and parameters in our positive results are different to the usual choices in practice, most notably, the use of hard attention for the case of the Transformer, and the piecewise linear activation functions for both architectures. As we have mentioned, Freivalds & Liepins (2018) showed that for Neural GPUs piecewise linear activations actually help in practice, but for the case of the Transformer architecture more experimentation is needed to have a conclusive response. This is part of our future work.

Although our results are mostly of theoretical interest, they might lead to observations of practical interest. For example, Chen et al. (2018) have established the undecidability of several practical problems related to probabilistic language modeling with RNNs. This means that such problems can only be approached in practice via heuristics solutions. Many of the results in Chen et al. (2018) are, in fact, a consequence of the Turing completeness of RNNs as established by Siegelmann & Sontag (1995). We plan to study to what extent our analogous undecidability results for Transformers and Neural GPUs imply undecidability for language modeling problems based on these architectures.

Finally, our results rely on being able to compute internal representations of arbitrary precision. It would be interesting to perform a theoretical study of the main properties of both architectures in a setting in which only finite precision is allowed, as have been recently carried out for RNNs (Weiss et al., 2018). We also plan to tackle this problem in our future work.

ACKNOWLEDGEMENTS

This work was supported by the Millennium Institute for Foundational Research on Data (IMFD).

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

# A  PROOFS FOR SECTION 2

## A.1  PROOF OF THEOREM 2.3

We first sketch the main idea of Siegelmann & Sontag's proof. We refer the reader to the original paper for details. Siegelmann & Sontag show how to simulate a two-stack machine $M$ (and subsequently, a Turing machine) with a single RNN $N$ with $\sigma$ as activation. They first construct a network $N_1$ that, with $\mathbf{0}$ as initial state ($\boldsymbol{h}_0^{N_1} = \mathbf{0}$) and with a binary string $w \in \{0,1\}^*$ as input sequence, produces a representation of $w$ as a rational number and stores it as one of its internal values. Their internal representation of strings encodes every $w$ as a rational number between $0$ and $1$. In particular, they use base $4$ such that, for example, a string $w = 100110$ is encoded as $(0.311331)_4$ that is, its encoding is

$$3 \times 4^{-1} + 1 \times 4^{-2} + 1 \times 4^{-3} + 3 \times 4^{-4} + 3 \times 4^{-5} + 1 \times 4^{-6}.$$

This representation allows one to easily simulate stack operations as affine transformations plus $\sigma$ activations. For instance, if $x_w$ is the value representing string $w = b_1 b_2 \cdots b_n$ seen as a stack, then the $\mathrm{top}(w)$ operation can be defined as simply $y = \sigma(4x_w - 2)$, since $y = 1$ if and only if $b_1 = 1$, and $y = 0$ if and only if $b_1 = 0$. Other stack operations can de similarly simulated. Using this representation, they construct a second network $N_2$ that simulates the two-stacks machine by using one neuron value to simulate each stack. The input $w$ for the simulated machine $M$ is assumed to be at an internal value given to $N_2$ as an initial state ($\boldsymbol{h}_0^{N_2}$). Thus, $N_2$ expects only zeros as input. Actually, to make $N_2$ work for $r$ steps, an input of the form $0^r$ should be provided.

Finally, they combine $N_1$ and $N_2$ to construct a network $N$ which expects an input of the following form: $(b_1, 1, 0)(b_2, 1, 0) \cdots (b_n, 1, 0)(0, 0, 1)(0, 0, 0)(0, 0, 0) \cdots (0, 0, 0)$. The idea is that the first component contains the input string $w = b_1 b_2 \cdots b_n$, the second component states when the input is *active*, and the third component is $1$ only when the input is inactive for the first time. Before the input vector $(0, 0, 1)$ the network $N_1$ is working. The input $(0, 0, 1)$ is used to simulate a change from $N_1$ to $N_2$, and the rest of input vectors $(0, 0, 0)$ are provided to continue with $N_2$ for as many steps as needed. The number neurons that this construction needs to simulate a machine $M$ with $m$ states, is $10m + 30$. [1]

It is clear that Siegelmann & Sontag's proof resembles a modern encoder-decoder RNN architecture, where $N_1$ is the encoder and $N_2$ is the decoder, thus it is straightforward to use the same construction to provide an RNN encoder-decoder $N'$ and a language recognizer $A$ that uses $N'$ and simulates the two-stacks machine $M$. There are some details that is important to notice. Assume that $N'$ is given by the formulas in Equations (2) and (3). First, since $N_2$ in the above construction expects no input, we can safely assume that $\boldsymbol{R}$ in Equation (3) is the null matrix. Moreover, since $A$ defines its own embedding function, we can ensure that every vector that we provide for the encoder part of $N'$ has a $1$ in a fixed component, and thus we do not need the bias $\boldsymbol{b}_1$ in Equation (2) since it can be simulated with one row of matrix $\boldsymbol{V}$. We can do a similar construction for the bias $\boldsymbol{b}_2$ (Equation (3)). Finally, Siegelmann & Sontag show that its construction can be modified such that a particular neuron of $N_2$, say $n^\star$, is always $0$ except for the first time an accepting state of $M$ is reached, in which case $n^\star = 1$. Thus, one can consider $O(\cdot)$ (Equation (3)) as the identity function and add to $A$ the stopping criterion that just checks if $n^\star$ is $1$. This completes the proof sketch of Theorem 2.3.

---

[1]The idea presented above allows one to linearly simulate $M$, that is, each step of $M$ is simulated with a constant number of steps of the corresponding RNN. Siegelmann & Sontag show that, with a refinement of the above encoding one can simulate $M$ in *real-time*, that is, a single step of $M$ is simulated with a single step of the recurrent network. The $10m + 30$ is the bound given by a simulation with slow-down of two. See the original paper for details (Siegelmann & Sontag, 1995).

# B  PROOFS FOR SECTION 3

## B.1  PROOF OF PROPOSITION 3.1

We extend the definition of the function PropInv to sequences of vectors. Given a sequence $\boldsymbol{X} = (\boldsymbol{x}_1, \ldots, \boldsymbol{x}_n)$ we use vals$(\boldsymbol{X})$ to denote the set of all vectors occurring in $\boldsymbol{X}$. Similarly as for strings, we use prop$(\boldsymbol{v}, \boldsymbol{X})$ as the number of times that $\boldsymbol{v}$ occurs in $\boldsymbol{X}$ divided by the length of $\boldsymbol{X}$. Now we are ready to extend PropInv with the following definition:

$$\text{PropInv}(\boldsymbol{X}) = \{\boldsymbol{X}' \mid \text{vals}(\boldsymbol{X}') = \text{vals}(\boldsymbol{X}) \text{ and } \text{prop}(\boldsymbol{v}, \boldsymbol{X}) = \text{prop}(\boldsymbol{v}, \boldsymbol{X}') \text{ for all } \boldsymbol{v} \in \text{vals}(\boldsymbol{X})\}$$

Notice that for every embedding function $f : \Sigma \to \mathbb{Q}^d$ and string $w \in \Sigma^*$, we have that if $u \in \text{PropInv}(w)$ then $f(u) \in \text{PropInv}(f(w))$. Thus in order to prove that $\text{Trans}(f(w), \boldsymbol{s}, r) = \text{Trans}(f(u), \boldsymbol{s}, r)$ for every $u \in \text{PropInv}(w)$, it is enough to prove that

$$\text{Trans}(\boldsymbol{X}, \boldsymbol{s}, r) = \text{Trans}(\boldsymbol{X}', \boldsymbol{s}, r) \text{ for every } \boldsymbol{X}' \in \text{PropInv}(\boldsymbol{X}) \tag{16}$$

To further simplify the exposition of the proof we introduce another notation. We denote by $p_{\boldsymbol{v}}^{\boldsymbol{X}}$ as the number of times that vector $\boldsymbol{v}$ occurs in $\boldsymbol{X}$. Thus we have that $\boldsymbol{X}' \in \text{PropInv}(\boldsymbol{X})$ if and only if, there exists a value $\gamma \in \mathbb{Q}^+$ such that for every $\boldsymbol{v} \in \text{vals}(\boldsymbol{X})$ it holds that $p_{\boldsymbol{v}}^{\boldsymbol{X}'} = \gamma p_{\boldsymbol{v}}^{\boldsymbol{X}}$.

We now have all the necessary to proceed with the proof of Proposition 3.1. We will prove it by proving the property in (16). Let $\boldsymbol{X} = (\boldsymbol{x}_1, \ldots, \boldsymbol{x}_n)$ be an arbitrary sequence of vectors, and let $\boldsymbol{X}' = (\boldsymbol{x}'_1, \ldots, \boldsymbol{x}'_m) \in \text{PropInv}(\boldsymbol{X})$. Moreover, let $\boldsymbol{Z} = (\boldsymbol{z}_1, \ldots, \boldsymbol{z}_n) = \text{Enc}(\boldsymbol{X}; \boldsymbol{\theta})$ and $\boldsymbol{Z}' = (\boldsymbol{z}'_1, \ldots, \boldsymbol{z}'_m) = \text{Enc}(\boldsymbol{X}'; \boldsymbol{\theta})$. We first prove the following property:

$$\text{For every pair of indices } (i, j) \in \{1, \ldots, n\} \times \{1, \ldots, m\}, \text{ if } \boldsymbol{x}_i = \boldsymbol{x}'_j \text{ then } \boldsymbol{z}_i = \boldsymbol{z}'_j. \tag{17}$$

Lets $(i, j)$ be a pair of indices such that $\boldsymbol{x}_i = \boldsymbol{x}'_j$. From Equations (6-7) we have that $\boldsymbol{z}_i = O(\boldsymbol{a}_i) + \boldsymbol{a}_i$ where $\boldsymbol{a}_i = \text{Att}(Q(\boldsymbol{x}_i), K(\boldsymbol{X}), V(\boldsymbol{X})) + \boldsymbol{x}_i$. Thus, since $\boldsymbol{x}_i = \boldsymbol{x}'_j$, in order to prove $\boldsymbol{z}_i = \boldsymbol{z}'_j$ it is enough to prove that $\text{Att}(Q(\boldsymbol{x}_i), K(\boldsymbol{X}), V(\boldsymbol{X})) = \text{Att}(Q(\boldsymbol{x}'_j), K(\boldsymbol{X}'), V(\boldsymbol{X}'))$. By equations (4-5) and the restriction over the form of normalization functions we have that

$$\text{Att}(Q(\boldsymbol{x}_i), K(\boldsymbol{X}), V(\boldsymbol{X})) = \frac{1}{\alpha} \sum_{\ell=1}^{n} f_\rho(\text{score}(Q(\boldsymbol{x}_i), K(\boldsymbol{x}_\ell))) V(\boldsymbol{x}_\ell)$$

where $\alpha = \sum_{\ell=1}^{n} f_\rho(\text{score}(Q(\boldsymbol{x}_\ell), K(\boldsymbol{x}_\ell)))$. The above equation can be rewritten as

$$\text{Att}(Q(\boldsymbol{x}_i), K(\boldsymbol{X}), V(\boldsymbol{X})) = \frac{1}{\alpha} \sum_{\boldsymbol{v} \in \text{vals}(\boldsymbol{X})} p_{\boldsymbol{v}}^{\boldsymbol{X}} f_\rho(\text{score}(Q(\boldsymbol{x}_i), K(\boldsymbol{v}))) V(\boldsymbol{v})$$

with $\alpha = \sum_{\boldsymbol{v} \in \text{vals}(\boldsymbol{X})} p_{\boldsymbol{v}}^{\boldsymbol{X}} f_\rho(\text{score}(Q(\boldsymbol{v}), K(\boldsymbol{v})))$. By a similar reasoning we can write

$$\text{Att}(Q(\boldsymbol{x}'_j), K(\boldsymbol{X}'), V(\boldsymbol{X}')) = \frac{1}{\beta} \sum_{\boldsymbol{v} \in \text{vals}(\boldsymbol{X}')} p_{\boldsymbol{v}}^{\boldsymbol{X}'} f_\rho(\text{score}(Q(\boldsymbol{x}'_j), K(\boldsymbol{v}))) V(\boldsymbol{v})$$

with $\beta = \sum_{\boldsymbol{v} \in \text{vals}(\boldsymbol{X}')} p_{\boldsymbol{v}}^{\boldsymbol{X}'} f_\rho(\text{score}(Q(\boldsymbol{v}), K(\boldsymbol{v})))$. Now, since $\boldsymbol{X}' \in \text{PropInv}(\boldsymbol{X})$ we know that vals$(\boldsymbol{X}) = \text{vals}(\boldsymbol{X}')$ and there exists a $\gamma \in \mathbb{Q}^+$ such that $p_{\boldsymbol{v}}^{\boldsymbol{X}'} = \gamma p_{\boldsymbol{v}}^{\boldsymbol{X}}$ for every $\boldsymbol{v} \in \text{vals}(\boldsymbol{X})$. Finally, from this last property, plus the fact that $\boldsymbol{x}_i = \boldsymbol{x}'_j$ we have

$$
\begin{aligned}
\text{Att}(Q(\boldsymbol{x}'_j), K(\boldsymbol{X}'), V(\boldsymbol{X}')) &= \frac{1}{\gamma \alpha} \sum_{\boldsymbol{v} \in \text{vals}(\boldsymbol{X})} \gamma p_{\boldsymbol{v}}^{\boldsymbol{X}} f_\rho(\text{score}(Q(\boldsymbol{x}'_j), K(\boldsymbol{v}))) V(\boldsymbol{v}) \\
&= \frac{1}{\alpha} \sum_{\boldsymbol{v} \in \text{vals}(\boldsymbol{X})} p_{\boldsymbol{v}}^{\boldsymbol{X}} f_\rho(\text{score}(Q(\boldsymbol{x}_i), K(\boldsymbol{v}))) V(\boldsymbol{v}) \\
&= \text{Att}(Q(\boldsymbol{x}_i), K(\boldsymbol{X}), V(\boldsymbol{X}))
\end{aligned}
$$

Which completes the proof of Property (17) above.

Consider now the complete encoder TEnc. Let $(\boldsymbol{K}, \boldsymbol{V}) = \text{TEnc}(\boldsymbol{X})$ and $(\boldsymbol{K}', \boldsymbol{V}') = \text{TEnc}(\boldsymbol{X}')$, and let $\boldsymbol{q}$ be an arbitrary vector. We will prove now that $\text{Att}(\boldsymbol{q}, \boldsymbol{K}, \boldsymbol{V}) = \text{Att}(\boldsymbol{q}, \boldsymbol{K}', \boldsymbol{V}')$. By

following a similar reasoning as for proving Property (17) (plus induction on the layers of TEnc) we obtain that if $\boldsymbol{x}_i = \boldsymbol{x}'_j$ then $\boldsymbol{k}_i = \boldsymbol{k}'_j$ and $\boldsymbol{v}_i = \boldsymbol{v}'_j$, for every $i \in \{1, \ldots, n\}$ and $j \in \{1, \ldots, m\}$. Thus, there exists a mapping $M_K : \mathrm{vals}(\boldsymbol{X}) \to \mathrm{vals}(\boldsymbol{K})$ such that $M_K(\boldsymbol{x}_i) = \boldsymbol{k}_i$ and $M_K(\boldsymbol{x}'_j) = \boldsymbol{k}'_j$ and similarly a mapping $M_V : \mathrm{vals}(\boldsymbol{X}) \to \mathrm{vals}(\boldsymbol{V})$ such that $M_V(\boldsymbol{x}_i) = \boldsymbol{v}_i$ and $M_V(\boldsymbol{x}'_j) = \boldsymbol{v}'_j$, for every $i \in \{1, \ldots, n\}$ and $j \in \{1, \ldots, m\}$. Lets focus now on $\mathrm{Att}(\boldsymbol{q}, \boldsymbol{K}, \boldsymbol{V})$. We have:

$$\mathrm{Att}(\boldsymbol{q}, \boldsymbol{K}, \boldsymbol{V}) \;\; = \;\; \frac{1}{\alpha} \sum_{i=1}^{n} f_\rho(\mathrm{score}(\boldsymbol{q}, \boldsymbol{k}_i))\boldsymbol{v}_i$$

with $\alpha = \sum_{i=1}^{n} f_\rho(\mathrm{score}(\boldsymbol{q}, \boldsymbol{k}_i))$. Similarly as before, we can rewrite this as

$$\mathrm{Att}(\boldsymbol{q}, \boldsymbol{K}, \boldsymbol{V}) \;\; = \;\; \frac{1}{\alpha} \sum_{i=1}^{n} f_\rho(\mathrm{score}(\boldsymbol{q}, M_K(\boldsymbol{x}_i)))M_V(\boldsymbol{x}_i)$$

$$= \;\; \frac{1}{\alpha} \sum_{\boldsymbol{v} \in \mathrm{vals}(\boldsymbol{X})} p_{\boldsymbol{v}}^{\boldsymbol{X}} f_\rho(\mathrm{score}(\boldsymbol{q}, M_K(\boldsymbol{v})))M_V(\boldsymbol{v})$$

with $\alpha = \sum_{\boldsymbol{v} \in \mathrm{vals}(\boldsymbol{X})} p_{\boldsymbol{v}}^{\boldsymbol{X}} f_\rho(\mathrm{score}(\boldsymbol{q}, M_K(\boldsymbol{v})))$. Similarly for $\mathrm{Att}(\boldsymbol{q}, \boldsymbol{K}', \boldsymbol{V}')$ we have

$$\mathrm{Att}(\boldsymbol{q}, \boldsymbol{K}', \boldsymbol{V}') \;\; = \;\; \frac{1}{\beta} \sum_{j=1}^{m} f_\rho(\mathrm{score}(\boldsymbol{q}, M_K(\boldsymbol{x}'_j)))M_V(\boldsymbol{x}'_j)$$

$$= \;\; \frac{1}{\beta} \sum_{\boldsymbol{v} \in \mathrm{vals}(\boldsymbol{X}')} p_{\boldsymbol{v}}^{\boldsymbol{X}'} f_\rho(\mathrm{score}(\boldsymbol{q}, M_K(\boldsymbol{v})))M_V(\boldsymbol{v})$$

And finally using that $\boldsymbol{X}' \in \mathrm{PropInv}(\boldsymbol{X})$ we obtain

$$\mathrm{Att}(\boldsymbol{q}, \boldsymbol{K}', \boldsymbol{V}') \;\; = \;\; \frac{1}{\beta} \sum_{\boldsymbol{v} \in \mathrm{vals}(\boldsymbol{X}')} p_{\boldsymbol{v}}^{\boldsymbol{X}'} f_\rho(\mathrm{score}(\boldsymbol{q}, M_K(\boldsymbol{v})))M_V(\boldsymbol{v})$$

$$= \;\; \frac{1}{\gamma\alpha} \sum_{\boldsymbol{v} \in \mathrm{vals}(\boldsymbol{X})} \gamma p_{\boldsymbol{v}}^{\boldsymbol{X}} f_\rho(\mathrm{score}(\boldsymbol{q}, M_K(\boldsymbol{v})))M_V(\boldsymbol{v})$$

$$= \;\; \frac{1}{\alpha} \sum_{\boldsymbol{v} \in \mathrm{vals}(\boldsymbol{X})} p_{\boldsymbol{v}}^{\boldsymbol{X}} f_\rho(\mathrm{score}(\boldsymbol{q}, K(\boldsymbol{v}))V(\boldsymbol{v})$$

$$= \;\; \mathrm{Att}(\boldsymbol{q}, \boldsymbol{K}, \boldsymbol{V})$$

which is what we wanted.

To complete the rest proof, consider $\mathrm{Trans}(\boldsymbol{X}, \boldsymbol{y}_0, r)$ which is defined by the recursion

$$\boldsymbol{y}_{k+1} \;\; = \;\; \mathrm{TDec}(\mathrm{TEnc}(\boldsymbol{X}), (\boldsymbol{y}_0, \boldsymbol{y}_1, \ldots, \boldsymbol{y}_k))$$

To prove that $\mathrm{Trans}(\boldsymbol{X}, \boldsymbol{y}_0, r) = \mathrm{Trans}(\boldsymbol{X}', \boldsymbol{y}_0, r)$ we use an inductive argument. We know that

$$\boldsymbol{y}_1 \;\; = \;\; \mathrm{TDec}(\mathrm{TEnc}(\boldsymbol{X}), (\boldsymbol{y}_0))$$
$$= \;\; \mathrm{TDec}((\boldsymbol{K}, \boldsymbol{V}), (\boldsymbol{y}_0)).$$

Now TDec only access $(\boldsymbol{K}, \boldsymbol{V})$ via attentions of the form $\mathrm{Att}(\boldsymbol{q}, \boldsymbol{K}, \boldsymbol{V})$ and for the case of $\boldsymbol{y}_1$ the vector $\boldsymbol{q}$ can only depend on $\boldsymbol{y}_0$, thus, from $\mathrm{Att}(\boldsymbol{q}, \boldsymbol{K}, \boldsymbol{V}) = \mathrm{Att}(\boldsymbol{q}, \boldsymbol{K}', \boldsymbol{V}')$ we have that

$$\boldsymbol{y}_1 \;\; = \;\; \mathrm{TDec}((\boldsymbol{K}, \boldsymbol{V}), (\boldsymbol{y}_0))$$
$$= \;\; \mathrm{TDec}((\boldsymbol{K}', \boldsymbol{V}'), (\boldsymbol{y}_0))$$
$$= \;\; \mathrm{TDec}(\mathrm{TEnc}(\boldsymbol{X}'), (\boldsymbol{y}_0)).$$

The rest of the steps follow by a simple induction on $k$.

## B.2 Proof of Corollary 3.2

To obtain a contradiction, assume that there is a language recognizer $A$ that uses a Transformer network and such that $L = L(A)$. Now consider the strings $w_1 = aabb$ and $w_2 = aaabbb$. Since $w_1 \in \mathrm{PropInv}(w_2)$ by Proposition 3.1 we have that $w_1 \in L(A)$ if and only if $w_2 \in L(A)$ which is a contradiction since $w_1 \in L$ but $w_2 \notin L$. This completes the proof of the corollary.

### B.3  PROOF OF PROPOSITION 3.3

We construct a language recognizer $A = (\Sigma, f, \mathrm{Trans}, \boldsymbol{s}, \mathbb{F})$ with $\mathrm{Trans}$ a very simple Transformer network with dimension $d = 2$ and using just one layer of encoder and one layer of decoder, such that $L(A) = \{w \in \{a, b\}^* \mid w$ has strictly more symbols $a$ than symbols $b\}$. As embedding function, we use $f(a) = [0, 1]$ and $f(b) = [0, -1]$.

Assume that the output for the encoder part of the transformer is $\boldsymbol{X} = (\boldsymbol{x}_1, \ldots, \boldsymbol{x}_n)$. First we use an encoder layer that implements the identity function. This can be trivially done using null functions for the self attention and through the residual connections this encoder layer shall preserve the original $\boldsymbol{x}_i$ values. For the final $V(\cdot)$ and $K(\cdot)$ functions of the Transformer encoder (Equation (8)), we use $V(\boldsymbol{x}) = \boldsymbol{x}$ the identity function and $K(\boldsymbol{x}) = [0, 0]$, giving $\boldsymbol{V^e} = \boldsymbol{X}$ and $\boldsymbol{K^e} = ([0, 0], [0, 0], \ldots, [0, 0])$.

For the decoder we use a similar approach. We consider the identity in the self attention plus the residual (which can be done by just using the null functions for the self attention). Considering the external attention, that is the attention over $(\boldsymbol{K^e}, \boldsymbol{V^e})$, we let score and $\rho$ be arbitrary scoring and normalization functions. And finally for the function $O(\cdot)$ (Equation (11)) we use a single layer neural network implementing the affine transformation $O([x, y]) = [y - x, -y]$ such that $O([x, y]) + [x, y] = [y, 0]$. The final function $F(\cdot)$ is just the identity function.

In order to complete the proof we introduce some notation. Lets denote by $\#_a(w)$ as the number of $a$'s in $w$, and similarly $\#_b(w)$ for the number of $b$'s in $w$. Lets call $c_w$ as the value $\frac{\#_a(w) - \#_b(w)}{n}$. We now prove that, for any string $w \in \{a, b\}^*$ if we consider $f(w) = \boldsymbol{X} = (\boldsymbol{x}_1, \ldots, \boldsymbol{x}_n)$ as the input sequence for $\mathrm{Trans}$ and we use initial value $\boldsymbol{s} = [0, 0]$ for the decoder, the complete network shall compute a sequence $\boldsymbol{y}_1, \boldsymbol{y}_2, \ldots, \boldsymbol{y}_r$ such that:

$$\boldsymbol{y}_i = \begin{cases} [0, 0] & i = 0 \\ [c_w, 0] & i > 0 \end{cases}$$

We proceed by induction. The base case trivially holds since $\boldsymbol{y}_0 = \boldsymbol{s} = [0, 0]$. Assume now that we are at step $r$ and the input for the decoder is $(\boldsymbol{y}_0, \boldsymbol{y}_1, \ldots, \boldsymbol{y}_r)$. We will show that $\boldsymbol{y}_{r+1} = [c_w, 0]$. Since we consider the identity in the self attention (Equation (9)), we have that $\boldsymbol{p}_i = \boldsymbol{y}_i$ for every $i$ in $\{0, \ldots, i\}$. Now considering the external attention, that is the attention over $(\boldsymbol{K^e}, \boldsymbol{V^e})$, Since all key vectors in $\boldsymbol{K^e}$ are $[0, 0]$, the external attention will produce the same score value for all positions. That is, $\mathrm{score}(\boldsymbol{p}_i, \boldsymbol{k}_{j_1}) = \mathrm{score}(\boldsymbol{p}_i, \boldsymbol{k}_{j_2})$ for every $j_1, j_2$. Lets call this value $s^\star$. Thus we have that

$$\begin{aligned} \rho(\mathrm{score}(\boldsymbol{p}_i, \boldsymbol{k}_1), \ldots, \mathrm{score}(\boldsymbol{p}_i, \boldsymbol{k}_n)) &= \rho(s^\star, s^\star, \ldots, s^\star) \\ &= \left( \frac{1}{n}, \frac{1}{n}, \ldots, \frac{1}{n} \right). \end{aligned}$$

Then, since $\boldsymbol{V^e} = \boldsymbol{X}$ we have that

$$\begin{aligned} \mathrm{Att}(\boldsymbol{p}_i, \boldsymbol{K^e}, \boldsymbol{V^e}) &= \frac{1}{n} \sum_{\ell=1}^{n} \boldsymbol{x}_\ell \\ &= \frac{1}{n} [0, \#_a(w) - \#_b(w)] \end{aligned}$$

for every $i \in \{0, \ldots, r\}$. The last equality holds since our embedding are $f(a) = [0, 1]$ and $f(b) = [0, -1]$, and so every $a$ in $w$ sums one and every $b$ subtracts one. Thus, we have that

$$\mathrm{Att}(\boldsymbol{p}_i, \boldsymbol{K^e}, \boldsymbol{V^e}) = [0, c_w].$$

for every $i \in \{0, \ldots, r\}$. In the next step, after the external attention plus the residual connection (Equation (10)) we have

$$\begin{aligned} \boldsymbol{a}_i &= \mathrm{Att}(\boldsymbol{p}_i, \boldsymbol{K^e}, \boldsymbol{V^e}) + \boldsymbol{p}_i \\ &= \mathrm{Att}(\boldsymbol{p}_i, \boldsymbol{K^e}, \boldsymbol{V^e}) + \boldsymbol{y}_i \\ &= [0, c_w] + [c_w, 0] \\ &= [c_w, c_w] \end{aligned}$$

Applying function $O(\cdot)$ plus the residual connection (Equation (11)) we have

$$
\begin{aligned}
\boldsymbol{z}_i &= O(\boldsymbol{a}_i) + \boldsymbol{a}_i \\
&= O([c_w, c_w]) + [c_w, c_w] \\
&= [c_w - c_w, -c_w] + [c_w, c_w] \\
&= [c_w, 0]
\end{aligned}
$$

Finally, $\boldsymbol{y}_{r+1} = F(\boldsymbol{z}_r) = \boldsymbol{z}_r = [c_w, 0]$ which is exactly what we wanted to prove.

To complete the proof, notice that $\#_a(w) > \#_b(w)$ if and only if $c_w > 0$. If we define $\mathbb{F}$ as $\mathbb{Q}^+ \times \mathbb{Q}$, the recognizer $A = (\Sigma, f, \text{Trans}, \boldsymbol{s}, \mathbb{F})$ will accept the string $w$ exactly when $c_w > 0$, that is, $w \in L(A)$ if and only if $\#_a(w) > \#_b(w)$. That is exactly the language $S$, and so the proof is complete.

## B.4 PROOF OF THEOREM 3.4

Let $M = (Q, \Sigma, \delta, q_{\text{init}}, F)$ be a Turing machine with a infinite tape and assume that the special symbol $\# \in \Sigma$ is used to mark blank positions in the tape. We make the following assumptions about how $M$ works when processing an input string:

- $M$ always moves its head either to the left or to the right (it never stays at the same cell).
- $M$ begins at state $q_{\text{init}}$ pointing to the cell immediately to the left of the input string.
- $M$ never makes a transition to the left of the initial position.
- $Q$ has a special state $q_{\text{read}}$ used to read the complete input.
- Initially (time 0), $M$ makes a transition to state $q_{\text{read}}$ and move its head to the right.
- While in state $q_{\text{read}}$ it moves to the right until symbol $\#$ is read.
- There are no transitions going out from accepting states (states in $F$).

It is easy to prove that every general Turing machine is equivalent to one that satisfies the above assumptions. We prove that one can construct a transformer network $\text{Trans}_M$ that is able to simulate $M$ on every possible input string.

The construction is somehow involved and uses several helping values, sequences and intermediate results. To make the reading more easy we divide the construction and proof in three parts. We first give a high-level view of the strategy we use. Then we give some details on the architecture of the encoder and decoder needed to implement our strategy, and finally we formally prove that every part of our architecture can be actually implemented.

### B.4.1 OVERVIEW OF THE CONSTRUCTION AND HIGH-LEVEL STRATEGY

In the encoder part of $\text{Trans}_M$ we receive as input the string $w = s_1 s_2 \ldots s_n$. We first use an embedding function to represent every $s_i$ as a one-hot vector and add a positional encoding for every index. The encoder produces output $(\boldsymbol{K}^e, \boldsymbol{V}^e)$ where $\boldsymbol{K}^e = (\boldsymbol{k}_1^e, \ldots, \boldsymbol{k}_n^e)$ and $\boldsymbol{V}^e = (\boldsymbol{v}_1^e, \ldots, \boldsymbol{v}_n^e)$ are sequences of keys and values such that $\boldsymbol{v}_i^e$ contains the information of $s_i$ and $\boldsymbol{k}_i^e$ contains the information of the $i$-th positional encoding. We later show that this allows us to attend to every specific position and copy every input symbol from the encoder to the decoder (Lemma B.1).

In the decoder part of $\text{Trans}_M$ we simulate a complete execution of $M$ over $w = s_1 s_2 \cdots s_n$. For this we define the following sequences (for $i \geq 0$):

$$
\begin{aligned}
q^{(i)} &: \quad \text{state of } M \text{ at time } i \\
s^{(i)} &: \quad \text{symbol under the head of } M \text{ at time } i \\
v^{(i)} &: \quad \text{symbol written by } M \text{ at time } i \\
m^{(i)} &: \quad \text{head direction in the transition of } M \text{ at time } i
\end{aligned}
$$

For the case of $m^{(i)}$ we assume that $-1$ represents a movement to the left and $1$ represents a movement to the right. In our construction we show how to build a decoder that computes all the above values for every time step $i$ using self attention plus attention over the encoder part. Since the above values contain all the needed information to reconstruct the complete history of the computation, we can effectively simulate $M$.

In particular our construction produces the sequence of output vectors $\boldsymbol{y}_1, \boldsymbol{y}_2, \ldots$ such that, for every $i$, the vector $\boldsymbol{y}_i$ contains information about $q^{(i)}$ and $s^{(i)}$ encoded as one-hot vectors. The construction and proof goes by induction. We begin with an initial vector $\boldsymbol{y}_0$ that represents the state of the computation before it has started, that is $q^{(0)} = q_{\text{init}}$ and $s^{(0)} = \#$. For the induction step we assume that we have already computed $\boldsymbol{y}_1, \ldots, \boldsymbol{y}_r$ such that $\boldsymbol{y}_i$ contains information about $q^{(i)}$ and $s^{(i)}$, and we show how with input $(\boldsymbol{y}_0, \boldsymbol{y}_1, \ldots, \boldsymbol{y}_r)$ the decoder produces the next vector $\boldsymbol{y}_{r+1}$ containing $q^{(r+1)}$ and $s^{(r+1)}$.

The overview of the construction is as follows. First notice that the transition function $\delta$ relates the above values with the following equation:

$$\delta(q^{(i)}, s^{(i)}) = (q^{(i+1)}, v^{(i)}, m^{(i)}). \tag{18}$$

We prove that we can use a two-layer feed-forward network to mimic the transition function $\delta$ (Lemma B.2). Thus, given that the input vector $\boldsymbol{y}_i$ contains $q^{(i)}$ and $s^{(i)}$, we can produce the values $q^{(i+1)}$, $v^{(i)}$ and $m^{(i)}$ (and store them as values in the decoder). In particular, since $\boldsymbol{y}_r$ is in the input, we can produce $q^{(r+1)}$ which is part of what we need for $\boldsymbol{y}_{r+1}$. In order to complete the construction we also need to compute the value $s^{(r+1)}$, that is, we need to compute the symbol under the head of machine $M$ at the next time step (time $r+1$). We next describe at a high level, how this symbol can be computed with two additional decoder layers.

We first make some observations about $s^{(i)}$ that are fundamental in our computation. Assume that at time $i$ the head of $M$ is pointing to the cell at index $k$. Then we have three possibilities:

1. If $i \leq n$, then $s^{(i)} = s_i$ since $M$ is still reading its input string.

2. If $i > n$ and $M$ has never written at index $k$, then $s^{(i)} = \#$, the blank symbol.

3. In other case, that is, if $i > n$ and time $i$ is not the first time that $M$ is pointing to index $k$, then $s^{(i)}$ is *the last symbol written by $M$ at index $k$*.

For the case (1) we can produce $s^{(i)}$ by simply attending to position $i$ in the encoder part. Thus, if $r + 1 \leq n$ to produce $s^{(r+1)}$ we can just attend to index $r + 1$ in the encoder and copy this value to $\boldsymbol{y}_{r+1}$. For cases (2) and (3) the solution is a bit more complicated, but almost all the important work is to compute what is the index that $M$ is going to be pointing to in time $r + 1$.

To formalize this computation, lets denote by $c^{(i)} \in \mathbb{Z}$ the following value:

$$c^{(i)} \quad : \quad \text{the index of the cell to which the head of } M \text{ is pointing to at time } i$$

Notice that value $c^{(i)}$ satisfies that $c^{(i)} = c^{(i-1)} + m^{(i-1)}$. If we unroll this equation and assuming that $c^{(0)} = 0$ we obtain that

$$c^{(i)} = m^{(0)} + m^{(1)} + \cdots + m^{(i-1)}.$$

Then, at the step $i$ in the decoder we have all the necessary to compute value $c^{(i)}$ but also the necessary to compute $c^{(i+1)}$. We actually show that the computation (of a representation) of $c^{(i)}$ and $c^{(i+1)}$ can be done by using one layer of self attention (Lemma B.3).

We still need to define a final notion. With $c^{(i)}$ one can define the helping value $\ell(i)$ as follows:

$$\ell(i) = \max\{j \mid j < i \text{ and } c^{(j)} = c^{(i)}\}.$$

Thus, $\ell(i)$ is a value such that $c^{(\ell(i))} = c^{(i)}$, which means that at time $i$ and at time $\ell(i)$ the head of $M$ was pointing to the same cell. Moreover, $\ell(i)$ is the maximum value less than $i$ that satisfies such condition. That is $\ell(i)$ is *the last time* (previous to $i$) in which $M$ was pointing to position $c^{(i)}$. First notice that in every step, $M$ moves its head either to the right or to the left (it never stays in the same cell). This implies that for every $i$ it holds that $c^{(i)} \neq c^{(i-1)}$, from which we obtain that $\ell(i) < i - 1$. Moreover, in the case that $c^{(i)}$ is visited for the first time at time step $i$, the value $\ell(i)$ is ill-defined. In such a case we let $\ell(i) = i - 1$. This makes $\ell(i) \leq i - 1$ for all $i$, and allows us to check that $c^{(i)}$ is visited for the first time at time step $i$ by just checking that $\ell(i) = i - 1$.

We now have all the necessary to explain how we compute our desired $s^{(r+1)}$ value. Assume that $r + 1 > n$ (the case $r + 1 \leq n$ was already covered before). We first note that if $\ell(r + 1) = r$ then $s^{(r+1)} = \#$ since this is the first time that cell $c^{(r+1)}$ is visited. On the other hand, if $\ell(r + 1) < r$ then $s^{(r+1)}$ is the value written by $M$ at time $\ell(r + 1)$ which is exactly $v^{(\ell(r+1))}$. Thus, in this case we only need to attend to position $\ell(r + 1)$ and copy the value $v^{(\ell(r+1))}$ to produce $s^{(r+1)}$. We show that all this can be done with an additional self-attention decoder layer (Lemma B.4).

We have described at a high-level a decoder that, with input $(\boldsymbol{y}_0, \boldsymbol{y}_1, \ldots, \boldsymbol{y}_r)$, computes the values $q^{(r+1)}$ and $s^{(r+1)}$ which is what we need to produce $\boldsymbol{y}_{r+1}$. We next show all the details of this construction.

### B.4.2 DETAILS OF THE ARCHITECTURE OF $\text{Trans}_M$

In this section we give more details on the architecture of the encoder and decoder needed to implement our strategy. We let several intermediate claims as lemmas that we formally prove in Section B.4.3.

ATTENTION MECHANISM

For our attention mechanism we use the following non-linear function:

$$\varphi(x) \quad = \quad \begin{cases} x & x \leq 0, \\ -x & x > 0. \end{cases} \tag{19}$$

We note that $\varphi(x) = -|x|$ and it can be implemented as $\varphi(x) = -\operatorname{relu}(x) - \operatorname{relu}(-x)$. We use $\varphi(\cdot)$ to define a scoring function $\text{score}_\varphi : \mathbb{R}^d \times \mathbb{R}^d \to \mathbb{R}$ such that

$$\text{score}_\varphi(\boldsymbol{u}, \boldsymbol{v}) = \varphi(\langle \boldsymbol{u}, \boldsymbol{v} \rangle) = -|\langle \boldsymbol{u}, \boldsymbol{v} \rangle|.$$

Now, let $\boldsymbol{q} \in \mathbb{Q}^d$, and $\boldsymbol{K} = (\boldsymbol{k}_1, \ldots, \boldsymbol{k}_n)$ and $\boldsymbol{V} = (\boldsymbol{v}_1, \ldots, \boldsymbol{v}_n)$ be tuples of elements in $\mathbb{Q}^d$. We now describe how $\text{Att}(\boldsymbol{q}, \boldsymbol{K}, \boldsymbol{V})$ is generally computed when hard attention is considered. Assume first that there exists a single $j^\star \in \{1, \ldots, n\}$ that maximizes $\text{score}_\varphi(\boldsymbol{q}, \boldsymbol{k}_j)$. In that case we have that $\text{Att}(\boldsymbol{q}, \boldsymbol{K}, \boldsymbol{V}) = \boldsymbol{v}_{j^\star}$ with

$$\begin{aligned} j^\star \quad &= \quad \underset{1 \leq j \leq n}{\arg\max} \ \text{score}_\varphi(\boldsymbol{q}, \boldsymbol{k}_j) \\ &= \quad \underset{1 \leq j \leq n}{\arg\max} \ -|\langle \boldsymbol{q}, \boldsymbol{k}_j \rangle| \\ &= \quad \underset{1 \leq j \leq n}{\arg\min} \ |\langle \boldsymbol{q}, \boldsymbol{k}_j \rangle| \end{aligned} \tag{20}$$

Thus, when computing hard attention with the function $\text{score}_\varphi(\cdot)$ we essentially select the vector $\boldsymbol{v}_j$ such that the dot product $\langle \boldsymbol{q}, \boldsymbol{k}_j \rangle$ is as close to $0$ as possible. If there is more than one index, say indexes $j_1, j_2, \ldots, j_r$, that minimizes the dot product $\langle \boldsymbol{q}, \boldsymbol{k}_j \rangle$ then we have that

$$\text{Att}(\boldsymbol{q}, \boldsymbol{K}, \boldsymbol{V}) = \frac{1}{r}\big(\boldsymbol{v}_{j_1} + \boldsymbol{v}_{j_2} + \cdots + \boldsymbol{v}_{j_r}\big).$$

Thus, in the extreme case in which all dot products are equal $\langle \boldsymbol{q}, \boldsymbol{k}_j \rangle$ for every index $j$, attention behaves just as an average of all value vectors, that is $\text{Att}(\boldsymbol{q}, \boldsymbol{K}, \boldsymbol{V}) = \frac{1}{n} \sum_{j=1}^{n} \boldsymbol{v}_j$. We use all these properties of the hard attention in our proof.

VECTORS AND ENCODINGS

We now describe the vectors that we use in the encoder and decoder parts of $\text{Trans}_M$. The vectors that we use in the $\text{Trans}_M$ layers are of dimension $d = 2|Q| + 4|\Sigma| + 11$. To simplify the exposition, whenever we use a vector $\boldsymbol{v} \in \mathbb{Q}^d$, we write it arranged in four groups of values as follows

$$\begin{aligned} \boldsymbol{v} \quad = \quad [ \quad &\boldsymbol{q}_1, \boldsymbol{s}_1, x_1, \\ &\boldsymbol{q}_2, \boldsymbol{s}_2, x_2, x_3, x_4, x_5, \\ &\boldsymbol{s}_3, x_6, \boldsymbol{s}_4, x_7 \\ &x_8, x_9, x_{10}, x_{11} \qquad ] \end{aligned}$$

where $\boldsymbol{q}_i \in \mathbb{Q}^{|Q|}$, $\boldsymbol{s}_i \in \mathbb{Q}^{|\Sigma|}$, and $x_i \in \mathbb{Q}$. Whenever in a vector of the above form any of the four groups of values is composed only of $0$'s, we just write '$0, \ldots, 0$' where the length of this sequence is implicit in the length of the corresponding group. Finally, we denote by $\boldsymbol{0}_q$ the vector in $\mathbb{Q}^{|Q|}$ that has only $0$'s, and similarly $\boldsymbol{0}_s$ the vector in $\mathbb{Q}^{|\Sigma|}$ that has only $0$'s.

For a symbol $s \in \Sigma$, we use $[\![ s ]\!]$ to denote a one-hot vector in $\mathbb{Q}^{|\Sigma|}$ that represents $s$. That is, given an enumeration $\pi : \Sigma \to \{1, \ldots, |\Sigma|\}$, the vector $[\![ s ]\!]$ has a $1$ in position $\pi(s)$ and a $0$ in all other positions. Similarly, for $q \in Q$, we use $[\![ q ]\!]$ to denote a one-hot vector in $\mathbb{Q}^{|Q|}$ that represents $q$.

EMBEDDINGS AND POSITIONAL ENCODINGS

We have the necessary to introduce the embedding and positional encoding used in our construction. We use an embedding function $f : \Sigma \to \mathbb{Q}^d$ defined as

$$
f(s) \;=\; [\quad 0, \ldots, 0, \\
0, \ldots, 0, \\
[\![\, s_i \,]\!], 0, \mathbf{0}_s, 0, \\
0, \ldots, 0 \quad]
$$

Our construction uses the positional encoding $\mathrm{pos} : \mathbb{N} \to \mathbb{Q}^d$ such that

$$
\mathrm{pos}(i) \;=\; [\quad 0, \ldots, 0, \\
0, \ldots, 0, \\
0, \ldots, 0, \\
1, i, 1/i, 1/i^2 \quad]
$$

Thus, given an input sequence $s_1 s_2 \cdots s_n \in \Sigma^*$, we have that

$$
f_{\mathrm{pos}}(s_i) = f(s_i) + \mathrm{pos}(i) \;=\; [\quad 0, \ldots, 0, \\
0, \ldots, 0, \\
[\![\, s_i \,]\!], 0, \mathbf{0}_s, 0, \\
1, i, 1/i, 1/i^2 \quad]
$$

We denote this last vector by $\boldsymbol{x}_i$. That is, if $M$ receives the input string $w = s_1 s_2 \cdots s_n$, then the input for $\mathrm{Trans}_M$ is the sequence $(\boldsymbol{x}_1, \boldsymbol{x}_2, \ldots, \boldsymbol{x}_n)$. The need for using a positional encoding having values $1/i$ and $1/i^2$ will be clear when we formally prove the correctness of our construction.

We need a final preliminary notion. In the formal construction of $\mathrm{Trans}_M$ we also use the following helping sequences:

$$
\alpha^{(i)} \;=\; \begin{cases} s_i & 1 \le i \le n \\ s_n & i > n \end{cases}
$$

$$
\beta^{(i)} \;=\; \begin{cases} i & i \le n \\ n & i > n \end{cases}
$$

These are used to identify when $M$ is still reading the input string.

CONSTRUCTION OF $\mathrm{TEnc}_M$

The encoder part of $\mathrm{Trans}_M$ is very simple. For $\mathrm{TEnc}_M$ we use a single-layer encoder, such that $\mathrm{TEnc}_M(\boldsymbol{x}_1, \ldots, \boldsymbol{x}_n) = (\boldsymbol{K^e}, \boldsymbol{V^e})$ where $\boldsymbol{K^e} = (\boldsymbol{k}_1, \ldots, \boldsymbol{k}_n)$ and $\boldsymbol{V^e} = (\boldsymbol{v}_1, \ldots, \boldsymbol{v}_n)$ such that

$$
\boldsymbol{k}_i \;=\; [\quad 0, \ldots, 0, \\
0, \ldots, 0, \\
0, \ldots, 0, \\
i, -1, 0, 0 \quad]
$$

$$
\boldsymbol{v}_i \;=\; [\quad 0, \ldots, 0, \\
0, \ldots, 0, \\
[\![\, s_i \,]\!], i, \mathbf{0}_s, 0, \\
0, \ldots, 0 \quad]
$$

It is straightforward to see that these vectors can be produced with a single encoder layer by using a trivial self attention, taking advantage of the residual connections in Equations (6) and (7), and then using linear transformations for $V(\cdot)$ and $K(\cdot)$ in Equation (8).

When constructing the decoder we use the following property.

**Lemma B.1.** *Let $\boldsymbol{q} \in \mathbb{Q}^d$ be a vector such that $\boldsymbol{q} = [\_, \ldots, \_, 1, j, \_, \_]$ where $j \in \mathbb{N}$ and '$\_$' denotes an arbitrary value. Then we have that*

$$
\mathrm{Att}(\boldsymbol{q}, \boldsymbol{K^e}, \boldsymbol{V^e}) \;=\; [\quad 0, \ldots, 0, \\
0, \ldots, 0, \\
[\![\, \alpha^{(j)} \,]\!], \beta^{(j)}, \mathbf{0}_s, 0, \\
0, \ldots, 0 \quad]
$$

CONSTRUCTION OF $\mathrm{TDec}_M$

We next show how to construct the decoder part of $\mathrm{Trans}_M$ to produce the sequence of outputs $\boldsymbol{y}_1, \boldsymbol{y}_2, \ldots$, where $\boldsymbol{y}_i$ is given by:

$$\boldsymbol{y}_i = [ \quad [\![ q^{(i)} ]\!], [\![ s^{(i)} ]\!], m^{(i-1)},$$
$$0, \ldots, 0,$$
$$0, \ldots, 0,$$
$$0, \ldots, 0 \qquad ]$$

That is, $\boldsymbol{y}_i$ contains information about the state of $M$ at time $i$, the symbol under the head of $M$ at time $i$, and the last direction followed by $M$ (the direction of the head movement at time $i-1$). The need to include $m^{(i-1)}$ will be clear in the construction.

We consider as the starting vector for the decoder the vector

$$\boldsymbol{y}_0 = [ \quad [\![ q_{\text{init}} ]\!], [\![ \# ]\!], 0,$$
$$0, \ldots, 0,$$
$$0, \ldots, 0,$$
$$0, \ldots, 0 \qquad ]$$

We are assuming that $m^{(-1)} = 0$ to represent that previous to time $0$ there was no head movement. Our construction resembles a proof by induction; we describe the architecture piece by piece and at the same time we show how for every $r \geq 0$ our architecture constructs $\boldsymbol{y}_{r+1}$ from the previous vectors $(\boldsymbol{y}_0, \ldots, \boldsymbol{y}_r)$.

Thus, assume that $\boldsymbol{y}_0, \ldots, \boldsymbol{y}_r$ satisfy the properties stated above. Since we are using positional encodings, the actual input for the first layer of the decoder is the sequence

$$\boldsymbol{y}_0 + \mathrm{pos}(1), \ \boldsymbol{y}_1 + \mathrm{pos}(2), \ \ldots, \ \boldsymbol{y}_r + \mathrm{pos}(r+1).$$

We denote by $\overline{\boldsymbol{y}}_i$ the vector $\boldsymbol{y}_i$ plus its positional encoding. Thus we have that

$$\overline{\boldsymbol{y}}_i = [ \quad [\![ q^{(i)} ]\!], [\![ s^{(i)} ]\!], m^{(i-1)},$$
$$0, \ldots, 0,$$
$$0, \ldots, 0,$$
$$1, (i+1), 1/(i+1), 1/(i+1)^2 \quad ]$$

For the first self attention in Equation (9) we just produce the identity which can be easily implemented with a trivial attention plus the residual connection. Thus, we produce the sequence of vectors $(\boldsymbol{p}_0^1, \boldsymbol{p}_1^1, \ldots, \boldsymbol{p}_r^1)$ such that $\boldsymbol{p}_i^1 = \overline{\boldsymbol{y}}_i$.

Since $\boldsymbol{p}_i^1$ is of the form $[\_, \ldots, \_, 1, i+1, \_, \_]$ by Lemma B.1 we know that if we use $\boldsymbol{p}_i^1$ to attend over the encoder we obtain

$$\mathrm{Att}(\boldsymbol{p}_i^1, \boldsymbol{K^e}, \boldsymbol{V^e}) = [ \quad 0, \ldots, 0,$$
$$0, \ldots, 0,$$
$$[\![ \alpha^{(i+1)} ]\!], \beta^{(i+1)}, \boldsymbol{0}_s, 0,$$
$$0, \ldots, 0 \qquad ]$$

Thus in Equation (10) we finally produce the vector $\boldsymbol{a}_i^1$ given by

$$\boldsymbol{a}_i^1 = \mathrm{Att}(\boldsymbol{p}_i^1, \boldsymbol{K^e}, \boldsymbol{V^e}) + \boldsymbol{p}_i^1 = [ \quad [\![ q^{(i)} ]\!], [\![ s^{(i)} ]\!], m^{(i-1)},$$
$$0, \ldots, 0,$$
$$[\![ \alpha^{(i+1)} ]\!], \beta^{(i+1)}, \boldsymbol{0}_s, 0, \qquad (21)$$
$$1, (i+1), 1/(i+1), 1/(i+1)^2 \quad ]$$

As the final piece of the first decoder layer we use a function $O_1(\cdot)$ (Equation (11)) that satisfies the following lemma.

**Lemma B.2.** *There exists a two-layer feed-forward network $O_1 : \mathbb{Q}^d \to \mathbb{Q}^d$ such that with input vector $\boldsymbol{a}_i^1$ (21) produces as output*

$$O_1(\boldsymbol{a}_i^1) = [ \quad -[\![ q^{(i)} ]\!], -[\![ s^{(i)} ]\!], -m^{(i-1)},$$
$$[\![ q^{(i+1)} ]\!], [\![ v^{(i)} ]\!], m^{(i)}, m^{(i-1)}, 0, 0$$
$$0, \ldots, 0,$$
$$0, \ldots, 0 \qquad ]$$

That is, function $O_1(\cdot)$ simulates transition $\delta(q^{(i)}, s^{(i)})$ to construct $[\![\, q^{(i+1)} \,]\!]$, $[\![\, v^{(i)} \,]\!]$, and $m^{(i)}$ besides some other linear transformations.

We finally produce as the output of the first decoder layer, the sequence $(z_0^1, z_1^1, \ldots, z_r^1)$ such that

$$
\begin{aligned}
\boldsymbol{z}_i^1 = O_1(\boldsymbol{a}_i^1) + \boldsymbol{a}_i^1 \;=\; [\quad & 0, \ldots, 0, \\
& [\![\, q^{(i+1)} \,]\!], [\![\, v^{(i)} \,]\!], m^{(i)}, m^{(i-1)}, 0, 0, \\
& [\![\, \alpha^{(i+1)} \,]\!], \beta^{(i+1)}, \mathbf{0}_s, 0, \\
& 1, (i+1), 1/(i+1), 1/(i+1)^2 \qquad ]
\end{aligned}
\tag{22}
$$

Notice that $z_r^1$ already holds info about $q^{(r+1)}$ and $m^{(r)}$ which we need for constructing vector $\boldsymbol{y}_{r+1}$. The single piece of information that we still need to construct is $s^{(r+1)}$, that is, the symbol under the head of machine $M$ at the next time step (time $r+1$). We next describe how this symbol can be computed with two additional decoder layers.

Recall that $c^{(i)}$ is the cell to which $M$ is pointing to at time $i$, and that it satisfies that $c^{(i)} = m^{(0)} + m^{(1)} + \cdots + m^{(i-1)}$. We can take advantage of this property to prove the following lemma.

**Lemma B.3.** *Let* $\boldsymbol{Z}_i^1 = (z_0^1, z_1^1, \ldots, z_i^1)$. *There exists functions* $Q_2(\cdot)$, $K_2(\cdot)$, *and* $V_2(\cdot)$ *defined by feed-forward networks such that*

$$
\begin{aligned}
\mathrm{Att}(Q_2(z_i^1), K_2(\boldsymbol{Z}_i^1), V_2(\boldsymbol{Z}_i^1)) \;=\; [\quad & 0, \ldots, 0, \\
& \mathbf{0}_q, \mathbf{0}_s, 0, 0, \tfrac{c^{(i+1)}}{(i+1)}, \tfrac{c^{(i)}}{(i+1)}, \\
& 0, \ldots, 0, \\
& 0, \ldots, 0 \qquad ]
\end{aligned}
\tag{23}
$$

Lemma B.3 essentially shows that one can construct a representation for values $c^{(i)}$ and $c^{(i+1)}$ for every possible index $i$. In particular we will know the value $c^{(r+1)}$ that represents the cell to which the machine is pointing to in the next time step.

Continuing with the decoder layer, when using the self attention above and after adding the residual in Equation (9) we obtain the sequence of vectors $(\boldsymbol{p}_0^2, \boldsymbol{p}_1^2, \ldots, \boldsymbol{p}_r^2)$ such that:

$$
\begin{aligned}
\boldsymbol{p}_i^2 \;=\;\;& \mathrm{Att}(Q_2(z_i^1), K_2(\boldsymbol{Z}_i^1), V_2(\boldsymbol{Z}_i^1)) + z_i^1 \\[4pt]
\;=\; [\quad & 0, \ldots, 0, \\
& [\![\, q^{(i+1)} \,]\!], [\![\, v^{(i)} \,]\!], m^{(i)}, m^{(i-1)}, \tfrac{c^{(i+1)}}{(i+1)}, \tfrac{c^{(i)}}{(i+1)}, \\
& [\![\, \alpha^{(i+1)} \,]\!], \beta^{(i+1)}, \mathbf{0}_s, 0, \\
& 1, (i+1), 1/(i+1), 1/(i+1)^2 \qquad ]
\end{aligned}
$$

From vectors $(\boldsymbol{p}_0^2, \boldsymbol{p}_1^2, \ldots, \boldsymbol{p}_r^2)$ and by using the residual connection in Equation (10) plus the output function $O(\cdot)$ in Equation (11) it is not difficult to produce the sequence of vectors $(z_0^2, z_1^2, \ldots, z_r^2)$ such that $z_i^2 = \boldsymbol{p}_i^2$, as the output of the second decoder layer. That is, we have that

$$
\begin{aligned}
\boldsymbol{z}_i^2 \;=\; \boldsymbol{p}_i^2 \;=\; [\quad & 0, \ldots, 0, \\
& [\![\, q^{(i+1)} \,]\!], [\![\, v^{(i)} \,]\!], m^{(i)}, m^{(i-1)}, \tfrac{c^{(i+1)}}{(i+1)}, \tfrac{c^{(i)}}{(i+1)}, \\
& [\![\, \alpha^{(i+1)} \,]\!], \beta^{(i+1)}, \mathbf{0}_s, 0, \\
& 1, (i+1), 1/(i+1), 1/(i+1)^2 \qquad ]
\end{aligned}
$$

We now describe how can we use a third and final decoder layer to produce our desired $s^{(r+1)}$ value (the symbol under the head of $M$ in the next time step). Recall that $\ell(i)$ is the last time (previous to $i$) in which $M$ was pointing to position $c^{(i)}$, or it is $i-1$ if this is the first time that $M$ is pointing to $c^{(i)}$. We can prove the following lemma.

**Lemma B.4.** *There exists functions* $Q_3(\cdot)$, $K_3(\cdot)$, *and* $V_3(\cdot)$ *defined by feed-forward networks such that*

$$
\begin{aligned}
\mathrm{Att}(Q_3(z_i^2), K_3(\boldsymbol{Z}_i^2), V_3(\boldsymbol{Z}_i^2)) \;=\; [\quad & 0, \ldots, 0, \\
& 0, \ldots, 0, \\
& \mathbf{0}_s, 0, [\![\, v^{(\ell(i+1))} \,]\!], \ell(i+1), \\
& 0, \ldots, 0 \qquad ]
\end{aligned}
$$

We prove Lemma B.4 by just showing that, for every $i$ one can attend exactly to position $\ell(i+1)$ and then just copy both values. We do this by taking advantage of the values $c^{(i)}$ and $c^{(i+1)}$ previously computed for every index $i$. Then we have that $\boldsymbol{p}_i^3$ is given by

$$
\begin{aligned}
\boldsymbol{p}_i^3 \;=\;&\; \text{Att}(Q_3(\boldsymbol{z}_i^2), K_3(\boldsymbol{Z}_i^2), V_3(\boldsymbol{Z}_i^2)) + \boldsymbol{z}_i^2 \\[4pt]
=\;&\; [\quad 0, \ldots, 0 \\
&\; [\![\, q^{(i+1)} \,]\!], [\![\, v^{(i)} \,]\!], m^{(i)}, m^{(i-1)}, \tfrac{c^{(i+1)}}{(i+1)}, \tfrac{c^{(i)}}{(i+1)}, \\
&\; [\![\, \alpha^{(i+1)} \,]\!], \beta^{(i+1)}, [\![\, v^{(\ell(i+1))} \,]\!], \ell(i+1), \\
&\; 1, (i+1), 1/(i+1), 1/(i+1)^2 \qquad\qquad ]
\end{aligned}
\tag{24}
$$

From vectors $(\boldsymbol{p}_0^3, \boldsymbol{p}_1^3, \ldots, \boldsymbol{p}_r^3)$ and by using the residual connection in Equation (10) plus the output function $O(\cdot)$ in Equation (11)) it is not difficult to produce the sequence of vectors $(\boldsymbol{z}_0^3, \boldsymbol{z}_1^3, \ldots, \boldsymbol{z}_r^3)$ such that $\boldsymbol{z}_i^3 = \boldsymbol{p}_i^3$, as the output of the third and final decoder layer, and thus we have that

$$
\begin{aligned}
\boldsymbol{z}_i^3 \;=\; \boldsymbol{p}_i^3 \;=\;&\; [\quad 0, \ldots, 0, \\
&\; [\![\, q^{(i+1)} \,]\!], [\![\, v^{(i)} \,]\!], m^{(i)}, m^{(i-1)}, \tfrac{c^{(i+1)}}{(i+1)}, \tfrac{c^{(i)}}{(i+1)}, \\
&\; [\![\, \alpha^{(i+1)} \,]\!], \beta^{(i+1)}, [\![\, v^{(\ell(i+1))} \,]\!], \ell(i+1), \\
&\; 1, (i+1), 1/(i+1), 1/(i+1)^2 \qquad\qquad ]
\end{aligned}
$$

We finish our construction by using the final transformation function $F(\cdot)$ in Equation (12) in the following lemma.

**Lemma B.5.** *There exists a function* $F : \mathbb{Q}^d \to \mathbb{Q}^d$ *defined by a feed-forward network such that*

$$
\begin{aligned}
F(\boldsymbol{z}_r^3) \;=\;&\; [\quad [\![\, q^{(r+1)} \,]\!], [\![\, s^{(r+1)} \,]\!], m^{(r)}, \\
&\; 0, \ldots, 0, \\
&\; 0, \ldots, 0, \\
&\; 0, \ldots, 0 \qquad\qquad ] \\[6pt]
=\;&\; \boldsymbol{y}_{r+1}
\end{aligned}
$$

We prove Lemma B.5 as follows (details in the next section). We show that one can construct a feed-forward network that with input $\boldsymbol{z}_r^3$ implements the following to produce $\boldsymbol{y}_{r+1}$. We move $[\![\, q^{(r+1)} \,]\!]$ and $m^{(r)}$ to its corresponding position in $\boldsymbol{y}_{r+1}$. Then

1. if $\beta^{(r+1)} = r + 1$ then we let $[\![\, s^{(r+1)} \,]\!] = [\![\, \alpha^{(r+1)} \,]\!]$,
2. if $\beta^{(r+1)} < r + 1$ and $\ell(r+1) = r$, then we let $[\![\, s^{(r+1)} \,]\!] = [\![\, \# \,]\!]$, and
3. if $\beta^{(r+1)} < r + 1$ and $\ell(r+1) \neq r$, then we let $[\![\, s^{(r+1)} \,]\!] = [\![\, v^{(\ell(r+1))} \,]\!]$.

Finally, we move $[\![\, s^{(r+1)} \,]\!]$ to its corresponding position in $\boldsymbol{y}_{r+1}$ and we make all other positions 0. The correctness of the above rules is given by the following argument. If $\beta^{(r+1)} = r+1$ then by the definition of $\beta^{(r+1)}$ we have that $r + 1 \leq n$ which implies that $\alpha^{(r+1)} = s_{r+1} = s^{(r+1)}$ and thus rule (1) above is correct. If $\beta^{(r+1)} < r+1$ then we know that $r + 1 > n$ and if $\ell(r+1) = r$ then by the definition of $\ell(\cdot)$ we know that $c^{(r+1)}$ is visited by $M$ for the first time at time $r + 1$, which implies that $s^{(r+1)} = \#$ and thus rule (2) is also correct. Finally, If $\beta^{(r+1)} < r+1$ and $\ell(r+1) \neq r$, then we know that $c^{(r+1)}$ has been visited before at time $\ell(r+1)$, and thus $s^{(r+1)} = v^{(\ell(r+1))}$ which implies the correctness of rule (3).

FINAL STEP

We now can use our $\text{Trans}_M$ network to construct the recognizer $A = (\Sigma, f_{\text{pos}}, \text{Trans}_M, \boldsymbol{y}_0, \mathbb{F})$ such that $A$ accepts $w$ if and only if $M = (Q, \Sigma, \delta, q_{\text{init}}, F)$ accepts $w$. Notice that $M$ accepts $w$ if and only if an accepting state $q_f \in F$ is reached at some time step, say $t^\star$. By our construction above we know that, with input $f_{\text{pos}}(w)$ our network $\text{Trans}_M$ produces a vector $\boldsymbol{y}_{t^\star}$ that contains $q_f$ as a one-hot vector. Thus, we can simply use $\mathbb{F}$ as the set of all vectors in $\mathbb{Q}^d$ that contains a one-hot representation of a state in $F$. Formally, $\mathbb{F} = \{ [\![\, q \,]\!] \mid q \in F \} \times \mathbb{Q}^{d-|Q|}$. It is straightforward to see that membership in $\mathbb{F}$ can be checked in linear time.

### B.4.3 DETAILED PROOFS OF INTERMEDIATE LEMMAS

**Proof of Lemma B.1.** Let $\boldsymbol{q} \in \mathbb{Q}^d$ be a vector such that $\boldsymbol{q} = [\_, \ldots, \_, 1, j, \_, \_]$ where $j \in \mathbb{N}$ and '$\_$' is an arbitrary value. We next prove that

$$\text{Att}(\boldsymbol{q}, \boldsymbol{K^e}, \boldsymbol{V^e}) \;=\; [\;\; 0, \ldots, 0,$$
$$0, \ldots, 0,$$
$$[\![\, \alpha^{(j)} \,]\!], \beta^{(j)}, \boldsymbol{0}_s, 0,$$
$$0, \ldots, 0 \qquad\qquad ]$$

where $\alpha^{(j)}$ and $\beta^{(j)}$ are defined as

$$\alpha^{(j)} \;=\; \begin{cases} s_j & 1 \le j \le n \\ s_n & j > n \end{cases}$$

$$\beta^{(j)} \;=\; \begin{cases} j & j \le n \\ n & j > n \end{cases}$$

Recall that $\boldsymbol{K^e} = (\boldsymbol{k}_1, \ldots, \boldsymbol{k}_n)$ is such that $\boldsymbol{k}_i = [\, 0, \ldots, 0, i, -1, 0, 0 \,]$. Then we have that

$$\text{score}_\varphi(\boldsymbol{q}, \boldsymbol{k}_i) = \varphi(\langle \boldsymbol{q}, \boldsymbol{k}_i \rangle) = -|\langle \boldsymbol{q}, \boldsymbol{k}_i \rangle| = -|i - j|.$$

Notice that, if $j \le n$, then the above expression is maximized when $i = j$. Otherwise, if $j > n$ then the expression is maximized when $i = n$. Then $\text{Att}(\boldsymbol{q}, \boldsymbol{K^e}, \boldsymbol{V^e}) = \boldsymbol{v}_{i^\star}$ where $i^\star = j$ if $j \le n$ and $i^\star = n$ if $j > n$. We note that $i^\star$ as just defined is exactly $\beta^{(j)}$. Thus, given that $\boldsymbol{v}_i$ is defined as

$$\boldsymbol{v}_i \;=\; [\;\; 0, \ldots, 0,$$
$$0, \ldots, 0,$$
$$[\![\, s_i \,]\!], i, \boldsymbol{0}_s, 0,$$
$$0, \ldots, 0 \qquad\qquad ]$$

we obtain that

$$\text{Att}(\boldsymbol{q}, \boldsymbol{K^e}, \boldsymbol{V^e}) \;=\; \boldsymbol{v}_{i^\star} \;=\; [\;\; 0, \ldots, 0,$$
$$0, \ldots, 0,$$
$$[\![\, s_{i^\star} \,]\!], i^\star, \boldsymbol{0}_s, 0,$$
$$0, \ldots, 0 \qquad\qquad ]$$

$$=\; [\;\; 0, \ldots, 0,$$
$$0, \ldots, 0,$$
$$[\![\, \alpha^{(j)} \,]\!], \beta^{(j)}, \boldsymbol{0}_s, 0,$$
$$0, \ldots, 0 \qquad\qquad ]$$

which is what we wanted to prove. $\qquad\square$

**Proof of Lemma B.2.** In order to prove the lemma we need some intermediate notions and properties. Assume that the enumeration $\pi_1 : \Sigma \to \{1, \ldots, |\Sigma|\}$ is the one used to construct the one-hot vectors $[\![\, s \,]\!]$ for $s \in \Sigma$, and that $\pi_2 : Q \to \{1, \ldots, |Q|\}$ is the one used to construct $[\![\, q \,]\!]$ with $q \in Q$. Using $\pi_1$ and $\pi_2$ one can construct an enumeration for the pairs in $Q \times \Sigma$ and then construct one-hot vectors for pairs in this set. Formally, given $(q, s) \in Q \times \Sigma$ we denote by $[\![\, (q, s) \,]\!]$ a one-hot vector with a $1$ in position $(\pi_1(s) - 1)|Q| + \pi_2(q)$ and a $0$ in every other position. To simplify the notation we use $\pi(q, s)$ to denote $(\pi_1(s) - 1)|Q| + \pi_2(q)$. One can similarly construct an enumeration $\pi'$ for $Q \times \Sigma \times \{-1, 1\}$ such that $\pi'(q, s, m) = \pi(q, s)$ if $m = -1$ and $\pi'(q, s, m) = |Q||\Sigma| + \pi(q, s)$ if $m = 1$. We denote by $[\![\, (q, s, m) \,]\!]$ the corresponding one-hot vector for every $(q, s, m) \in Q \times \Sigma \times \{-1, 1\}$. We next prove three helping properties. In every case $q \in Q$, $s \in \Sigma$, $m \in \{-1, 1\}$, and $\delta(\cdot, \cdot)$ is the transition function of machine $M$.

1. There exists $f_1 : \mathbb{Q}^{|Q|+|\Sigma|} \to \mathbb{Q}^{|Q||\Sigma|}$ such that $f_1([\![\, q \,]\!], [\![\, s \,]\!]) = [\![\, (q, s) \,]\!]$.

2. There exists $f_\delta : \mathbb{Q}^{|Q||\Sigma|} \to \mathbb{Q}^{2|Q||\Sigma|}$ such that $f_\delta([\![\, (q, s) \,]\!]) = [\![\, \delta(q, s) \,]\!]$.

3. There exists $f_2 : \mathbb{Q}^{2|Q||\Sigma|} \to \mathbb{Q}^{|Q|+|\Sigma|+1}$ such that $f_2([\![\, (q, s, m) \,]\!]) = [\![\, q \,]\!], [\![\, s \,]\!], m ]$.

To show (1), lets denote by $\boldsymbol{S}_i$, with $i \in \{1, \ldots, |\Sigma|\}$, a matrix of dimensions $|\Sigma| \times |Q|$ such that $\boldsymbol{S}_i$ has its $i$-th row with 1's and it is 0 everywhere else. We note that for every $s \in \Sigma$ it holds that $[\![\, s \,]\!] \boldsymbol{S}_i = \mathbf{1}$ if and only if $i = \pi_1(s)$ and it is $\mathbf{0}$ otherwise. Now, consider the vector $\boldsymbol{v}_{(q,s)}$

$$\boldsymbol{v}_{(q,s)} = [\, [\![\, q \,]\!] + [\![\, s \,]\!] \boldsymbol{S}_1, [\![\, q \,]\!] + [\![\, s \,]\!] \boldsymbol{S}_2, \ldots, [\![\, q \,]\!] + [\![\, s \,]\!] \boldsymbol{S}_{|\Sigma|} \,]$$

We first note that for every $i \in \{1, \ldots, |\Sigma|\}$, if $i \neq \pi_1(s)$ then $[\![\, q \,]\!] + [\![\, s \,]\!] \boldsymbol{S}_i = [\![\, q \,]\!] + \mathbf{0} = [\![\, q \,]\!]$. Moreover $[\![\, q \,]\!] + [\![\, s \,]\!] \boldsymbol{S}_{\pi_1(s)} = [\![\, q \,]\!] + \mathbf{1}$ is a vector that has a 2 exactly at index $\pi_2(q)$, and it is 1 in all other positions. Thus, the vector $\boldsymbol{v}_{(q,s)}$ has a 2 exactly at position $(\pi_1(s) - 1)|Q| + \pi_2(q)$ and it is either 0 or 1 in every other position. Now, lets denote by $\boldsymbol{o}$ a vector in $\mathbb{Q}^{|Q||\Sigma|}$ that has a 1 in every position and consider the following affine transformation

$$g_1([\, [\![\, q \,]\!], [\![\, s \,]\!] \,]) = \boldsymbol{v}_{(q,s)} - \boldsymbol{o}. \tag{25}$$

Vector $g_1([\, [\![\, q \,]\!], [\![\, s \,]\!] \,])$ has a 1 only at position $(\pi_1(s) - 1)|Q| + \pi_2(q) = \pi(q, s)$ and it is less than or equal to 0 in every other position. Thus, to construct $f_1(\cdot)$ we apply the piecewise-linear sigmoidal activation $\sigma(\cdot)$ (see Equation (1)) to obtain

$$f_1([\, [\![\, q \,]\!], [\![\, s \,]\!] \,]) = \sigma(g_1([\, [\![\, q \,]\!], [\![\, s \,]\!] \,])) = \sigma(\boldsymbol{v}_{(q,s)} - \boldsymbol{o}) = [\![\, (q, s) \,]\!],$$

which is what we wanted.

Now, to show (2), lets denote by $\boldsymbol{M}^\delta$ a matrix of dimensions $(|Q||\Sigma|) \times (2|Q||\Sigma|)$ constructed as follows. For $(q, s) \in Q \times \Sigma$, if $\delta(q, s) = (p, r, m)$ then $\boldsymbol{M}^\delta$ has a 1 at position $(\pi(q, s), \pi'(p, r, m))$ and it has a 0 in every other position, that is

$$\boldsymbol{M}^\delta_{\pi(q,s),:} = [\![\, (p, r, m) \,]\!] = [\![\, \delta(q, s) \,]\!].$$

It is straightforward to see that $[\![\, (q, s) \,]\!] \boldsymbol{M}^\delta = [\![\, \delta(q, s) \,]\!]$, and thus we can define $f_2(\cdot)$ as

$$f_2([\![\, (q, s) \,]\!]) = [\![\, (q, s) \,]\!] \boldsymbol{M}^\delta = [\![\, \delta(q, s) \,]\!].$$

To show (3), consider the matrix $\boldsymbol{A}$ of dimensions $(2|Q||\Sigma|) \times (|Q| + |\Sigma| + 1)$ such that

$$\boldsymbol{A}_{\pi'(q,s,m),:} = [\, [\![\, q \,]\!], [\![\, s \,]\!], m \,].$$

Then we define $f_3(\cdot)$ as

$$f_3([\![\, (q, s, m) \,]\!]) = [\![\, (q, s, m) \,]\!] \boldsymbol{A} = [\, [\![\, q \,]\!], [\![\, s \,]\!], m \,].$$

We are now ready to begin with the proof of the lemma. Recall that $\boldsymbol{a}_i^1$ is given by

$$
\begin{aligned}
\boldsymbol{a}_i^1 \;=\; [\; & [\![\, q^{(i)} \,]\!], [\![\, s^{(i)} \,]\!], m^{(i-1)}, \\
& 0, \ldots, 0, \\
& [\![\, \alpha^{(i+1)} \,]\!], \beta^{(i+1)}, \mathbf{0}_s, 0, \\
& 1, (i + 1), 1/(i + 1), 1/(i + 1)^2 \;]
\end{aligned}
$$

We need to construct a function $O_1 : \mathbb{Q}^d \to \mathbb{Q}^d$ such that

$$
\begin{aligned}
O_1(\boldsymbol{a}_i^1) \;=\; [\; & -[\![\, q^{(i)} \,]\!], -[\![\, s^{(i)} \,]\!], -m^{(i-1)}, \\
& [\![\, q^{(i+1)} \,]\!], [\![\, v^{(i)} \,]\!], m^{(i)}, m^{(i-1)}, 0, 0 \\
& 0, \ldots, 0, \\
& 0, \ldots, 0 \;]
\end{aligned}
$$

We first use function $h_1(\cdot)$ that works as follows. Lets denote by $\hat{m}^{(i-1)}$ the value $\frac{1}{2}m^{(i-1)} + \frac{1}{2}$. Note that $\hat{m}^{(i-1)}$ is 0 if $m^{(i-1)} = -1$, it is $\frac{1}{2}$ if $m^{(i-1)} = 0$ and it is 1 if $m^{(i-1)} = 1$. We use this transformation just to represent $m^{(i-1)}$ with a value between 0 and 1. Now, consider $h_1(\boldsymbol{a}_i^1)$ defined by

$$h_1(\boldsymbol{a}_i^1) \;=\; [\, [\![\, q^{(i)} \,]\!], [\![\, s^{(i)} \,]\!], \hat{m}^{(i-1)}, g_1([\, [\![\, q^{(i)} \,]\!], [\![\, s^{(i)} \,]\!] \,]) \,]$$

where $g_1(\cdot)$ is the function defined above in Equation (25). It is clear that $h_1(\cdot)$ is an affine transformation. Moreover, we note that except for $g_1([\, [\![\, q^{(i)} \,]\!], [\![\, s^{(i)} \,]\!] \,])$ all values in $h_1(\boldsymbol{a}_i^1)$ are between 0 and 1. Thus if we apply function $\sigma(\cdot)$ to $h_1(\boldsymbol{a}_i^1)$ we obtain

$$
\begin{aligned}
\sigma(h_1(\boldsymbol{a}_i^1)) \;&=\; [\, [\![\, q^{(i)} \,]\!], [\![\, s^{(i)} \,]\!], \hat{m}^{(i-1)}, \sigma(g_1([\, [\![\, q^{(i)} \,]\!], [\![\, s^{(i)} \,]\!] \,])) \,] \\
&=\; [\, [\![\, q^{(i)} \,]\!], [\![\, s^{(i)} \,]\!], \hat{m}^{(i-1)}, [\![\, (q^{(i)}, s^{(i)}) \,]\!] \,]
\end{aligned}
$$

Then we can define $h_2(\cdot)$ such that

$$
\begin{aligned}
h_2(\sigma(h_1(\boldsymbol{a}_i^1))) &= [\,[\![\,q^{(i)}\,]\!],[\![\,s^{(i)}\,]\!],2\hat{m}^{(i-1)}-1,f_2([\![\,(q^{(i)},s^{(i)})\,]\!])\,] \\
&= [\,[\![\,q^{(i)}\,]\!],[\![\,s^{(i)}\,]\!],m^{(i-1)},[\![\,\delta(q^{(i)},s^{(i)})\,]\!]\,] \\
&= [\,[\![\,q^{(i)}\,]\!],[\![\,s^{(i)}\,]\!],m^{(i-1)},[\![\,(q^{(i+1)},v^{(i)},m^{(i)})\,]\!]\,]
\end{aligned}
$$

Now we can define $h_3(\cdot)$ as

$$
\begin{aligned}
h_3(h_2(\sigma(h_1(\boldsymbol{a}_i^1)))) &= [\,[\![\,q^{(i)}\,]\!],[\![\,s^{(i)}\,]\!],m^{(i-1)},f_3([\![\,(q^{(i+1)},v^{(i)},m^{(i)})\,]\!])\,] \\
&= [\,[\![\,q^{(i)}\,]\!],[\![\,s^{(i)}\,]\!],m^{(i-1)},[\![\,q^{(i+1)}\,]\!],[\![\,v^{(i)}\,]\!],m^{(i)}\,]
\end{aligned}
$$

Finally we can apply a function $h_4(\cdot)$ to just reorder the values and multiply some components by $-1$ to complete our construction

$$
\begin{aligned}
O_1(\boldsymbol{a}_i^1) = h_4(h_3(h_2(\sigma(h_1(\boldsymbol{a}_i^1))))) = [\quad &-[\![\,q^{(i)}\,]\!],-[\![\,s^{(i)}\,]\!],-m^{(i-1)}, \\
&[\![\,q^{(i+1)}\,]\!],[\![\,v^{(i)}\,]\!],m^{(i)},m^{(i-1)},0,0 \\
&0,\ldots,0, \\
&0,\ldots,0 \qquad\qquad ]
\end{aligned}
$$

We note that we applied a single non-linearity and all other functions are affine transformations. Thus $O_1(\cdot)$ can be implemented with a two-layer feed-forward network.

$\square$

**Proof of Lemma B.3.** Recall that $\boldsymbol{z}_i^1$ is the following vector

$$
\begin{aligned}
\boldsymbol{z}_i^1 = [\quad &0,\ldots,0, \\
&[\![\,q^{(i+1)}\,]\!],[\![\,v^{(i)}\,]\!],m^{(i)},m^{(i-1)},0,0, \\
&[\![\,\alpha^{(i+1)}\,]\!],\beta^{(i+1)},\boldsymbol{0}_s,0, \\
&1,(i+1),1/(i+1),1/(i+1)^2 \qquad ]
\end{aligned}
$$

We consider $Q_2 : \mathbb{Q}^d \to \mathbb{Q}^d$ and $K_2 : \mathbb{Q}^d \to \mathbb{Q}^d$ as trivial functions that for every input produce an output vector composed of only 0's. Moreover, we consider $V_2 : \mathbb{Q}^d \to \mathbb{Q}^d$ such that for every $j \in \{0,1,\ldots,i\}$

$$
\begin{aligned}
V_2(\boldsymbol{z}_j^1) = [\quad &0,\ldots,0, \\
&\boldsymbol{0}_q,\boldsymbol{0}_s,0,0,m^{(j)},m^{(j-1)}, \\
&0,\ldots,0, \\
&0,\ldots,0 \qquad\qquad ]
\end{aligned}
$$

Then, since $K_2(\boldsymbol{z}_j^1)$ is the vector with only zeros, then $\mathrm{score}_\varphi(Q_2(\boldsymbol{z}_i^1),K_2(\boldsymbol{z}_j^1)) = 0$ for every $j \in \{0,\ldots,i\}$. Thus, we have that the attention $\mathrm{Att}(Q_2(\boldsymbol{z}_i^1),K_2(\boldsymbol{Z}_i^1),V_2(\boldsymbol{Z}_i^1))$ that we need to compute is just the average of all the vectors in $V_2(\boldsymbol{Z}_i^1) = (V_2(\boldsymbol{z}_0^1),\ldots,\boldsymbol{z}_i^1)$, that is

$$
\begin{aligned}
\mathrm{Att}(Q_2(\boldsymbol{z}_i^1),K_2(\boldsymbol{Z}_i^1),V_2(\boldsymbol{Z}_i^1)) &= \tfrac{1}{(i+1)}\sum_{j=0}^i V_2(\boldsymbol{z}_j^1) \\
&= [\quad 0,\ldots,0, \\
&\qquad \boldsymbol{0}_q,\boldsymbol{0}_s,0,0,\tfrac{1}{(i+1)}\sum_{j=0}^i m^{(j)},\tfrac{1}{(i+1)}\sum_{j=0}^i m^{(j-1)}, \\
&\qquad 0,\ldots,0, \\
&\qquad 0,\ldots,0 \qquad\qquad ]
\end{aligned}
$$

Then, since $m^{(0)}+\cdots+m^{(i)} = c^{(i+1)}$ and $m^{(-1)}+m^{(0)}+\cdots+m^{(i-1)} = c^{(i)}$ we have that

$$
\begin{aligned}
\mathrm{Att}(Q_2(\boldsymbol{z}_i^1),K_2(\boldsymbol{Z}_i^1),V_2(\boldsymbol{Z}_i^1)) &= [\quad 0,\ldots,0, \\
&\qquad \boldsymbol{0}_q,\boldsymbol{0}_s,0,0,\tfrac{c^{(i+1)}}{(i+1)},\tfrac{c^{(i)}}{(i+1)}, \\
&\qquad 0,\ldots,0, \\
&\qquad 0,\ldots,0 \qquad\qquad ]
\end{aligned}
$$

which is exactly what we wanted to show.

$\square$

**Proof of Lemma B.4.** Recall that $\boldsymbol{z}_i^2$ is the following vector

$$
\begin{aligned}
\boldsymbol{z}_i^2 = [\quad &0,\ldots,0, \\
&[\![\,q^{(i+1)}\,]\!],[\![\,v^{(i)}\,]\!],m^{(i)},m^{(i-1)},\tfrac{c^{(i+1)}}{(i+1)},\tfrac{c^{(i)}}{(i+1)}, \\
&[\![\,\alpha^{(i+1)}\,]\!],\beta^{(i+1)},\boldsymbol{0}_s,0, \\
&1,(i+1),1/(i+1),1/(i+1)^2 \qquad ]
\end{aligned}
$$

We need to construct functions $Q_3(\cdot)$, $K_3(\cdot)$, and $V_3(\cdot)$ such that

$$\text{Att}(Q_3(\boldsymbol{z}_i^2), K_3(\boldsymbol{Z}_i^2), V_3(\boldsymbol{Z}_i^2)) \quad = \quad [ \quad 0,\ldots,0,$$
$$0,\ldots,0,$$
$$\boldsymbol{0}_s, 0, [\![\, v^{(\ell(i+1))} \,]\!], \ell(i+1),$$
$$0,\ldots,0 \qquad ]$$

We first define the query function $Q_3 : \mathbb{Q}^d \to \mathbb{Q}^d$ such that

$$Q_3(\boldsymbol{z}_i^2) \quad = \quad [ \quad 0,\ldots,0$$
$$0,\ldots,0,$$
$$0,\ldots,0,$$
$$0, \frac{c^{(i+1)}}{(i+1)}, \frac{1}{(i+1)}, \frac{1}{3(i+1)^2} \quad ]$$

Now, for every $j \in \{0, 1, \ldots, i\}$ we define $K_3 : \mathbb{Q}^d \to \mathbb{Q}^d$ and $V_3 : \mathbb{Q}^d \to \mathbb{Q}^d$ such that

$$K_3(\boldsymbol{z}_j^2) \quad = \quad [ \quad 0,\ldots,0$$
$$0,\ldots,0,$$
$$0,\ldots,0,$$
$$0, \frac{1}{(j+1)}, \frac{-c^{(j)}}{(j+1)}, \frac{1}{(j+1)^2} \quad ]$$

$$V_3(\boldsymbol{z}_j^2) \quad = \quad [ \quad 0,\ldots,0,$$
$$0,\ldots,0,$$
$$\boldsymbol{0}_s, 0, [\![\, v^{(j)} \,]\!], j,$$
$$0,\ldots,0 \qquad ]$$

It is clear that the three functions are linear transformations and thus they can be defined by feed-forward networks. Consider now the attention $\text{Att}(Q_3(\boldsymbol{z}_i^2), K_3(\boldsymbol{Z}_i^2), V_3(\boldsymbol{Z}_i^2))$. In order to compute this value, and since we are considering hard attention, we need to find the value $j \in \{0, 1, \ldots, i\}$ that maximizes

$$\text{score}_\varphi(Q_3(\boldsymbol{z}_i^2), K_3(\boldsymbol{z}_j^2)) = \varphi(\langle Q_3(\boldsymbol{z}_i^2), K_3(\boldsymbol{z}_j^2)\rangle).$$

Actually, assuming that such value is unique, lets say $j^\star$, then we have that

$$\text{Att}(Q_3(\boldsymbol{z}_i^2), K_3(\boldsymbol{Z}_i^2), V_3(\boldsymbol{Z}_i^2)) = V_3(\boldsymbol{z}_{j^\star}^2).$$

We next show that given our definitions above, it always holds that $j^\star = \ell(i+1)$ and then $V_3(\boldsymbol{z}_{j^\star}^2)$ is exactly the vector that we wanted to obtain.

To simplify the notation, we denote by $\chi_j^i$ the dot product $\langle Q_3(\boldsymbol{z}_i^2), K_3(\boldsymbol{z}_j^2)\rangle$. Thus, we need to find $j^\star = \arg\max_j \varphi(\chi_j^i)$. Moreover, given the definition of $\varphi$ (see Equation (20))we have that

$$\underset{j\in\{0,\ldots,i\}}{\arg\max} \varphi(\chi_j^i) = \underset{j\in\{0,\ldots,i\}}{\arg\min} |\chi_j^i|.$$

Then, it is enough to prove that

$$\underset{j\in\{0,\ldots,i\}}{\arg\min} |\chi_j^i| = \ell(i+1).$$

Now, by our definition of $Q_3(\cdot)$ and $K_3(\cdot)$ we have that

$$\chi_j^i \quad = \quad \frac{c^{(i+1)}}{(i+1)(j+1)} - \frac{c^{(j)}}{(i+1)(j+1)} + \frac{1}{3(i+1)^2(j+1)^2}$$
$$= \quad \varepsilon_i\varepsilon_j \cdot \left( c^{(i+1)} - c^{(j)} + \frac{\varepsilon_i\varepsilon_j}{3} \right)$$

where $\varepsilon_k = \frac{1}{(k+1)}$. We next prove the following auxiliary property.

If $j_1$ is such that $c^{(j_1)} \neq c^{(i+1)}$ and $j_2$ is such that $c^{(j_2)} = c^{(i+1)}$, then $|\chi_{j_2}^i| < |\chi_{j_1}^i|$. $\qquad$ (26)

In order to prove (26), assume first that $j_1 \in \{0, \ldots, i\}$ is such that $c^{(j_1)} \neq c^{(i+1)}$. Then we have that $|c^{(i+1)} - c^{(j_1)}| \geq 1$ since $c^{(i+1)}$ and $c^{(j_1)}$ are integer values. From this we have two possibilities for $\chi_{j_1}^i$:

- If $c^{(i+1)} - c^{(j_1)} \leq -1$, then

$$\chi^i_{j_1} \leq -\varepsilon_i \varepsilon_{j_1} + \frac{(\varepsilon_i \varepsilon_{j_1})^2}{3}.$$

Notice that $1 \geq \varepsilon_{j_1} \geq \varepsilon_i > 0$. Then we have that $\varepsilon_i \varepsilon_{j_1} \geq (\varepsilon_i \varepsilon_{j_1})^2 > \frac{1}{3}(\varepsilon_i \varepsilon_{j_1})^2$, and thus

$$|\chi^i_{j_1}| \geq \varepsilon_i \varepsilon_{j_1} - \frac{(\varepsilon_i \varepsilon_{j_1})^2}{3}$$

Finally, and using again that $1 \geq \varepsilon_{j_1} \geq \varepsilon_i > 0$, from the above equation we obtain that

$$|\chi^i_{j_1}| \geq \varepsilon_i \varepsilon_i - \frac{(\varepsilon_i \varepsilon_{j_1})^2}{3} \geq (\varepsilon_i)^2 - \frac{(\varepsilon_i)^2}{3} \geq \frac{2(\varepsilon_i)^2}{3}.$$

- If $c^{(i+1)} - c^{(j_1)} \geq 1$, then $\chi^i_{j_1} \geq \varepsilon_i \varepsilon_{j_1} + \frac{1}{3}(\varepsilon_i \varepsilon_{j_1})^2$ and since $1 \geq \varepsilon_{j_1} \geq \varepsilon_i > 0$ we obtain that $|\chi^i_{j_1}| \geq \varepsilon_i \varepsilon_{j_1} \geq \varepsilon_i \varepsilon_i \geq \frac{2}{3}(\varepsilon_i)^2$.

Thus, we have that if $c^{(j_1)} \neq c^{(i+1)}$ then $|\chi^i_{j_1}| \geq \frac{2}{3}(\varepsilon_i)^2$.

Now assume $j_2 \in \{0, \ldots, i\}$ is such that $c^{(j_2)} = c^{(i+1)}$. In this case we have that

$$|\chi^i_{j_2}| = \frac{(\varepsilon_i \varepsilon_{j_2})^2}{3} = \frac{(\varepsilon_i)^2(\varepsilon_{j_2})^2}{3} \leq \frac{(\varepsilon_i)^2}{3}.$$

We showed that if $c^{(j_1)} \neq c^{(i+1)}$ then $|\chi^i_{j_1}| \geq \frac{2}{3}(\varepsilon_i)^2$ and if $c^{(j_2)} = c^{(i+1)}$ then $|\chi^i_{j_2}| \leq \frac{1}{3}(\varepsilon_i)^2$ which implies that $|\chi^i_{j_2}| < |\chi^i_{j_1}|$. This completes the proof of the property in (26).

We have now all the necessary to prove that $\arg\min_j |\chi^i_j| = \ell(i+1)$. Recall first that $\ell(i+1)$ is defined as

$$\ell(i+1) = \begin{cases} \max\{j \mid j \leq i \text{ and } c^{(j)} = c^{(i+1)}\} & \text{if there exists } j \leq i \text{ s.t. } c^{(j)} = c^{(i+1)}, \\ i & \text{in other case.} \end{cases}$$

Assume first that there exists $j \leq i$ such that $c^{(j)} = c^{(i+1)}$. By (26) we know that

$$
\begin{aligned}
\arg\min_{j \in \{0,\ldots,i\}} |\chi^i_j| &= \arg\min_{j \text{ s.t. } c^{(j)}=c^{(i+1)}} |\chi^i_j| \\
&= \arg\min_{j \text{ s.t. } c^{(j)}=c^{(i+1)}} \frac{(\varepsilon_i \varepsilon_j)^2}{3} \\
&= \arg\min_{j \text{ s.t. } c^{(j)}=c^{(i+1)}} \varepsilon_j \\
&= \arg\min_{j \text{ s.t. } c^{(j)}=c^{(i+1)}} \frac{1}{j+1} \\
&= \max_{j \text{ s.t. } c^{(j)}=c^{(i+1)}} j \\
&= \max\{j \mid c^{(j)} = c^{(i+1)}\}
\end{aligned}
$$

On the contrary, assume that for every $j \leq i$ it holds that $c^{(j)} \neq c^{(i+1)}$. We will prove that in this case $|\chi^j_i| < |\chi^i_i|$ for every $j < i$ and thus $\arg\min_{j \in \{0,\ldots,i\}} |\chi^i_j| = i$. Now, since $c^{(j)} \neq c^{(i+1)}$ for every $j \leq i$, then $c^{(i+1)}$ is a cell that has never been visited before by $M$. Given that $M$ never makes a transition to the left of its initial cell, then cell $c^{(i+1)}$ is *a cell to the right* of every other previously visited cell. This implies that $c^{(i+1)} > c^{(j)}$ for every $j \leq i$. Thus, for every $j \leq i$ we have $c^{(i+1)} - c^{(j)} \geq 1$. This implies that $|\chi^i_j| = \chi^i_j \geq \varepsilon_i \varepsilon_j + \frac{1}{3}(\varepsilon_i \varepsilon_j)^2$. Moreover, notice that if $j < i$ then $\varepsilon_j > \varepsilon_i$ and thus, if $j < i$ we have that

$$|\chi^i_j| \geq \varepsilon_i \varepsilon_j + \frac{(\varepsilon_i \varepsilon_j)^2}{3} > \varepsilon_i \varepsilon_i + \frac{(\varepsilon_i \varepsilon_i)^2}{3} = |\chi^i_i|$$

which implies that $\arg\min_{j\in\{0,\dots,i\}}|\chi_j^i| = i$. Summing it up, we have shown that

$$\arg\min_{j\in\{0,\dots,i\}}|\chi_j^i| = \begin{cases} \max\{j \mid c^{(j)} = c^{(i+1)}\} & \text{if there exists } j \leq i \text{ s.t. } c^{(j)} = c^{(i+1)}, \\ i & \text{in other case.} \end{cases}$$

which is exactly the definition of $\ell(i+1)$. This completes the proof of the lemma. $\quad\square$

**Proof of Lemma B.5.** Before going to the proof of Lemma-B.5 we prove the following helping result that allows us to implement a particular type of *if* statement with a feed-forward network.

**Lemma B.6.** *Let* $\boldsymbol{x} \in \{0,1\}^m$ *and* $\boldsymbol{y}, \boldsymbol{z} \in \{0,1\}^n$ *be binary vectors, and let* $b \in \{0,1\}$. *There exists a two-layer feed-forward network* $f : \mathbb{Q}^{m+2n+1} \to \mathbb{Q}^{m+n}$ *such that*

$$f([\boldsymbol{x},\boldsymbol{y},\boldsymbol{z},b]) = \begin{cases} [\boldsymbol{x},\boldsymbol{y}] & \text{if } b = 0, \\ [\boldsymbol{x},\boldsymbol{z}] & \text{if } b = 1. \end{cases}$$

*Proof.* Consider the function $f_1 : \mathbb{Q}^{m+2n+1} \to \mathbb{Q}^{m+2n}$ such that

$$f_1([\boldsymbol{x},\boldsymbol{y},\boldsymbol{z},b]) = [\boldsymbol{x},\boldsymbol{y}-b\mathbf{1},\boldsymbol{z}+b\mathbf{1}-\mathbf{1}]$$

where $\mathbf{1}$ is the $n$-dimensional vector with only ones. Thus, we have that

$$f_1([\boldsymbol{x},\boldsymbol{y},\boldsymbol{z},b]) = \begin{cases} [\boldsymbol{x},\boldsymbol{y},\boldsymbol{z}-\mathbf{1}] & \text{if } b = 0, \\ [\boldsymbol{x},\boldsymbol{y}-\mathbf{1},\boldsymbol{z}] & \text{if } b = 1. \end{cases}$$

Now, since $\boldsymbol{x}$, $\boldsymbol{y}$ and $\boldsymbol{z}$ are all binary vectors, it is easy to obtain that

$$\sigma(f_1([\boldsymbol{x},\boldsymbol{y},\boldsymbol{z},b])) = \begin{cases} [\boldsymbol{x},\boldsymbol{y},\mathbf{0}] & \text{if } b = 0, \\ [\boldsymbol{x},\mathbf{0},\boldsymbol{z}] & \text{if } b = 1. \end{cases}$$

Finally, consider the function $f_2 : \mathbb{Q}^{m+2n} \to \mathbb{Q}^{m+n}$ such that $f_2([\boldsymbol{x},\boldsymbol{y},\boldsymbol{z}]) = [\boldsymbol{x},\boldsymbol{y}+\boldsymbol{z}]$. Then we have that

$$f_2(\sigma(f_1([\boldsymbol{x},\boldsymbol{y},\boldsymbol{z},b]))) = \begin{cases} [\boldsymbol{x},\boldsymbol{y}] & \text{if } b = 0, \\ [\boldsymbol{x},\boldsymbol{z}] & \text{if } b = 1. \end{cases}$$

We note that $f_1(\cdot)$ and $f_2(\cdot)$ are affine transformations, and thus $f(\cdot) = f_2(\sigma(f_1(\cdot)))$ is a two-layer feed-forward network. This completes our proof. $\quad\square$

We can now continue with the proof of Lemma B.5. Recall that $\boldsymbol{z}_r^3$ is the following vector

$$\begin{aligned} \boldsymbol{z}_r^3 \;\; = \;\; [ \;\;\; & 0,\dots,0, \\ & [\![\, q^{(r+1)} \,]\!], [\![\, v^{(r)} \,]\!], m^{(r)}, m^{(r-1)}, \tfrac{c^{(r+1)}}{(r+1)}, \tfrac{c^{(r)}}{(r+1)}, \\ & [\![\, \alpha^{(r+1)} \,]\!], \beta^{(r+1)}, [\![\, v^{(\ell(r+1))} \,]\!], \ell(r+1), \\ & 1, (r+1), 1/(r+1), 1/(r+1)^2 \;\;\;\;\;\;\;\;\;\;\;\; ] \end{aligned}$$

Lets denote by $[\![\, m^{(r)} \,]\!]$ a vector such that

$$[\![\, m^{(r)} \,]\!] = \begin{cases} [1,0] & \text{if } m^{(r)} = 1, \\ [0,1] & \text{if } m^{(r)} = -1. \end{cases}$$

We first consider the function $f_1(\cdot)$ such that

$$f_1(\boldsymbol{z}_r^3) = [\, [\![\, q^{(r+1)} \,]\!], [\![\, m^{(r)} \,]\!], [\![\, \alpha^{(r+1)} \,]\!], (r+1)-\beta^{(r+1)}, [\![\, v^{(\ell(r+1))} \,]\!], [\![\, \# \,]\!], \ell(r+1)-(r-1) \,]$$

It is straightforward that $f_1(\cdot)$ can be implemented as an affine transformation. Just notice that $[\![\, \# \,]\!]$ is a fixed vector, $\ell(r+1)-(r-1) = \ell(r+1)-(r+1)+2$ and that $[\![\, m^{(r)} \,]\!] = [\tfrac{m^{(r)}}{2}, \tfrac{-m^{(r)}}{2}]+[\tfrac{1}{2},\tfrac{1}{2}]$. Moreover, all values in $f_1(\boldsymbol{z}_r^3)$ are binary values except for $(r+1)-\beta^{(r+1)}$ and $\ell(r+1)-(r-1)$. Thus, if we apply function $\sigma(\cdot)$ to $f_1(\boldsymbol{z}_r^3)$ we obtain

$$\sigma(f_1(\boldsymbol{z}_r^3)) = [\, [\![\, q^{(r+1)} \,]\!], [\![\, m^{(r)} \,]\!], [\![\, \alpha^{(r+1)} \,]\!], b_1, [\![\, v^{(\ell(r+1))} \,]\!], [\![\, \# \,]\!], b_2 \,]$$

where $b_1 = \sigma((r+1) - \beta^{(r+1)})$ and $b_2 = \sigma(\ell(r+1) - (r-1))$. By the definition of $\beta^{(r+1)}$ we know that $\beta^{(r+1)} = r + 1$ whenever $r + 1 \leq n$, and $\beta^{(r+1)} = n$ if $r + 1 > n$. Thus we have that

$$
b_1 = \begin{cases} 0 & \text{if } r+1 \leq n \\ 1 & \text{if } r+1 > n \end{cases}
$$

For the case of $b_2$, since $\ell(r+1) \leq r$ we have that $b_2 = 1$ if $\ell(r+1) = r$ and it is 0 otherwise, thus

$$
b_2 = \begin{cases} 1 & \text{if } \ell(r+1) = r \\ 0 & \text{if } \ell(r+1) \neq r \end{cases}
$$

Then, we can use the *if* function in Lemma B.6 to implement a function $f_2(\cdot)$ such that

$$
f_2(\sigma(f_1(\boldsymbol{z}_r^3))) = \begin{cases} [\, [\![\, q^{(r+1)}\, ]\!], [\![\, m^{(r)}\, ]\!], [\![\, \alpha^{(r+1)}\, ]\!], b_1, [\![\, v^{(\ell(r+1))}\, ]\!]\, ] & \text{if } b_2 = 0, \\ [\, [\![\, q^{(r+1)}\, ]\!], [\![\, m^{(r)}\, ]\!], [\![\, \alpha^{(r+1)}\, ]\!], b_1, [\![\, \#\, ]\!]\, ] & \text{if } b_2 = 1. \end{cases}
$$

We can use again the *if* function in Lemma B.6 to implement a function $f_3(\cdot)$ such that

$$
f_3(f_2(\sigma(f_1(\boldsymbol{z}_r^3)))) = \begin{cases} [\, [\![\, q^{(r+1)}\, ]\!], [\![\, m^{(r)}\, ]\!], [\![\, \alpha^{(r+1)}\, ]\!]\, ] & \text{if } b_2 = 0 \text{ and } b_1 = 0, \\ [\, [\![\, q^{(r+1)}\, ]\!], [\![\, m^{(r)}\, ]\!], [\![\, v^{(\ell(r+1))}\, ]\!]\, ] & \text{if } b_2 = 0 \text{ and } b_1 = 1, \\ [\, [\![\, q^{(r+1)}\, ]\!], [\![\, m^{(r)}\, ]\!], [\![\, \alpha^{(r+1)}\, ]\!]\, ] & \text{if } b_2 = 1 \text{ and } b_1 = 0, \\ [\, [\![\, q^{(r+1)}\, ]\!], [\![\, m^{(r)}\, ]\!], [\![\, \#\, ]\!]\, ] & \text{if } b_2 = 1 \text{ and } b_1 = 1, \end{cases}
$$

which can be rewritten as

$$
f_3(f_2(\sigma(f_1(\boldsymbol{z}_r^3)))) = \begin{cases} [\, [\![\, q^{(r+1)}\, ]\!], [\![\, m^{(r)}\, ]\!], [\![\, \alpha^{(r+1)}\, ]\!]\, ] & \text{if } r+1 \leq n, \\ [\, [\![\, q^{(r+1)}\, ]\!], [\![\, m^{(r)}\, ]\!], [\![\, \#\, ]\!]\, ] & \text{if } r+1 > n \text{ and } \ell(r+1) = r, \\ [\, [\![\, q^{(r+1)}\, ]\!], [\![\, m^{(r)}\, ]\!], [\![\, v^{(\ell(r+1))}\, ]\!]\, ] & \text{if } r+1 > n \text{ and } \ell(r+1) \neq r. \end{cases}
$$

From this, it is easy to prove that

$$
f_3(f_2(\sigma(f_1(\boldsymbol{z}_r^3)))) = [\, [\![\, q^{(r+1)}\, ]\!], [\![\, m^{(r)}\, ]\!], [\![\, s^{(r+1)}\, ]\!]\, ].
$$

This can be obtained from the following observation. If $r + 1 \leq n$ then $\alpha^{(r+1)} = s_{r+1} = s^{(r+1)}$. If $r + 1 > n$ and $\ell(r+1) = r$ then we know that $c^{(r+1)}$ is visited by $M$ for the first time at time $r + 1$ and it is *outside the original input*, which implies that $s^{(r+1)} = \#$. Finally, If $r + 1 > n$ and $\ell(r+1) \neq r$, then we know that $c^{(r+1)}$ has been visited before at time $\ell(r+1)$, and thus $s^{(r+1)} = v^{(\ell(r+1))}$.

The final piece of the proof is to just convert $[\![\, m^{(r)}\, ]\!]$ back to its value $m^{(r)}$, reorder the values and add 0's to obtain $\boldsymbol{y}_{r+1}$. We do all this with a final linear transformation $f_4(\cdot)$ such that

$$
\begin{aligned}
f_4(f_3(f_2(\sigma(f_1(\boldsymbol{z}_r^3))))) &= [\quad [\![\, q^{(r+1)}\, ]\!], [\![\, s^{(r+1)}\, ]\!], m^{(r)}, \\
&\qquad 0, \ldots, 0, \\
&\qquad 0, \ldots, 0, \\
&\qquad 0, \ldots, 0 \qquad\qquad ] \\
&= \boldsymbol{y}_{r+1}
\end{aligned}
$$

which completes our proof. $\qquad\square$

## C  PROOFS FOR SECTION 4

### C.1  PROOF OF THEOREM 4.1

The formulas of the Neural GPU in detail are as follows (with $\mathbf{S}^0$ the initial input tensor):

$$\begin{aligned}
\mathbf{U}^t &= U(\mathbf{S}^{t-1}) \\
\mathbf{R}^t &= R(\mathbf{S}^{t-1}) \\
\mathbf{S}^t &= \mathbf{U}^t \odot \mathbf{S}^{t-1} + (\mathbf{1} - \mathbf{U}) \odot F(\mathbf{R}^t \odot \mathbf{S}^{t-1})
\end{aligned}$$

With $U(\cdot)$, $R(\cdot)$, and $F(\cdot)$ defined as

$$\begin{aligned}
U(\mathbf{X}) &= f_U(\mathbf{K}^U * \mathbf{X} + \mathbf{B}^U) \\
R(\mathbf{X}) &= f_R(\mathbf{K}^R * \mathbf{X} + \mathbf{B}^R) \\
F(\mathbf{X}) &= f_F(\mathbf{K}^F * \mathbf{X} + \mathbf{B}^F)
\end{aligned}$$

Consider now an RNN encoder-decoder $N$ of dimension $d$ and composed of the equations

$$\begin{aligned}
\boldsymbol{h}_i &= \sigma(\boldsymbol{x}_i \boldsymbol{W} + \boldsymbol{h}_{i-1} \boldsymbol{V}) \\
\boldsymbol{g}_t &= \sigma(\boldsymbol{g}_{t-1} \boldsymbol{U})
\end{aligned}$$

with $\boldsymbol{h}_0 = \mathbf{0}$ and $\boldsymbol{g}_0 = \boldsymbol{h}_n$ where $n$ is the length of the input.

### CONSTRUCTING THE NEURAL GPU TO SIMULATE $N$

We construct a Neural GPU network NGPU that simulates $N$ as follows. Assume that the input of $N$ is $\boldsymbol{X} = (\boldsymbol{x}_1, \ldots, \boldsymbol{x}_n)$. Then we first construct the sequence $\boldsymbol{X}' = (\boldsymbol{x}'_1, \ldots, \boldsymbol{x}'_n)$ such that $\boldsymbol{x}'_i = [\boldsymbol{x}_i, \mathbf{0}, \mathbf{0}, 1, 1, 0]$ with $\mathbf{0} \in \mathbb{Q}^d$ the vector with all values as 0. Notice that $\boldsymbol{x}'_i \in \mathbb{Q}^{3d+3}$, moreover it is straightforward that if $\boldsymbol{x}_i$ was constructed from an embedding function $f : \Sigma \to \mathbb{Q}^d$ applied to a symbol $a \in \Sigma$, then $\boldsymbol{x}'_i$ can also be constructed with an embedding function $f' : \Sigma \to \mathbb{Q}^{3d+3}$ such that $f'(a) = [f(a), \mathbf{0}, \mathbf{0}, 1, 1, 0]$.

We consider an input tensor $\mathbf{S} \in \mathbb{Q}^{n \times 1 \times 3d+3}$ such that for every $i \in \{1, \ldots, n\}$ it holds that $\mathbf{S}_{i,1,:} = \boldsymbol{x}'_i = [\boldsymbol{x}_i, \mathbf{0}, \mathbf{0}, 1, 1, 0]$. Notice that since we picked $w = 1$, our tensor $\mathbf{S}$ is actually a $2D$ grid. Our proof shows that a bi-dimensional tensor is enough for simulating an RNN.

We now describe how to construct the kernel banks $\mathbf{K}^U$, $\mathbf{K}^R$ and $\mathbf{K}^F$ of shape $(2, 1, 3d+3, 3d+3)$. Notice that for each kernel $\mathbf{K}^X$ we essentially have to define two matrices $\mathbf{K}^X_{1,1,:,:}$ and $\mathbf{K}^X_{2,1,:,:}$ each one of dimension $(3d+3) \times (3d+3)$. We begin by defining every matrix in $\mathbf{K}^F$ as block matrices. When defining the matrices, all blank spaces are considered to be 0.

$$\mathbf{K}^F_{1,1,:,:} = \left[ \begin{array}{ccc|ccc|ccc}
\boldsymbol{V} & \boldsymbol{V} & & & \\
\hline
& & & & \\
\hline
& & & & \boldsymbol{F}_1
\end{array} \right]$$

$$\mathbf{K}^F_{2,1,:,:} = \left[ \begin{array}{ccc|ccc|ccc}
\boldsymbol{W} & \boldsymbol{W} & & & \\
\hline
& & & \boldsymbol{U} & \\
& & & \boldsymbol{U} & \\
\hline
& & & & \boldsymbol{F}_2
\end{array} \right]$$

where $\boldsymbol{F}_1$ and $\boldsymbol{F}_2$ are $3 \times 3$ matrices defined by

$$\boldsymbol{F}_1 = \left[ \begin{array}{ccc} 1 & 0 & 0 \\ 0 & 0 & 0 \\ 0 & 0 & 0 \end{array} \right] \qquad \boldsymbol{F}_2 = \left[ \begin{array}{ccc} 0 & 1 & 0 \\ 0 & 0 & 0 \\ 0 & 0 & 0 \end{array} \right]$$

Tensors $\mathbf{K}^U$ and $\mathbf{K}^R$ are considerable simpler. For the case of $\mathbf{K}^U$ we have

$$\mathbf{K}^U_{1,1,:,:} = \left[ \begin{array}{ccc|c} & & & \\ \hline & & & \\ \hline \boldsymbol{A} & \boldsymbol{A} & & \boldsymbol{U}_1 \end{array} \right]$$

$$\mathbf{K}^U_{2,1,:,:} = \left[ \begin{array}{ccc|c} & & & \\ \hline & & & \\ \hline & & \boldsymbol{A} & \boldsymbol{U}_2 \end{array} \right]$$

where $\boldsymbol{U}_1$ and $\boldsymbol{U}_2$ are $3 \times 3$ matrices defined by

$$\boldsymbol{U}_1 = \left[ \begin{array}{ccc} 1 & 1 & 0 \\ 0 & 0 & 0 \\ 0 & 0 & 0 \end{array} \right] \quad \boldsymbol{U}_2 = \left[ \begin{array}{ccc} 0 & 0 & 1 \\ 0 & 0 & 0 \\ 0 & 0 & 0 \end{array} \right]$$

and where $\boldsymbol{A}$ is the $3 \times d$ matrix defined by

$$\boldsymbol{A} = \left[ \begin{array}{cccc} 1 & 1 & \cdots & 1 \\ 0 & 0 & \cdots & 0 \\ 0 & 0 & \cdots & 0 \end{array} \right]$$

Finally, we define $\mathbf{K}^R$ as

$$\mathbf{K}^R_{1,1,:,:} = \left[ \begin{array}{ccc|c} & & & \\ \hline & & & \\ \hline & & & \end{array} \right]$$

$$\mathbf{K}^R_{2,1,:,:} = \left[ \begin{array}{ccc|c} & & & \\ \hline & & & \\ \hline \boldsymbol{A} & \boldsymbol{B} & & \boldsymbol{R}_2 \end{array} \right]$$

where $\boldsymbol{R}_2$ is the $3 \times 3$ matrix defined by

$$\boldsymbol{R}_2 = \left[ \begin{array}{ccc} 1 & 0 & 0 \\ 0 & 1 & 0 \\ 0 & 0 & 0 \end{array} \right]$$

and where $\boldsymbol{B}$ is the $3 \times d$ matrix defined by

$$\boldsymbol{B} = \left[ \begin{array}{cccc} 0 & 0 & \cdots & 0 \\ 1 & 1 & \cdots & 1 \\ 0 & 0 & \cdots & 0 \end{array} \right]$$

The bias tensors $\mathbf{B}^U$ and $\mathbf{B}^F$ are $\mathbf{0}$ (the tensor with all values 0). Finally, to construct tensor $\mathbf{B}^R$ we consider the matrix $\boldsymbol{D}$ of dimension $1 \times (3d + 3)$ such that

$$\boldsymbol{D} = \left[ \begin{array}{cccccc} \mathbf{0} & \mathbf{0} & 1 & 0 & 0 & 1 \end{array} \right]$$

Then we let $\mathbf{B}^R_{i,:,:} = \boldsymbol{D}$ for all $i$. Finally, we consider $f_U = f_R = f_F = \sigma$. The constructed Neural GPU is a uniform Neural GPU.

Before continuing with the proof we note that for every kernel $\mathbf{K}^X$ and tensor $\mathbf{S}$ we have that

$$(\mathbf{K}^X * \mathbf{S})_{i,1,:} = \mathbf{S}_{i-1,1,:}\mathbf{K}^X_{1,1,:,:} + \mathbf{S}_{i,1,:}\mathbf{K}^X_{2,1,:,:}$$

CORRECTNESS OF THE CONSTRUCTION

We now prove that the following properties hold for every $t \geq 0$:

$$\mathbf{S}^t_{i,1,:} = \begin{cases} [\mathbf{0}, \mathbf{0}, \boldsymbol{\alpha}^i_{t-i}, 0, 0, 0] & \text{for } i < t \\ [\boldsymbol{h}_i, \boldsymbol{h}_i, \mathbf{0}, 0, 1, 0] & \text{for } i = t \\ [\boldsymbol{x}_i, \mathbf{0}, \mathbf{0}, 1, 1, 0] & \text{for } i > t \end{cases} \tag{27}$$

where $\boldsymbol{\alpha}_j^k$ is given by the recurrence $\boldsymbol{\alpha}_0^k = \boldsymbol{h}_k$ and $\boldsymbol{\alpha}_j^k = \sigma(\boldsymbol{\alpha}_{j-1}^k \boldsymbol{U})$. Notice that $\boldsymbol{g}_j = \boldsymbol{\alpha}_j^n$. That is, we are going to prove that our construction actually simulates $N$. By (27) one can see that the intuition in our construction is to use the first $d$ components to simulate the encoder part, the next $d$ components to communicate data between the encoder and decoder simulation, and the next $d$ components to simulate the decoder part. The last three components are needed as gadgets for the gates to actually simulate a sequential read of the input, and to ensure that the hidden state of the encoder and decoder are updated properly.

We prove the above statement by induction in $t$. First notice that the property trivially holds for $\mathbf{S}^0$. Now assume that this holds for $t-1$ and lets prove it for $t$. We know that $\mathbf{U}^t$ is computed as

$$\mathbf{U}^t = \sigma(\mathbf{K}^U * \mathbf{S}^{t-1} + \mathbf{B}^U) = \sigma(\mathbf{K}^U * \mathbf{S}^{t-1})$$

Thus we have that:

$$\begin{aligned}
\mathbf{U}_{i,1,:}^t &= \sigma((\mathbf{K}^U * \mathbf{S}^{t-1})_{i,1,:}) \\
&= \sigma(\mathbf{S}_{i-1,1,:}^{t-1} \mathbf{K}_{1,1,:,:}^U + \mathbf{S}_{i,1,:}^{t-1} \mathbf{K}_{2,1,:,:}^U)
\end{aligned}$$

By the induction hypothesis we have

$$\mathbf{S}_{i,1,:}^{t-1} = \begin{cases} [\mathbf{0}, \mathbf{0}, \boldsymbol{\alpha}_{t-1-i}^i, 0, 0, 0] & \text{for } i < t-1 \\ [\boldsymbol{h}_i, \boldsymbol{h}_i, \mathbf{0}, 0, 1, 0] & \text{for } i = t-1 \\ [\boldsymbol{x}_i, \mathbf{0}, \mathbf{0}, 1, 1, 0] & \text{for } i > t-1 \end{cases} \tag{28}$$

Now, notice that $\mathbf{K}_{1,1,:,:}^U$ and $\mathbf{K}_{2,1,:,:}^U$ are not zero only in its three last rows, thus we can focus on the three last components of the vectors in $\mathbf{S}^{t-1}$, and then we can compute $\mathbf{U}_{i,1,:}^t$ as

$$\begin{aligned}
\mathbf{U}_{i,1,:}^t &= \begin{cases} \sigma([\text{-},\text{-},\text{-},0,0,0]\mathbf{K}_{1,1,:,:}^U + [\text{-},\text{-},\text{-},0,0,0]\mathbf{K}_{2,1,:,:}^U) & \text{for } i < t-1 \\ \sigma([\text{-},\text{-},\text{-},0,0,0]\mathbf{K}_{1,1,:,:}^U + [\text{-},\text{-},\text{-},0,1,0]\mathbf{K}_{2,1,:,:}^U) & \text{for } i = t-1 \\ \sigma([\text{-},\text{-},\text{-},0,1,0]\mathbf{K}_{1,1,:,:}^U + [\text{-},\text{-},\text{-},1,1,0]\mathbf{K}_{2,1,:,:}^U) & \text{for } i = t \\ \sigma([\text{-},\text{-},\text{-},1,1,0]\mathbf{K}_{1,1,:,:}^U + [\text{-},\text{-},\text{-},1,1,0]\mathbf{K}_{2,1,:,:}^U) & \text{for } i > t \end{cases} \\
&= \begin{cases} \sigma([\mathbf{0},\mathbf{0},\mathbf{0},0,0,0]) & \text{for } i < t-1 \\ \sigma([\mathbf{0},\mathbf{0},\mathbf{0},0,0,0]) & \text{for } i = t-1 \\ \sigma([\mathbf{0},\mathbf{0},\mathbf{1},0,0,1]) & \text{for } i = t \\ \sigma([\mathbf{1},\mathbf{1},\mathbf{1},1,1,1]) & \text{for } i > t \end{cases} \\
&= \begin{cases} [\mathbf{0},\mathbf{0},\mathbf{0},0,0,0] & \text{for } i < t \\ [\mathbf{0},\mathbf{0},\mathbf{1},0,0,1] & \text{for } i = t \\ [\mathbf{1},\mathbf{1},\mathbf{1},1,1,1] & \text{for } i > t \end{cases}
\end{aligned}$$

Now, for $\mathbf{R}^t$ we have

$$\mathbf{R}^t = \sigma(\mathbf{K}^R * \mathbf{S}^{t-1} + \mathbf{B}^R)$$

and thus for $\mathbf{R}_{i,1,:}^t$ we have

$$\begin{aligned}
\mathbf{R}_{i,1,:}^t &= \sigma((\mathbf{K}^R * \mathbf{S}^{t-1})_{i,1,:} + \mathbf{B}_{i,1,:}^R) \\
&= \sigma(\mathbf{S}_{i-1,1,:}^{t-1} \mathbf{K}_{1,1,:,:}^R + \mathbf{S}_{i,1,:}^{t-1} \mathbf{K}_{2,1,:,:}^R + \mathbf{B}_{i,1,:}^R) \\
&= \sigma(\mathbf{S}_{i,1,:}^{t-1} \mathbf{K}_{2,1,:,:}^R + \mathbf{B}_{i,1,:}^R) \\
&= \sigma(\mathbf{S}_{i,1,:}^{t-1} \mathbf{K}_{2,1,:,:}^R + [\mathbf{0},\mathbf{0},\mathbf{1},0,0,1])
\end{aligned}$$

where we deleted the term with $\mathbf{K}^R_{1,1,:,:}$ since it is the null matrix. Then by using the definition of $\mathbf{S}^{t-1}_{i,1,:}$ above (Equation (28)) we have

$$
\mathbf{R}^t_{i,1,:} = 
\begin{cases}
\sigma([-,-,-,0,0,0]\mathbf{K}^R_{2,1,:,:} + [\mathbf{0},\mathbf{0},\mathbf{1},0,0,1]) & \text{for } i < t-1 \\
\sigma([-,-,-,0,1,0]\mathbf{K}^R_{2,1,:,:} + [\mathbf{0},\mathbf{0},\mathbf{1},0,0,1]) & \text{for } i = t-1 \\
\sigma([-,-,-,1,1,0]\mathbf{K}^R_{2,1,:,:} + [\mathbf{0},\mathbf{0},\mathbf{1},0,0,1]) & \text{for } i = t \\
\sigma([-,-,-,1,1,0]\mathbf{K}^R_{2,1,:,:} + [\mathbf{0},\mathbf{0},\mathbf{1},0,0,1]) & \text{for } i > t
\end{cases}
$$
$$
= 
\begin{cases}
\sigma([\mathbf{0},\mathbf{0},\mathbf{1},0,0,1]) & \text{for } i < t-1 \\
\sigma([\mathbf{0},\mathbf{1},\mathbf{1},0,1,1]) & \text{for } i = t-1 \\
\sigma([\mathbf{1},\mathbf{1},\mathbf{1},1,1,1]) & \text{for } i = t \\
\sigma([\mathbf{1},\mathbf{1},\mathbf{1},1,1,1]) & \text{for } i > t
\end{cases}
$$
$$
= 
\begin{cases}
[\mathbf{0},\mathbf{0},\mathbf{1},0,0,1] & \text{for } i < t-1 \\
[\mathbf{0},\mathbf{1},\mathbf{1},0,1,1] & \text{for } i = t-1 \\
[\mathbf{1},\mathbf{1},\mathbf{1},1,1,1] & \text{for } i \geq t
\end{cases}
$$

We can now compute $\mathbf{S}^t_{i,1,:}$. By the definition of $\mathbf{S}^t$ we have

$$
\mathbf{S}^t_{i,1,:} = \mathbf{U}^t_{i,1,:} \odot \mathbf{S}^{t-1}_{i,1,:} + (\mathbf{1}_{i,1,:} - \mathbf{U}_{i,1,:}) \odot \sigma((\mathbf{K}^F * (\mathbf{R}^t \odot \mathbf{S}^{t-1}))_{i,1,:})
$$

where we dropped $\mathbf{B}^F$ that has only zeros. First, by using what we already computed for $\mathbf{U}^t_{i,1,:}$ we have that

$$
\mathbf{S}^t_{i,1,:} = 
\begin{cases}
\sigma((\mathbf{K}^F * (\mathbf{R}^t \odot \mathbf{S}^{t-1}))_{i,1,:}) & \text{for } i < t \\
[\mathbf{0},\mathbf{0},\mathbf{1},0,0,1] \odot \mathbf{S}^{t-1}_{i,1,:} + [\mathbf{1},\mathbf{1},\mathbf{0},1,1,0] \odot \sigma((\mathbf{K}^F * (\mathbf{R}^t \odot \mathbf{S}^{t-1}))_{i,1,:}) & \text{for } i = t \\
\mathbf{S}^{t-1}_{i,1,:} & \text{for } i > t
\end{cases}
$$

When $i = t$ we have that $\mathbf{S}^{t-1}_{i,1,:} = [\boldsymbol{x}_i,\mathbf{0},\mathbf{0},1,0,0]$ (Equation (28)), thus $[\mathbf{0},\mathbf{0},\mathbf{1},0,0,1] \odot \mathbf{S}^{t-1}_{i,1,:} = [\mathbf{0},\mathbf{0},\mathbf{0},0,0,0]$. Then

$$
\mathbf{S}^t_{i,1,:} = 
\begin{cases}
\sigma((\mathbf{K}^F * (\mathbf{R}^t \odot \mathbf{S}^{t-1}))_{i,1,:}) & \text{for } i < t \\
[\mathbf{1},\mathbf{1},\mathbf{0},1,1,0] \odot \sigma((\mathbf{K}^F * (\mathbf{R}^t \odot \mathbf{S}^{t-1}))_{i,1,:}) & \text{for } i = t \\
[\boldsymbol{x}_i,\mathbf{0},\mathbf{0},1,1,0] & \text{for } i > t
\end{cases}
\tag{29}
$$

We are almost done with the inductive step, we only need to compute $\sigma((\mathbf{K}^F * (\mathbf{R}^t \odot \mathbf{S}^{t-1}))_{i,1,:})$. Given what we have for $\mathbf{R}^t$ and $\mathbf{S}^{t-1}$ we have that $\mathbf{R}^t \odot \mathbf{S}^{t-1}$ is

$$
(\mathbf{R}^t \odot \mathbf{S}^{t-1})_{i,1,:} = 
\begin{cases}
[\mathbf{0},\mathbf{0},\mathbf{1},0,0,1] \odot [\mathbf{0},\mathbf{0},\boldsymbol{\alpha}^i_{t-1-i},0,0,0] & \text{for } i < t-1 \\
[\mathbf{0},\mathbf{1},\mathbf{1},0,1,1] \odot [\boldsymbol{h}_i,\boldsymbol{h}_i,\mathbf{0},0,1,0] & \text{for } i = t-1 \\
[\mathbf{1},\mathbf{1},\mathbf{1},1,1,1] \odot [\boldsymbol{x}_i,\mathbf{0},\mathbf{0},1,1,0] & \text{for } i \geq t
\end{cases}
$$
$$
= 
\begin{cases}
[\mathbf{0},\mathbf{0},\boldsymbol{\alpha}^i_{t-1-i},0,0,0] & \text{for } i < t-1 \\
[\mathbf{0},\boldsymbol{h}_i,\mathbf{0},0,1,0] & \text{for } i = t-1 \\
[\boldsymbol{x}_i,\mathbf{0},\mathbf{0},1,1,0] & \text{for } i \geq t
\end{cases}
$$

Lets $\mathbf{T}^t = \sigma(\mathbf{K}^F * (\mathbf{R}^t \odot \mathbf{S}^{t-1}))$. Notice that from Equation (29) we actually need to know the values in $\mathbf{T}^t_{i,1,:}$ only for $i \leq t$. Now we have that

$$
\begin{aligned}
\mathbf{T}^t_{i,1,:} &= \sigma((\mathbf{K}^F * (\mathbf{R}^t \odot \mathbf{S}^{t-1}))_{i,1,:}) \\
&= \sigma((\mathbf{R}^t \odot \mathbf{S}^{t-1}))_{i,1,:}\mathbf{K}^F_{1,1,:,:} + (\mathbf{R}^t \odot \mathbf{S}^{t-1}))_{i,1,:}\mathbf{K}^F_{2,1,:,:}) \\
&= \begin{cases} \sigma([\mathbf{0},\mathbf{0},\boldsymbol{\alpha}^{i-1}_{t-i},0,0,0]\mathbf{K}^F_{1,1,:,:} + [\mathbf{0},\mathbf{0},\boldsymbol{\alpha}^i_{t-1-i},0,0,0]\mathbf{K}^F_{2,1,:,:}) & \text{for } i < t-1 \\ \sigma([\mathbf{0},\mathbf{0},\boldsymbol{\alpha}^{i-1}_{t-i},0,0,0]\mathbf{K}^F_{1,1,:,:} + [\mathbf{0},\boldsymbol{h}_i,\mathbf{0},0,1,0]\mathbf{K}^F_{2,1,:,:}) & \text{for } i = t-1 \\ \sigma([\mathbf{0},\boldsymbol{h}_{i-1},\mathbf{0},0,1,0]\mathbf{K}^F_{1,1,:,:} + [\boldsymbol{x}_i,\mathbf{0},\mathbf{0},1,1,0]\mathbf{K}^F_{2,1,:,:}) & \text{for } i = t \end{cases} \\
&= \begin{cases} \sigma([\mathbf{0},\mathbf{0},\boldsymbol{\alpha}^i_{t-1-i}\boldsymbol{U},0,0,0]) & \text{for } i < t-1 \\ \sigma([\mathbf{0},\mathbf{0},\boldsymbol{h}_i\boldsymbol{U},0,0,0]) & \text{for } i = t-1 \\ \sigma([\boldsymbol{x}_i\boldsymbol{W} + \boldsymbol{h}_{i-1}\boldsymbol{V}, \boldsymbol{x}_i\boldsymbol{W} + \boldsymbol{h}_{i-1}\boldsymbol{V},\mathbf{0},0,1,0]) & \text{for } i = t \end{cases} \\
&= \begin{cases} [\mathbf{0},\mathbf{0},\boldsymbol{\alpha}^i_{t-i},0,0,0] & \text{for } i < t-1 \\ [\mathbf{0},\mathbf{0},\boldsymbol{\alpha}^i_1,0,0,0] & \text{for } i = t-1 \\ [\boldsymbol{h}_i,\boldsymbol{h}_i,\mathbf{0},0,1,0] & \text{for } i = t \end{cases} \\
&= \begin{cases} [\mathbf{0},\mathbf{0},\boldsymbol{\alpha}^i_{t-i},0,0,0] & \text{for } i < t \\ [\boldsymbol{h}_i,\boldsymbol{h}_i,\mathbf{0},0,1,0] & \text{for } i = t \end{cases}
\end{aligned}
$$

Putting the value of $\mathbf{T}^t_{i,1,:}$ in Equation (29) we obtain

$$
\begin{aligned}
\mathbf{S}^t_{i,1,:} &= \begin{cases} \mathbf{T}^t_{i,1,:} & \text{for } i < t \\ [\mathbf{1},\mathbf{1},\mathbf{0},1,1,0] \odot \mathbf{T}^t_{i,1,:} & \text{for } i = t \\ [\boldsymbol{x}_i,\mathbf{0},\mathbf{0},1,1,0] & \text{for } i > t \end{cases} \\
&= \begin{cases} [\mathbf{0},\mathbf{0},\boldsymbol{\alpha}^i_{t-i},0,0,0] & \text{for } i < t \\ [\boldsymbol{h}_i,\boldsymbol{h}_i,\mathbf{0},0,1,0] & \text{for } i = t \\ [\boldsymbol{x}_i,\mathbf{0},\mathbf{0},1,1,0] & \text{for } i > t \end{cases}
\end{aligned}
$$

which is exactly what we needed to prove (Equation (27)).

Now, lets focus on $\mathbf{S}^{n+t}_{n,1,:}$ for $t \geq 1$. By what we have just proved, we obtain that

$$
\mathbf{S}^{n+t}_{n,1,:} = [\mathbf{0},\mathbf{0},\boldsymbol{\alpha}^n_t,0,0,0] = [\mathbf{0},\mathbf{0},\boldsymbol{g}_t,0,0,0]
$$

which is the decoder part of the RNN $N$. Thus, we can simulate the complete network $N$ with a Neural GPU.

### C.2 PROOF OF PROPOSITION 4.2

We first prove the following claim: Assume that $\mathbf{S} \in \mathbb{Q}^{h \times w \times d}$ is a tensor that satisfies the following property: there exists a $p \geq 1$ that divides $h$ and such that, for every $i \in \{1, 2, \ldots, h - p\}$ it holds that

$$
\mathbf{S}_{i,:,:} = \mathbf{S}_{i+p,:,:}.
$$

Given that we will be considering circular convolutions, then we have that for every $\ell \leq 0$ we have that $\mathbf{S}_{\ell,j,:} = \mathbf{S}_{h+\ell,j,:}$ and for every $\ell > h$ we have that $\mathbf{S}_{\ell,j,:} = \mathbf{S}_{h-\ell,j,:}$. With this we have that for every $i \in \mathbb{N}$ it holds that $\mathbf{S}_{i,:,:} = \mathbf{S}_{i+p,:,:}$ that is, we do not need to restrict to values in $i \in \{1, 2, \ldots, h - p\}$.

Now lets $\mathbf{K}$ be an arbitrary kernel bank of shape $(k_H, k_W, d, d)$. Let $\mathbf{T} = \mathbf{K} \circledast \mathbf{S}$ where $\circledast$ denotes the circular convolution. We prove next that

$$
\mathbf{T}_{i,:,:} = \mathbf{T}_{i+p,:,:}
$$

for every $i$. This is a simple fact that follows from the way in which the convolution is defined. We use the notation in the body of the paper for $\mathbf{T} = \mathbf{K} \circledast \mathbf{S}$, that is, we denote by $\boldsymbol{s}_{ij}$ the vector $\mathbf{S}_{i,j,:}$ and $\boldsymbol{K}_{ij}$ the matrix $\mathbf{K}_{i,j,:,:}$. Notice that $\boldsymbol{s}_{ij} = \boldsymbol{s}_{i+p,j}$ for every $i \in \mathbb{N}$. Now the circular convolution is

$$
\mathbf{T}_{i,j,:} = (\mathbf{K} \circledast \mathbf{S})_{i,j,:} = \sum_{u=1}^{k_H} \sum_{v=1}^{k_W} \boldsymbol{s}_{i+\Delta_1(u),j+\Delta_2(v)} \boldsymbol{K}_{uv}
$$

where $\Delta_1(u) = u - \lfloor k_H/2 \rfloor - 1$ and $\Delta_2(v) = v - \lfloor k_W/2 \rfloor - 1$. Then, given that $s_{ij} = s_{i+p,j}$ for every $i \in \mathbb{N}$ we have that

$$\mathbf{T}_{i,j,:} = (\mathbf{K} \circledast \mathbf{S})_{i,j,:} = \sum_{u=1}^{k_H} \sum_{v=1}^{k_W} s_{i+\Delta_1(u)+p, j+\Delta_2(v)} K_{uv} = (\mathbf{K} \circledast \mathbf{S})_{i+p,j,:} = \mathbf{T}_{i+p,j,:}$$

and then, $\mathbf{T}_{i,:,:} = \mathbf{T}_{i+p,:,:}$.

Consider now an arbitrary uniform Neural GPU that processes tensor $\mathbf{S}$ above, and assume that $\mathbf{S}^1, \mathbf{S}^2, \ldots, \mathbf{S}^r$ is the sequence produced by it. Next we prove that for every $t$ and for every $i$ it holds that $\mathbf{S}^t_{i,:,:} = \mathbf{S}^t_{i+p,:,:}$. We prove it by induction in $t$. For the case $\mathbf{S}^0$ it holds by definition. Thus assume that $\mathbf{S}^{t-1}$ satisfies the property. Let

$$\begin{aligned}
\mathbf{U}^t &= f_U(\mathbf{K}^U * \mathbf{S}^{t-1} + \mathbf{B}^U) \\
\mathbf{R}^t &= f_R(\mathbf{K}^R * \mathbf{S}^{t-1} + \mathbf{B}^R) \\
\mathbf{S}^t &= \mathbf{U}^t \odot \mathbf{S}^{t-1} + (\mathbf{1} - \mathbf{U}) \odot f_F(\mathbf{K}^F * (\mathbf{R}^t \odot \mathbf{S}^{t-1}) + \mathbf{B}^F)
\end{aligned}$$

Since we are considering uniform Neural GPUs, we know that there exist three matrices $\boldsymbol{B}^U$, $\boldsymbol{B}^R$ and $\boldsymbol{B}^F$ such that for every $i$ it holds that $\mathbf{B}^U_{i,:,:} = \boldsymbol{B}^U$, $\mathbf{B}^R_{i,:,:} = \boldsymbol{B}^R$, and $\mathbf{B}^F_{i,:,:} = \boldsymbol{B}^F$. It is easy to prove that $\mathbf{U}^t_{i,:,:} = \mathbf{U}^t_{i+p,:,:}$. First note that by inductive hypothesis, we have that $\mathbf{S}^{t-1}_{i,:,:} = \mathbf{S}^{t-1}_{i+p,:,:}$ and thus by the property proved above we have that $(\mathbf{K}^U * \mathbf{S}^{t-1})_{i,:,:} = (\mathbf{K}^U * \mathbf{S}^{t-1})_{i+p,:,:}$. Thus we have that

$$\mathbf{U}^t_{i,:,:} = f_U((\mathbf{K}^U * \mathbf{S}^{t-1})_{i,:,:} + \boldsymbol{B}^U) = f_U((\mathbf{K}^U * \mathbf{S}^{t-1})_{i+p,:,:} + \boldsymbol{B}^U) = \mathbf{U}^t_{i+p,:,:}$$

With a similar argument we can prove that $\mathbf{R}^t_{i,:,:} = \mathbf{R}^t_{i+p,:,:}$. Moreover, notice that $(\mathbf{R}^t \odot \mathbf{S}^{t-1})_{i,:,:} = (\mathbf{R}^t \odot \mathbf{S}^{t-1})_{i+p,:,:}$, and thus $(\mathbf{K}^F * (\mathbf{R}^t \odot \mathbf{S}^{t-1}))_{i,:,:} = (\mathbf{K}^F * (\mathbf{R}^t \odot \mathbf{S}^{t-1}))_{i+p,:,:}$. With all this we finally have that

$$\begin{aligned}
\mathbf{S}^t_{i,:,:} &= \mathbf{U}^t_{i,:,:} \odot \mathbf{S}^{t-1}_{i,:,:} + (\mathbf{1}_{i,:,:} - \mathbf{U}_{i,:,:}) \odot f_F((\mathbf{K}^F * (\mathbf{R}^t \odot \mathbf{S}^{t-1}))_{i,:,:} + \boldsymbol{B}^F) \\
&= \mathbf{U}^t_{i+p,:,:} \odot \mathbf{S}^{t-1}_{i+p,:,:} + (\mathbf{1}_{i+p,:,:} - \mathbf{U}_{i+p,:,:}) \odot f_F((\mathbf{K}^F * (\mathbf{R}^t \odot \mathbf{S}^{t-1}))_{i+p,:,:} + \boldsymbol{B}^F) \\
&= \mathbf{S}^t_{i+p,:,:}
\end{aligned}$$

This completes the first part of the proof.

We have shown that if the input of a uniform neural GPU is periodic, then the output is also periodic. We make a final observation. Let $N$ be a uniform Neural GPU, and $\mathbf{S} \in \mathbb{Q}^{kp \times w \times d}$ be a tensor such that $\mathbf{S}_{i,:,:} = \mathbf{S}_{i+p,:,:}$ for every $i$. Moreover, let $\mathbf{T} \in \mathbb{Q}^{k'p \times w \times d}$ be a tensor such that $\mathbf{T}_{i,:,:} = \mathbf{T}_{i+p,:,:}$ for every $i$, and assume that $\mathbf{S}_{1:p,:,:} = \mathbf{T}_{1:p,:,:}$. Lets $\mathbf{S}^1, \mathbf{S}^2, \ldots$ and $\mathbf{T}^1, \mathbf{T}^2, \ldots$ be the sequences produced by $N$. Then with a similar argument as above it is easy to prove that for every $t$ it holds that $\mathbf{S}^t_{1:p,:,:} = \mathbf{T}^t_{1:p,:,:}$.

From this it is easy to prove that uniform Neural GPUs will no be able to recognize the length of periodic inputs. Thus assume that there is a language recognizer $A$ defined by of a uniform neural GPU $N$ such that $L(A)$ contains all strings of even length. Assume that $u$ is an arbitrary string in $\Sigma$ such that $|u| = p$ with $p$ an odd number, and let $w = uu$ and $w' = uuu$. Notice that $|w| = 2p$ and thus $w \in L(A)$, but $|w'| = 3p$ and thus $w' \notin L(A)$.

Let $f : \Sigma \to \mathbb{Q}^d$ and let $\boldsymbol{X} = f(w) = (\boldsymbol{x}_1, \boldsymbol{x}_2, \ldots, \boldsymbol{x}_{2p})$ and $\boldsymbol{X}' = f(w') = (\boldsymbol{x}'_1, \boldsymbol{x}'_2, \ldots, \boldsymbol{x}'_{3p})$. Consider now the tensor $\mathbf{S} \in \mathbb{Q}^{2p \times w \times d}$ such that $\mathbf{S}_{i,1,:} = \boldsymbol{x}_i$ for $i \in \{1, \ldots, 2p\}$, thus $\mathbf{S}_{i,:,:} = \mathbf{S}_{i+p,:,:}$. Similarly, consider $\mathbf{T} \in \mathbb{Q}^{3p \times w \times d}$ such that such that $\mathbf{T}_{i,1,:} = \boldsymbol{x}'_i$ for $i \in \{1, \ldots, 3p\}$, and thus $\mathbf{T}_{i,:,:} = \mathbf{T}_{i+p,:,:}$. Notice that $\mathbf{S}_{1:p,:,:} = \mathbf{T}_{1:p,:,:}$ then by the property above we have that for every $t$ it holds that $\mathbf{S}^t_{1:p,:,:} = \mathbf{T}^t_{1:p,:,:}$. In particular, we have $\mathbf{S}^t_{p,:,:} = \mathbf{T}^t_{p,:,:}$. We also know that $\mathbf{S}^t_{p,:,:} = \mathbf{S}^t_{2p,:,:}$ and that $\mathbf{T}^t_{p,:,:} = \mathbf{T}^t_{2p,:,:} = \mathbf{T}^t_{3p,:,:}$. Thus we have that for every $t$ it holds that $\mathbf{S}^t_{2p,1,:} = \mathbf{T}^t_{3p,1,:}$. From this we conclude that the outputs of $N$ for both inputs $\boldsymbol{X}$ and $\boldsymbol{X}'$ are the same, and thus if $A$ accepts $w$ then $A$ accepts $w'$ which is a contradiction.

