# OpenReview forum: "On the Turing Completeness of Modern Neural Network Architectures"
_ICLR.cc/2019/Conference_

### Official Review · AnonReviewer2 · 2018-11-02
**Clear and well written, some questions about how these results map to training the real models.**

**Rating:** 7
**Confidence:** 2

**Review:**

This paper seeks to answer the question of whether models which process sequences, but are not strictly classical RNNs, are Turing complete.

The authors present proofs that both the Transformer and Neural GPU are turing complete, under certain conditions. I do not consider myself qualified to properly verify the proof but it seems to be presented clearly. The authors note that the conditions involved are not how these models are used in the real world. Given the complex construction required for this more theoretically based proof, it seems reasonable that this should be published now, rather than waiting until the further work discussed in the final section is completed.

I have a number of questions where if a brief answer is possible, this would enhance the manuscript. The main question is, of the simplifications and approximations used for the proof, how much does that take the model away from what is used in practice? For example, the assumption of the piecewise linear sigmoid seems like a quite big change, as there are large regions of the space which now have zero gradients. If you run a real implementation of these models, with the normal sigmoid replaced by this one, does training still work? If not, what are the implications for the proof?

The rational numbers assumption is interesting - again I wonder how this would affect the model in reality, obviously all floating points on a computer represent rationals, but it would be interesting to get a better understanding on how the lack of infinite precision rationals on real hardware affects the main results.

Does the proof rely on the input and output dimensionality being the same? Eg in the preliminaries, x_i and y_i are both d-dimensional - could this be changed?

Overall this paper is novel and interesting, I have to give a slightly low confidence score because I'm unfamiliar with a lot of the background here (eg the Siegelamnn & Sontag work). The paper does seem concise and well written.

typos and minor points:

Circular convolution definition only appears to define the values directly adjacent to the border, would it be more appropriate to define S_{h+n, :, :} = S{n, :, :}?

paragraph above equation 5, 'vectores' -> 'vectors'

---

### Official Review · AnonReviewer3 · 2018-11-03
**Better fit for a journal**

**Rating:** 7
**Confidence:** 2

**Review:**

This paper presents interesting theoretical results on Turing completeness of the Transformer and Neural GPU architectures, as modern architectures based on attention and convolutions, under particular assumptions. The basis of proofs in the paper relies on Turing completeness of the seq2seq architecture, which is Turing complete since it contains Turing complete RNNs. Turing completeness of the Transformer and the Neural GPU is proven by showing they can simulate seq2seq architecture.

The Transformer, using additive hard attention and residual connections, is Turing complete in the case when positional encoding is used. Otherwise, if no positional encoding is used, the model is order-invariant which makes it not Turing complete.

A version of the Neural GPU, dubbed Uniform Neural GPU is proven to be Turing complete. Moreover, the presented theoretical results are backed by a recent publication by Karlis and Liepins. Interestingly, Neural GPUs using circular convolutions are not Turing complete, while the ones using zero padding are.

The repercussion of the paper for similar architectures is the not just in the theoretical section but also in a set of discoveries of practical importance, like the importance of the use of residual connections, positional coding in Transformers, and zero padding in Neural GPUs.

Albeit the paper presents an original and significant theoretical progress and is well written, it is not fit for ICLR, primarily as the paper is impossible to review and verify without a thorough perusal and analysis of the appendix. Although the results and the proof sketches fit the body of the paper, the necessity of verifying proofs makes this paper 23 pages long and makes it a better fit for a journal and not a conference.

---

> ### Public Comment · (anonymous) · 2018-12-12
> **Result in this paper is potentially wrong...**
>
> It's observed in [1] that Transformer is not universal. Also, the proofs in this paper are very technical without any intuitive explanation. The results seem very questionable.
>
> [1] https://openreview.net/forum?id=HyzdRiR9Y7&noteId=HyxfZDmCk4

---

### Official Review · AnonReviewer1 · 2018-11-06
**Potentially interesting results, very dense and confusing writing**

**Rating:** 6
**Confidence:** 2

**Review:**

The paper shows Turing completeness of two modern neural architectures, the Transformer and the Neural GPU. The paper is technically very heavy and gives very little insight and intuition behind the results. Right after surveying the previous work the paper starts stacking definitions and theorems without much explanations.

While technical results are potentially quite strong I believe a major revision to the paper might be necessary in order to clarify the ideas. I would even suggest to split the paper into two, one about each architecture as in the current form it is quite long and difficult to follow.

Results are claimed to hold without access to external memory, relying just on the network itself to represent the intermediate results of the computation. I am a bit confused by this statement -- what if the problem at hand is, say EXPSPACE-complete? Then the network would have to be of exponential size (or more generally of arbitrary size which is independent of the input). In this case the claim about not using external memory seems to be kind of vacuous as the network itself has unbounded size. The whole point of Turing-completeness is that the program size is independent of the input size so there seems to be some confusion here.

---

> ### Comment · Area_Chair1 · 2018-11-30
> **Please consider rebuttal**
>
> Dear Reviewer 1,
>
> Thanks again for you review. As you can see, the authors have written a detailed rebuttal to you and the other reviewers in separate post. Please take the time to consider it, and the other reviews, and respond if needed. I would appreciate it if you can review your own assessment of the paper, and, if you decide to stand by your score, present a short explanation of why you think the paper still falls short in light of the comments made by the authors.

---

### Public Comment · (anonymous) · 2018-10-02
**You assume infinite precision, right?**

Do I understand correctly that your results only hold for weights in Q, meanining unbounded precision? I suppose it's not true with limited precision, like float32, or am I misunderstanding?

---

> ### Author Response · Authors · 2018-10-02
> **We need unbounded precision for internal representations**
>
> Our proofs are based on having unbounded precision for internal representations (neuron values). For weights one can prove that fixed precision (actually very small) is enough.
>
> Our results say nothing about the computational power when fixed precision (like float32) is assumed for internal representations. We actually state the fixed-precision case as an interesting topic for future research.

---

### Author Response · Authors · 2018-11-13
**Rebuttal**

We thank the reviewers for their comments. We first make some general comments and then answer directly to each reviewer.

All reviewers appear to agree that our technical results on the Turing completeness of the Transformer and the Neural GPU are potentially interesting/important. In two reviews, there is however a general question about the fit of our results for ICLR. One reviewer advised to go directly to a journal. We did consider submitting to a journal or to a theoretical conference, but we felt it important to discuss the computational properties of the Transformer and the Neural GPU directly with the community involved in their design, implementation, and practical use. We felt that submitting to ICLR would generate more impact noting that the Neural GPU was initially proposed at ICLR2016, and the Transformer architecture (proposed at NIPS2017) is used in several ICLR2018 papers (and also now ICLR2019 submissions).

We also observe that there is a need for more theoretical foundations with regards to the computational power of modern NN architectures at ICLR, and in particular about the two architectures that we study. Consider for example the following ICLR2019 submission: “Universal Transformers” (https://openreview.net/forum?id=HyzdRiR9Y7&noteId=HyzdRiR9Y7). Universal Transformers are networks that combine the parallelizability and ease of train of recently proposed feed-forward mechanisms based on self-attention, such as the Transformer, with the learning abilities of recurrent NNs. This is a strong paper, in our opinion, with a thorough experimental part and the potential for significant practical impact. Though it received three positive reviews, two reviewers would like to see a more thorough theoretical analysis of the proposed architecture (which is, admittedly, beyond the scope of the paper). One of the reviewers states “I miss a proof that the Universal Transformer is computationally equivalent to a Turing machine.” while the other states “I am having trouble understanding the universal aspect of the transformer”. Our paper brings light into this, by showing what are some of the minimal sets of features that make self-attention networks, in particular, the Transformer, Turing-complete. Moreover, in that paper, Neural GPUs are used as a yardstick to compare the computational power of the Transformer. Thus our paper presents a formal theoretical basis to address problems that are currently being discussed at ICLR. (We emphasize that we are not involved in any way with the “Universal Transformer” paper, and that we are not reviewers of it.)

Below we provide detailed responses to each one of the individual reviews.

---

> ### Author Response · Authors · 2018-11-13
> **Responses to AnonReviewer3**
>
> Responses to AnonReviewer3:
>
> ** [comment] “Albeit the paper presents an original and significant theoretical progress and is well written, it is not fit for ICLR, primarily as the paper is impossible to review and verify without a thorough perusal and analysis of the appendix. Although the results and the proof sketches fit the body of the paper, the necessity of verifying proofs makes this paper 23 pages long and makes it a better fit for a journal and not a conference.”
>
> [response] We included the appendices to allow the interested reader to see the techniques used in our theoretical proof and potentially extend it or apply it to other architectures, to understand the full implications of the results, and to validate the results for themselves. We see the proofs in our appendix more as a “companion code to backup our findings” as one usually do for an experimental paper, and we include it mostly for reproducibility purposes. As we stated in the general comments, although submitting to a journal is an option, we do want  to discuss the theoretical implications of our work face-to-face with people of the interested community without waiting for a long journal review process.

---

> > ### Comment · AnonReviewer3 · 2018-11-25
> > **Response to Responses to AnonReviewer3**
> >
> > Please, do not get me wrong - I do think your paper is well written, that the insights from it are important for the theoretical understanding of architectures many researchers are using, and that the results from the paper can be of practical significance (and that they should entice other theoretical result).
> >
> > However, given that this is a theoretical paper, of a theoretical contribution, the proofs in it are not akin to 'companion code' to just back up the findings, nor should they be there for 'reproducibility purposes' - they are the core of the paper. Without a verification of these proofs, there is no contribution. This is my main concern. Proof sketches seem ok, the reasoning in the main body of the paper seems sound, but without proofs that have been verified, the conclusions are open to refutation. And the verification of the proofs requires detailed perusal of the appendix, which doesn't fit into the 11 page limit proposed by ICLR.
> >
> > I would leave the opinion of whether a thorough verification of the proofs is or is not warranted in this case to area chairs. In the case of latter, I support the paper.

---

> > > ### Author Response · Authors · 2018-11-26
> > > **We empathize with your concern**
> > >
> > > We appreciate the comment. We empathize with your concern about the difficulty of checking long technical proofs in appendices, and in fact we often have to struggle with this ourselves as reviewers. Still, we decided to present our proofs in full detail so that they could be verified exhaustively by reviewers if needed. The only way we could do this was by presenting them as supplementary material. While we believe that in general the writing of the proof in the body of the paper and in the appendix is good and can be more or less easily followed, we will do our best to improve it further if the paper gets accepted.

---

> > > ### Comment · Area_Chair1 · 2018-11-30
> > > **Theoretical work**
> > >
> > > Dear Reviewer 3,
> > >
> > > To weigh in, the CFP for the conference calls for—amongst other things—"theoretical issues in deep learning", a category under which this work falls.
> > >
> > > Theoretical work requiring expansive proofs is indubitably better suited for journals for proper treatment. However, in a fast moving field, the role of conferences is to share work in this more preliminary form, provided suitable rigor has been applied in presenting and framing the work. In the case of theoretical work, this may mean that proof sketches are offered in lieu of proofs, especially in the main body of the paper, with further details to be included in the supplementary materials. This is perfectly acceptable, in my mind, as the role of the main body of the paper is to present the results and motivate them.
> > >
> > > If you feel the paper has done so appropriately, and if you agree there is space for theoretical work at ICLR in line with what i have written and what is in the CFP, I invite you to reconsider your evaluation in light of your own appreciation for the papers contributions.
> > >
> > > AC

---

> > > > ### Comment · AnonReviewer3 · 2018-11-30
> > > > **Reply to "Theoretical Work"**
> > > >
> > > > Dear Area Chair,
> > > >
> > > > As per my comment earlier, and given your comment saying the proof sketches are admissible, without the necessity to go through the main proof, I will increase my score. Essentially, that was my only issue with the paper.
> > > >
> > > > Other than that I still stick by what I've stated before - that the paper presents an original and significant theoretical progress with discoveries of practical importance, particularly as it fits well with related work corroborating said discoveries. As such, it should be welcomed to the community.

---

> ### Author Response · Authors · 2018-11-13
> **Responses to AnonReviewer2**
>
> Responses to AnonReviewer2:
>
> ** [comment] “of the simplifications and approximations used for the proof, how much does that take the model away from what is used in practice?”
>
> [response] Most of our changes are actually simplifications, which means that models as used in practice can have even more space to simulate computations. Take for example the relationship between Uniform Neural GPUs that we use, and (regular) Neural GPUs that are used in practice. Uniform Neural GPUs have a number of parameters that cannot depend on the input size, while (regular) Neural GPUs have a number of parameters that depend linearly on the input size. Transformer on the other hand can use multiple heads per layer but we only use one head. For the case of the Transformer one difference is that we use additive attention in our proof while multiplicative attention is used in practice most of the time. A detailed comparison between both uses in terms of computational power is a good topic for future research.
>
> ** [comment] “For example, the assumption of the piecewise linear sigmoid seems like a quite big change, as there are large regions of the space which now have zero gradients. If you run a real implementation of these models, with the normal sigmoid replaced by this one, does training still work? If not, what are the implications for the proof?”
>
> [response] This is a really interesting question. For the case of the Neural GPU, as we mention in the paper, there is a recent work by Freivalds and Liepins [2] showing that piece-wise linear activations dramatically increase the training performance. These activations (along other changes) allowed the learning of decimal multiplication from examples which was impossible with the original Neural GPU [2]. Thus  having piecewise linear activations actually helps in practice. For the case of the Transformer more experimentation is needed to have a conclusive response. We will add some comments on this in the next version of the paper.
>
> ** [comment] “[...] all floating points on a computer represent rationals, but it would be interesting to get a better understanding on how the lack of infinite precision rationals on real hardware affects the main results.”
>
> [response] This is similar to a comparison between a computer with bounded vs unbounded memory. With bounded memory a computer is, theoretically, just a finite state machine. Similarly, with rationals of bounded precision, a Transformer is computationally very weak. Actually, your question made us realize that from our results it follows that bounded precision Transformers cannot even simulate finite automaton (this is a corollary of Proposition 3.1 in our submission). We will add a discussion on this result since it will definitely improve the paper. Thank you for the comment.
>
> ** [comment] “Does the proof rely on the input and output dimensionality being the same? Eg in the preliminaries, x_i and y_i are both d-dimensional - could this be changed?”
>
> [response] The short answer is “yes” it can be changed, as one can always pad the shorter with zeroes as a trick to make them of the same dimension. But having both of the same dimension is more of a practical concern of the architectures we use. For the case of the Transformer, the fact that the decoder puts attention over the output of the encoder, plus the use of residual connections in every layer, forces dimensions to coincide. For the case of the Neural GPU, input vectors are transformed without changing their dimensions, thus input and output vectors have naturally the same size.
>
> ** [comment] “Circular convolution definition only appears to define the values directly adjacent to the border, would it be more appropriate to define S_{h+n, :, :} = S{n, :, :}?”
>
> [response] Yes, you are right. We will include this change in the next version, thank you.
>
>
> [2] Freivalds and Liepins. Improving the Neural GPU Architecture for Algorithm Learning. NAMPI-18 (workshop at ICML-18)

---

> ### Author Response · Authors · 2018-11-13
> **Responses to AnonReviewer1**
>
> Responses to AnonReviewer1:
>
>
> ** [comment] “Results are claimed to hold without access to external memory [...] what if the problem at hand is, say EXPSPACE-complete? Then the network would have to be of exponential size [...] The whole point of Turing-completeness is that the program size is independent of the input size so there seems to be some confusion here.”
>
> [response] As stated in the paper, Turing completeness for Transformer and Neural GPU is obtained by taking advantage of the internal representations used by both architectures. We prove that the Transformer and the Neural GPU can use the values in their internal activations to carry out the computations while having a network with a fixed number of neurons and connections. For the case of Neural GPUs we even restrict the architecture to ensure a fixed number of parameters (Uniform Neural GPUs). Thus our proof actually uses a “program size which is independent of the input size” as mentioned by the reviewer. The confusion might arise because of our assumption that internal representations are rational numbers with arbitrary precision; we are trading external memory by internal precision. This is a classical assumption in the study of the computational power of neural networks (e.g. Universal Approximation Theorem for FFNs and Turing Completeness for RNNs). We mention this property in the Introduction, in the Conclusions, and also when formally proving the results, but we will make it more explicit in the next version of the paper.
>
> ** [comment] “The paper is technically very heavy [...] I believe a major revision to the paper might be necessary in order to clarify the ideas.”
>
> [response] It is true that the paper is a bit dense, but we prove a technically involved result. To be precise in our claims we needed to include all the definitions in the paper. Moreover, our formal definitions can be used in the future to prove more properties for these and similar architectures with theoretical and practical implications. Though technical, the two other reviewers explicitly mention that the paper is well written.
>
> ** [comment] “The paper [...] gives very little insight and intuition behind the results.”
>
> [response] The main intuition in our results is that both architectures can effectively simulate an (Elman)RNN-seq2seq computation, which by Siegelmann and Sontag’s classical result [1] are Turing complete when internal representations are rational numbers of arbitrary precision. We mentioned this in the Introduction and in each proof sketch, but we will make it more explicit in the next version of the paper.
>
> **[comment] “I would even suggest to split the paper into two, one about each architecture”.
>
> [response] We wanted to have both architectures in the paper as they are two of the most popular architectures in use today, yet based on different paradigms; namely, self-attention mechanisms and convolution. We wanted to understand to what extent the use of these features could be exploited in order to show Turing completeness for the models. Moreover, the computational power of Transformers has been compared with that of Neural GPUs in the current literature, but both are only informally used. We wanted to provide a formal way of approaching this comparison.
>
> [1] Siegelmann and Sontag. On the computational power of neural nets. JCSS-95

---

### Public Comment · (anonymous) · 2018-12-12
**Result in this paper is potentially wrong...**

It's observed in [1] that Transformer is not universal. Also, the proofs in this paper are very technical without any intuitive explanation. The results seem very questionable. It is definitely necessary to address this concern before this paper can be accepted.

[1] https://openreview.net/forum?id=HyzdRiR9Y7&noteId=HyxfZDmCk4

---

> ### Public Comment · (anonymous) · 2018-12-13
> **Transformer with fixed-precision is not Turing-complete**
>
> We are the authors of the Universal Transformer paper ([1] above). As this comment is very similar to what was posted on that submission, please see our response there:  https://openreview.net/forum?id=HyzdRiR9Y7&noteId=HyxfZDmCk4&noteId=rkginvfklN
>
> The TLDR is that in this work the authors assume arbitrary-precision arithmetic, whereas in our case we focus on the fixed-precision setting and provide a fairly short and intuitive counterexample showing that the Transformer is not universal in that setting, whereas the Universal Transformer is (see our comment above). Our main focus in that work, however, is to show how this increased theoretical capacity leads to significant practical advantages by expanding the number of tasks the Transformer can solve, and by improving accuracies on multiple real-world sequence-to-sequence learning tasks such as MT.

---

> ### Author Response · Authors · 2018-12-13
> **On your comment**
>
> Thanks for your comment.
>
> We believe that your doubt has been already clarified by the authors of the paper mentioned in your comment ("Universal Transformers"), and we thank the authors for their response. We just want to emphasize that our results only hold when unbounded precision is admitted, which is a standard assumption in the theoretical analysis of the computational power of neural networks (see, e.g., the Universal  Approximation Theorem, or Turing Completeness for RNNs). As mentioned in the response provided by the authors of the Universal Transformers paper, when only bounded precision is allowed, then the model is no longer Turing complete. In fact, we formally prove in our paper that the latter holds even if one sees the Transformer as a seq-to-seq network that produces an arbitrary long output.  We will include some further comments about this in the final version of our paper.

---

### Meta-Review · Area_Chair1 · 2018-12-13
**Interesting theoretical work**

**Confidence:** 5
**Recommendation:** Accept (Poster)

**Metareview:**

This paper provides a theoretical analysis of the Turing completeness of popular neural network architectures, specifically Neural Transformers and the Neural GPU. The reviewers agreed that this paper provides a meaningful theoretical contribution and should be accepted to the conference. Work of a theoretical nature is, amongst other types of work, called for by the ICLR CFP, but is not a very popular category for submissions, nor is it an easy one. As such, I am happy to follow the reviewers' recommendation and support this paper.